# Deep brain stimulation of symptom-specific networks in Parkinson's disease

Nanditha Rajamani [1] ✉, Helen Friedrich[2,3], Konstantin Butenko [2], Till Dembek[2,4], Florian Lange[5], Pavel Navrátil[5], Patricia Zvarova[1,6], Barbara Hollunder [1,6,7], Rob M. A. de Bie[8], Vincent J. J. Odekerken[8], Jens Volkmann[5], Xin Xu[9], Zhipei Ling[10], Chen Yao[11], Petra Ritter [1,6,7,12], Wolf-Julian Neumann[1], Georgios P. Skandalakis[13,14], Spyridon Komaitis[14,15], Aristotelis Kalyvas[14,16], Christos Koutsarnakis[14], George Stranjalis[14], Michael Barbe[4], Vanessa Milanese[17,18,19], Michael D. Fox [2,20,21], Andrea A. Kühn [1,6,7], Erik Middlebrooks[22], Ningfei Li [1], Martin Reich [5], Clemens Neudorfer[1,2,20,21] & Andreas Horn [1,2,20,21]

Deep Brain Stimulation can improve tremor, bradykinesia, rigidity, and axial symptoms in patients with Parkinson's disease. Potentially, improving each symptom may require stimulation of different white matter tracts. Here, we study a large cohort of patients (N = 237 from five centers) to identify tracts associated with improvements in each of the four symptom domains. Tremor improvements were associated with stimulation of tracts connected to primary motor cortex and cerebellum. In contrast, axial symptoms are associated with stimulation of tracts connected to the supplementary motor cortex and brainstem. Bradykinesia and rigidity improvements are associated with the stimulation of tracts connected to the supplementary motor and premotor cortices, respectively. We introduce an algorithm that uses these symptom-response tracts to suggest optimal stimulation parameters for DBS based on individual patient's symptom profiles. Application of the algorithm illustrates that our symptom-tract library may bear potential in *personalizing* stimulation treatment based on the symptoms that are most burdensome in an individual patient.

Deep Brain Stimulation (DBS) of the subthalamic nucleus (STN) is an established treatment for Parkinson's disease (PD). The efficacy of DBS on symptoms such as tremor and bradykinesia has been established in randomized clinical trials[1], but its effects on gait and other axial symptoms have been variable, even including detrimental effects of electrical stimulation under certain circumstances[2–5]. Hence, while many patients strongly benefit from DBS, not all do[6]. One reason could be that we generally target the same brain region to treat different symptoms of the disease. For instance, in STN-DBS, we surgically target a coordinate within the posterolateral part of the nucleus defined by direct imaging and/or surgical landmarks such as the Bejjani line[7].

While stimulation is adjusted postoperatively during DBS programming in a symptom-specific manner[8], this titration often follows a trial-and-error method since the optimal stimulation site for treating different symptoms is largely unknown. Furthermore, segmented electrodes with up to sixteen contacts per lead are implanted, making the programming process increasingly complex.

The notion that different symptoms of PD map to different brain regions or networks is not new[9,10]. For instance, in seminal work by the Freiburg school of stereotaxy based on 560 ablation cases between 1950 and 1958, Hassler et al. concluded that optimal control of tremor involved lesioning a loop between cerebellum (and Mollaret triangle),

the posterior nucleus ventrooralis and primary motor cortex[9]. In contrast, optimal control of bradykinesia and rigidity involved lesioning connections from pallidum to the anterior nucleus ventrooralis and a subregion of the supplementary motor area (defined by the Vogt/Hassler/Brodmann school as area 6aα). Much later, Akram et al. among others confirmed and extended these findings using DBS and modern neuroimaging methods[11,12]. Aside from DBS and lesion data but using task-based functional MRI, Helmich et al. associated Parkinsonian rest and, likely, action tremor with the cerebellothalamocortical circuit, as well[13–15].

In the light of these findings, DBS for PD could potentially be optimized by focusing on each individual symptom rather than global metrics that combine multiple symptoms (e.g., the Unified Parkinson's Disease Rating Scale, UPDRS). From this, two points follow: First, there is a need for an accurate symptom-response circuit model in stereotactic standard space. Once established, patient-specific electrode placement could be related to such a model to determine optimal stimulation settings for each patient based on their specific symptoms. Second, it may become possible to deliver treatment to *several* segregated circuits with a single DBS electrode by simultaneous stimulation of different contacts. Such complex parameter choices could benefit from automated algorithms to suggest stimulation settings that maximally improve prevalent symptoms in each patient.

Using a method called "DBS fiber-filtering,"[16] it has become possible to pinpoint connections that associate with symptom improvements following DBS on a group level. Since at first approximation, DBS is thought to act as an "informational lesion,"[17] the circuits that associate with symptom improvements might be exactly the ones that become dysfunctional as a consequence of the disorder[18]. Indeed, multiple reports have used DBS fiberfiltering to characterize the circuits that become dysfunctional in PD[19–23] and other disorders[16,24,25]. Recently, the theoretical entirety of dysfunctional tracts has been termed the "human dysfunctome", i.e. a library of circuits that may become dysfunctional in the human brain, and which lead to disorders, if they do[18,26]. In STN-DBS for PD, the general connection that emerged was a specific (hyperdirect) cortical projection from premotor cortices to the STN[19–21], as well as indirect pathway connections from pallidum to the motor STN[26]. However, a symptom-response breakdown of this connection has not yet been established. If such a model were available, it could potentially be used to personalize treatments, by stimulating the required parts of the circuit for a given patient, which would correspond to the most burdensome symptoms. This concept has been termed "network blending"[27].

Here, we pursue exactly this goal: First, we create a circuit model in stereotactic standard space for four cardinal motor symptom categories (tremor, bradykinesia, rigidity, and axial symptoms). Second, we introduce an algorithm which builds upon the circuit model and can suggest optimal stimulation parameters as a function of the baseline symptom severity profile in each patient.

## Results

### Clinical results

Our model was derived and cross-validated on a discovery dataset that consisted of cohorts from three independent centers (Table 1 & Fig. 1, left panel: Würzburg (N = 43), Amsterdam (N = 35) and Berlin (N = 51)). All patients underwent bilateral STN-DBS using 4 contact omnidirectional electrodes (Medtronic 3389) with stimulation applied to both hemispheres. Electrodes were localized and active contacts resided in the subthalamic region across all 129 patients. Figure S1 (supplementary discussion; section S1) shows native space imaging of example patients together with reconstructed electrodes. Clinical scores across all patients had an average UPDRS-III baseline score of 44.59 ± 14.30 (SD) and mean improvement of 51.71 ± 24.26%. Our results were then

**Table 1 | Demographic information of sub-cohorts analyzed in the study**

| | Aim | DBS center [N/female] | Age [years] | Disease Duration | UPDRS-III Baseline (Med OFF) [Points] | Levodopa response [%] | UPDRS-III Improvement (Med OFF DBS ON) [%] | LEDD reduction (%) | Postop imaging |
|---|---|---|---|---|---|---|---|---|---|
| Discovery Cohort (N = 129) | Establishment of multi-symptom model | Würzburg (44/12) | 60.4 ± 8.20 | 12.6 ± 4.5 | 49.84 ± 12.35 | 61.8 ± 18.4 | 49.3 ± 24.8 | 61.4 ± 25.1 | CT |
| | | Berlin (51/18)[74] | 60.0 ± 7.90 | 10.4 ± 3.9 | 38.6 ± 12.9 | 53.5 ± 17.2 | 45.3 ± 23.0 | 52.8 ± 41.6 | MRI (N = 45) CT (N = 6) |
| | | Amsterdam (34/8) | 50.53 ± 9.80 | | 12.7 ± 6.1 | 47.03 ± 15.5 | 68.8 ± 26.1 | 46.9 ± 15.7 | 41.82 ± 38.17 CT |
| Validation cohort I (N = 93) | Validation & Replication of multi symptom model | Würzburg (52/19) | 59.22 ± 3.99 | 10.84 ± 4.18 | 42.46 ± 16 | 62.39 ± 17.44 | 46.37 ± 22.23 | 50.94 ± | CT |
| | | Beijing (41/20) | 65.38 ± 7.18 | 8.85 ± 3.77 | 53.26 ± 18.05 | 52.65 ± 15.31 | 52.65 ± 15.13 | 32 ± 23 | MRI (N = 8) CT (N = 33) |
| Validation cohort II (N = 10) | Monopolar Review Data: Validation of Symptom Specificity of model | Cologne (10/4; 186 stimulation settings) | 61.54 ± 9.15 | 9.72 ± 3.03 | 44.2 ± 17.5 | 22.7 ± 14.5 | 27.99 ± 34.6 | 48 ± 11 | CT |
| Prospective feasibility cohort (N = 5) | Prospective Application of Cleartune | Würzburg (5/1) | 60.6 ± 4.98 | 14 ± 4 | 49.8 ± 22.1 | 62.8 ± 15.87 | 65.4 ± 12.05 [Cleartune: 73.1 ± 11.8] | 57 ± 24.43 | CT |

An overview of demographic information across all sub-cohorts used in the study. The discovery cohort represents the original cohort of patients used to develop the symptom library. We used validation cohorts I and II to replicate the findings from the discovery cohort. The prospective cohort included five patients, who were enrolled to test the applicability of Cleartune in a clinical setting.

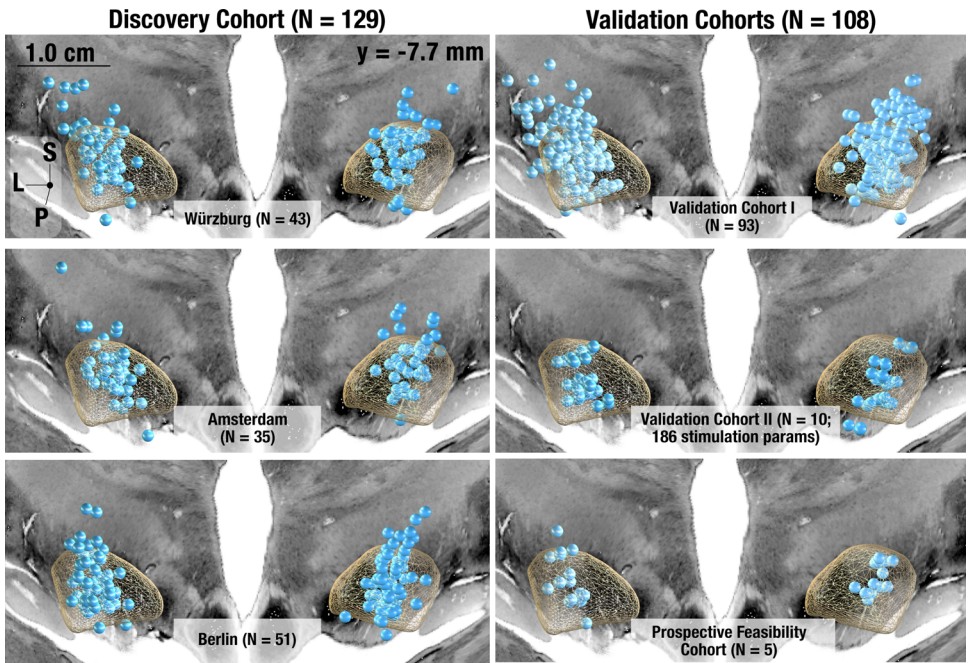

**Fig. 1 | Electrode placement.** Active contacts visualized on a coronal slice of the cortex separately for each of the three subcohorts of the discovery cohort (total $N$ = 129, left) and the three validation cohorts ($N$ = 93, 10, and 5, respectively, right). Please note that orientations here refer to Superior (S), Posterior (P), and Lateral (L).

validated using several independent cohorts (Table 1 & Fig. 1, right panel).

### Symptom-Response Multi-Tract Model (Discovery Cohort)

An extended version of the DBS Tractography atlas[28] was used to define anatomical connections from and to, as well as passing, the STN (see methods and supplementary methods; Section S3). Using the DBS fiber filtering method[29] across the $N$ = 129 discovery cohort, we investigated which stimulated streamlines (included in the pathway atlas) correlated with improvements in bradykinesia, rigidity, tremor, and axial symptoms. After FDR correction, this statistically significant set of fibers revealed a distinct rostrocaudal gradient of symptom improvements at the subthalamic level (Fig. 2).

Connections between primary motor cortex and the most posterior region of the motor STN associated with tremor improvements. When lowering the threshold (i.e. when including streamlines with correlation coefficients that did not reach significance after corrections for multiple comparisons), tremor tracts additionally included the decussating cerebellothalamic pathway. Both of these connections have been widely implicated with tremor across a large body of the literature[9,12,30–32]. Connections between pre-Supplementary Motor Area (SMA) and the anterior part of the subthalamic premotor region associated with rigidity improvements. In between, streamlines that associated with improvements of bradykinesia and axial symptoms overlapped on the anteroposterior axis. While bradykinesia tracts entered from the medial surface of the STN, axial tracts terminated at its lateral aspect (see insets in Fig. 2A) – both originating from SMA and laterally adjacent cortical regions. Prior findings hinted at shared neural substrates for bradykinesia and rigidity (in contrast to tremor)[33], which does not directly match the degree of separation our results show. To explore this further, we regressed out rigidity improvements from bradykinesia improvements and vice-versa and repeated the analysis, which led to the same segregated results (Fig. S27). Axial tracts further included a connection to the brainstem confined to the region surrounding the pedunculopontine nucleus (PPN). The PPN is a promising stimulation target to treat gait problems – which are part of the axial symptom group – although with variable success[34–36]. Given this clinical relevance, we tested whether these connections could be

specific to gait improvement (or would instead be associated with all axial symptoms). To do so, we separated gait-specific symptoms from all other axial symptoms. While gait-specific symptom improvements alone isolated the same brainstem connection, repeating the analysis with all axial symptoms *except* the gait-items did not include this connection (Fig. 3C, D).

As previously mentioned, all tracts shown in Fig. 2 were significant after correction for multiple comparisons. We still sought to test the robustness of this model further within the discovery cohort before validating it using additional data. To do so, first, we subjected symptom-response tracts to a permutation analysis. Here, the bradykinesia and rigidity tracts significantly explained more variance in outcomes than re-calculated tract models after permuting improvement values across patients 1,000 times ($p < 0.05$). Second, we subjected tract models to cross-validations. Here, all but the tremor tract model explained statistically significant amounts of variance when subjected to 10-fold cross-validations (bradykinesia: $R = 0.20$, $p = 0.02$; rigidity $R = 0.20$, $p = 0.02$; axial symptoms $R = 0.22$, $p = 0.01$, also see Fig. 2). Second, we tested how robust our results were regarding spatial inaccuracies of each stimulation site. To test this, we iteratively recalculated the symptom-response tract model 1,000 times, each time after spatially jittering each electrical field based on a 3D Gaussian distribution with 2 mm full width half maximum. Critically, this introduced random noise to the electrode placements on a group level (not all electrodes were moved in the same direction). The resulting models were highly similar to one another (and to the unjittered version) with an average mean spatial correlation of $R > 0.8$. Details and example visualizations of jittered models are shown in Fig. S28. Third, we aimed at ruling out that our results would be specific to the processing pipeline used for biophysical modeling (FieldTrip / SimBio pipeline[37] as adapted for Lead-DBS). Thus, we employed the pathway activation modeling concept using a more elaborate pipeline that was independently created by a different team, called OSS-DBS[38]. The resulting model shared a highly similar topography with the one created by our default pipeline and performing a 10-fold cross-validation yielded statistically significant correlation coefficients ($R_{\text{multitract}} = 0.38$, $p = 0.0002$; $R_{\text{singletract}} = 0.34$, $p = 0.0004$; Fig. S29).

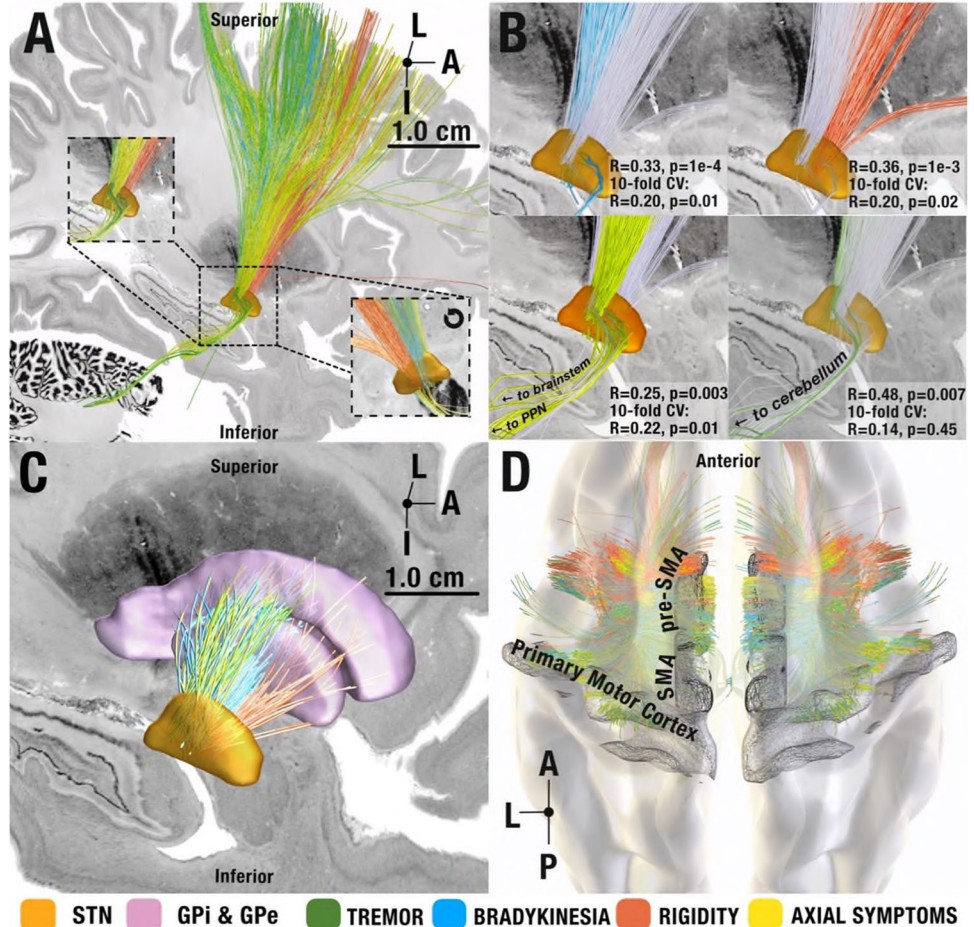

**Fig. 2 | Symptom-network library.** Views (**A**)–(**C**) from medial. **A** symptom-response tracts shown in sagittal view from medial and magnified at the level of the STN (orange, insets, one rotated by 180 degrees, i.e., shown from lateral view). Symptom-response tracts follow a rostrocaudal gradient with tremor most occipital, followed by bradykinesia, axial symptoms, and rigidity. All shown tracts significantly correlated with symptom improvements after correcting for multiple comparisons ($\alpha < 0.05$) using a two-tailed correlation analysis test. Note that tracts are in proximity to one another, making it possible to modulate all of them with a single well-placed electrode (matching clinical experience). **B** Symptom-response tracts visualized separately at the STN level with the other tracts grayed out for spatial comparison. Insets represent circular and 10-fold cross-validation results for each symptom tract. **C** Segregation of symptoms within indirect pathway streamlines between STN and pallidum, following a similar rostrocaudal gradient. **D** Cortical origins of hyperdirect projections. Streamlines associated with tremor improvements originated in primary motor cortex, whereas the ones associated with improvements in hypokinetic symptoms originated from premotor regions in a more interspersed fashion. I = Inferior, A = Anterior, L = Lateral, P = Posterior.

To further explore whether the entire model (and not each individual symptom tract) would be able to estimate variance in global motor improvements, we applied the network blending concept (see methods). Estimated global improvements significantly correlated with relative-UPDRS-III improvements ($R = 0.33$, $p = 0.00016$, mean absolute error: 17.87% ± 13.7%. Fig. 4A). To control for subcohorts within the discovery cohort, we reran the original model and applied a mixed-effects model that controlled for dataset as a random effect. Results were similar and remained statistically significant ($R = 0.30$, $p = 0.0015$). Repeating these analyses with a five-tract model (informed by tremor, bradykinesia, rigidity, gait, and other axial symptoms except gait) led to similar results (Fig. S30).

To compare the symptom-segregated model with a simpler model that was directly trained on UPDRS-III improvements, we repeated the analysis after calculating a single tract that directly coded for global %-UPDRS-III improvements. This single tract model mimicked our previous work, which aimed at determining the optimal structural connectivity profile for global motor improvement[19,20]. In direct comparison to the four-symptom model, the global symptom model performed worse ($R = 0.28$, $p = 0.0015$, mean absolute error: 18.11% ± 13.9%. Fig. 4B). When testing across multiple iterations with shuffled folds, correlations based on the multi-tract model were significantly higher than the ones based on the single tract model ($T = 93.7$, $p = 2$ e-16, Fig. S31). To rule out that the selection of individual patients in our 10-fold design did not bias the results, we repeated 10-fold cross-validations iteratively (for $N = 1000$ times) using random shuffling and observed that 5-fold and 7-fold cross validations led to similar results (details given in supplementary methods; section S4, Fig. S31).

**Symptom-Response Multi-Tract model (validation cohorts)**
To test generalizability of our model, next, we recalculated the same multi-tract model on an independent set of 93 patients from the Universities of Würzburg and Beijing *(Validation cohort I)*. Qualitatively, the result resembled the original model. Namely, connections between M1 and the STN as well as cerebellar tracts associated with tremor improvements. Axial symptom improvements correlated with streamlines adjacently anteriorly followed by the ones that associated with rigidity improvements (SMA and prefrontal regions). Using network blending, we were able to estimate variance in UPDRS-III improvements in this validation cohort purely based on the original model calculated from the discovery cohort. These estimates significantly correlated with empirical improvements in the test dataset ($R = 0.37$, $p = 0.0006$, Fig. 5C).

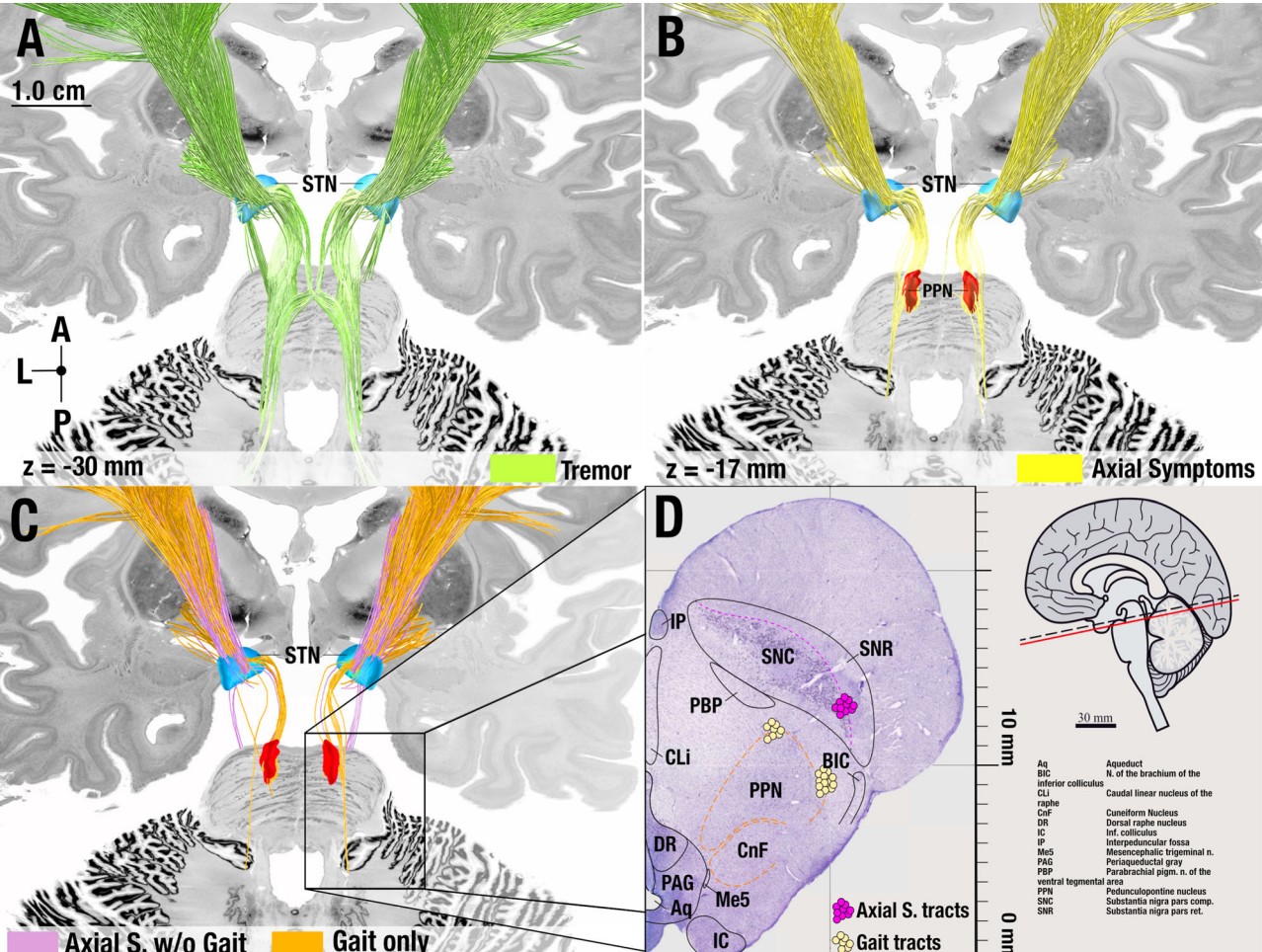

**Fig. 3 | Anatomical considerations of circuits associated with improvements of tremor and axial symptoms.** As opposed to the other figures, tracts in this figure are not thresholded at significance after FDR correction but include a broader set of tracts to appreciate the broader distribution of symptoms across streamlines (lower threshold). **A** Tremor tracts included projections from the cerebellar nuclei to thalamus as well as the cortical projections from primary motor cortex to STN, matching current pathophysiological models of tremor[32]. **B** Tracts associated with axial symptoms included a brainstem connection to the pedunculopontine nucleus region. **C** Segregating axial symptoms into gait vs. all other (axial) items revealed that this connection was driven by gait (and not by other axial symptoms). **D** Comparison to the projection site with a matching slice from a histological atlas published by Coulombe and colleagues at z = +5.08 mm (panel adapted under the Creative Commons Attribution (CC-BY) license from Coulombe et al., 2021 Frontiers in Neuroanatomy[75]). A = Anterior, L = Lateral, P = Posterior.

Given the moderate strength of the correlation coefficients between the estimated improvement and empirical clinical improvements, we investigated whether a linear model considering other demographic factors could explain additional variance. To do so, we fit a linear model that additionally included UPDRS-III baseline, patient age at surgery, sex, and levodopa equivalent dose (LEDD) reduction as covariates. This model explained 25.5% of the variance in clinical improvements ($R^2 = 0.26$, $p < 10^{-6}$). The estimated improvements of the multi-tract model remained a significant regressor ($t = 3.2$, $p = 0.0017$). UPDRS-III baseline scores ($t = 3.3$, $p = 0.001$) and sex also explained statistically significant amounts of variance ($t = 3.0$, $p = 0.03$), while the other variables did not (LEDD reduction: $p = 0.43$, age: $p = 0.39$). Of note, none of these variables may be influenced due to medical practice, with the sole exception of the electrode placement and stimulation settings, which renders the multi-tract model estimates (which are based on these factors) the critical anchor point with an opportunity to potentially improve patient care (also see Fig. S35).

### *Cleartune* – an algorithm to suggest stimulation parameters

In the next step, we created an algorithm capable of suggesting optimal stimulation settings by maximizing stimulation of a specific set of symptom tracts in novel patients. Termed *Cleartune*, this algorithm tests stimulation fields based on the entire parameter space of stimulation parameters and suggests the one that receives the highest estimated improvement. Supplementary Movie 1 visualizes the process of how the algorithm tests parameters to maximize outcomes in the four symptom domains for a specific directional electrode. To test the utility of the algorithm, it was first applied to all patients within the retrospective cohort. This led to an alternate set of stimulation volumes which could be compared to the ones applied in clinical practice using spatial correlations. Here, higher spatial correlations meant greater similarity between the clinically applied E-fields and the ones suggested by the algorithm. Higher similarities correlated with better UPDRS-III improvements ($R = 0.22$, $p = 0.001$). The same was true when repeating the analysis on the *validation cohort I*, which the model had not seen ($R = 0.23$, $p = 0.03$). Intuitively, this finding may be understood as follows: In cases in which parameters suggested by *Cleartune* agreed with the clinical ones, improvement was higher than in the ones for which the two settings disagreed.

In a second step, we aimed at testing symptom-specificity of suggestions derived by *Cleartune*. To do so, we leveraged a unique dataset of 10 patients (20 electrodes; *Validation cohort II*), for which

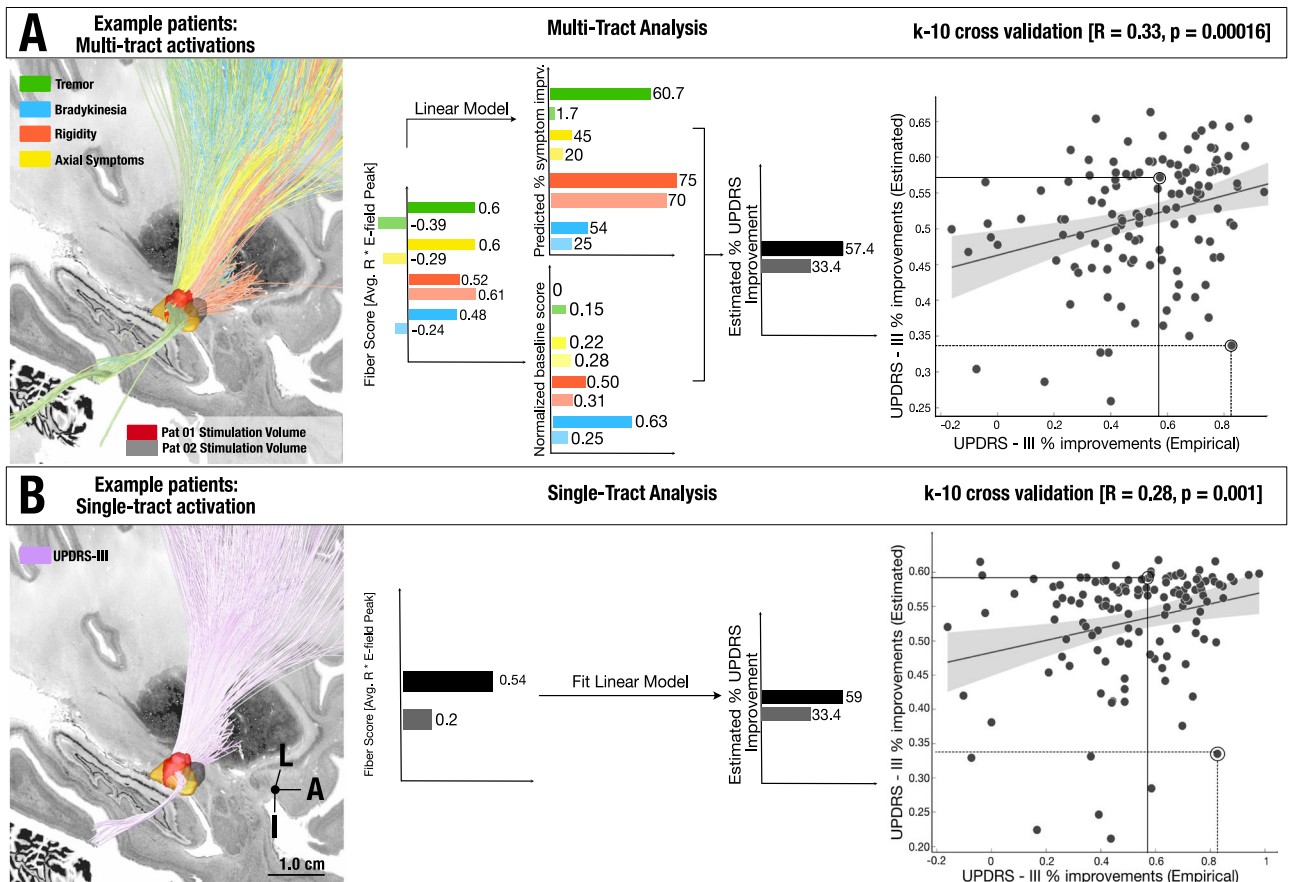

**Fig. 4 | Network Blending. A** Two example patients' stimulation volumes are shown alongside the optimal streamlines associated with symptom-response tracts. These two patient examples illustrate both extremes of the model estimation: one, where the absolute error value of model estimate is low and the other, where the absolute error is higher. To derive group level statistics, we employ the multi tract model across the four symptoms, and this process led to four scores, each coding for one symptom. These were linearly weighted by the symptoms prevalent in each patient (since, for instance, a patient with severe tremor would profit more from modulating the tremor streamlines) and averaged, leading to a weighted-average score that was converted to UPDRS-III improvements based on the training data. These estimated improvements significantly correlated with

actual improvements when analyzed via a two-tailed correlation analysis ($R = 0.33$ $p = 0.00016$, mean absolute error: 17.87%, RMSE: 0.22, $R^2 = 0.08$). **B** Stimulation volume of the same two patients shown alongside the optimal streamlines associated with global UPDRS-III improvements. These fiber scores (0.54, 0.20) were transformed to estimated values of global UPDRS-III improvements based on the training data within the 10-fold cross-validation process. These estimated improvements significantly correlated with empirical improvements, when analyzed through a two-tailed correlation analysis ($R = 0.28$, $p = 0.01$, mean absolute error: 18.11%, RMSE: 0.22, $R^2 = 0.05$). The shaded area in the correlation plot signifies the 95% confidence interval on the slope of the line. L = Lateral, A = Anterior, I = Inferior. Source data available as source data file.

multiple settings had been tested in a prospective double-blinded clinical trial ($N = 186$)[39]. These patients had been implanted with directional electrodes (Boston Scientific Vercise Cartesia) and for the directional levels with best clinical response, each segment had been tested in increasing 1 mA steps until a side effect occurred or until reaching 5 mA. In addition, the omnidirectional setting (switching on all three segments) was tested in the same way. As above, we calculated estimates for each setting using the original model (informed by the $N = 129$ discovery cohort). In 17 of the 20 electrodes, rank estimates positively correlated with clinical improvements (all correlation plots with over six data points are shown in Fig. S32). Naturally, a one-sample t-test across these R-values was statistically significant ($T = 4.155$, $p = 0.00053$; Fig. 6).

For each stimulation setting, bradykinesia and rigidity improvements were available separately. Only three of the ten cases had substantial tremor at baseline, so tremor could not be analyzed. To test for symptom-specificity, we repeated the analysis two more times, each time maximally weighting either bradykinesia or rigidity when running Cleartune optimization. The model weighted for the correct symptom led to significantly higher correlations between estimates and empirical improvements across settings in each electrode for the

correct vs. respective other symptom ($p < 0.05$ for both analyses; Fig. 6B, C).

**Prospective application of *Cleartune***

Finally, we prospectively applied DBS stimulation parameters suggested by *Cleartune* in a small set of five patients (study design shown in fig. S33). UPDRS-III scores were taken by raters that were blinded to which protocol was active and then compared between standard of care stimulation settings and the ones suggested by *Cleartune*. Figure S34 shows electrode localizations and the two stimulation protocols (*Cleartune* vs. Standard of Care; SoC) together with their tract overlaps from the multi-tract model. Detailed results are given in the Supplementary Information (Supplementary Methods; Section S5). In brief, from a baseline of 49.8 ± 22.1 UPDRS-III points, under *Cleartune* settings, scores improved by 34.4 ± 13.1 points (73 ± 11.8%). Under standard of care settings, scores improved by 31.8 ± 15.1 points (65.4 ± 12.1%). In four of the five patients, *Cleartune* settings led to a higher improvement than SoC settings. In the fifth patient, improvements were comparable (36 vs. 38 points improvement). While three of the five patients preferred *Cleartune* over SoC settings, in two patients, *Cleartune* settings led to side-effects (dyskinesia in patient 05 and

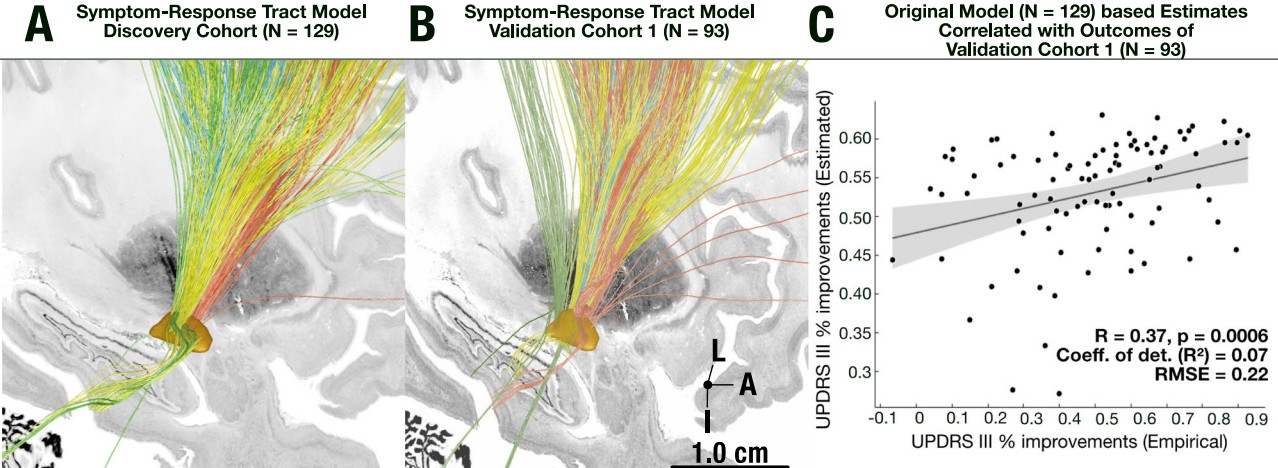

**Fig. 5 | Retrospective validation on long-term clinical outcome data. A** The fiber distribution of the original model as shown in previous figures, **B** fiber distribution when recalculating the same model on the independent test dataset (N = 93). **C** Estimation of UPDRS-III improvements in the test set (R = 0.37, p = 0.0006, $R^2 = 0.07$, RMSE = 0.22, MAE = 17.16), based on the original symptom response model, using two-tailed correlation analysis. The shaded area of the correlation plot signifies 95% confidence interval on the slope of the line. L = Lateral, A = Anterior, I = Inferior. Source data available as source data file.

dizziness in patient 04). This emphasizes that the current model was purely driven by improvements (and not by side-effects), which is a clear limitation for clinical applicability. Tracts of avoidance that code for side-effects should be added to the model in future studies. Alternatively (and additionally), clinicians may reduce the stimulation amplitude suggested by *Cleartune* in case of side-effects (while keeping the remaining parameter choices unchanged). This would still reduce the parameter space and could hence help clinicians to converge on a beneficial solution faster. While generally promising, given the low N, these results should not be overinterpreted. Rather, this trial was carried out to test feasibility of applying *Cleartune* in a clinical setting and to gather first experience in preparation for a proper prospective trial. As such, the trial was not powered to compare *Cleartune* vs. SoC settings (non-inferiority or superiority).

## Discussion

Three conclusions may be drawn from this study. Most critically, our results include the first model of symptom-response tracts in stereotactic space created in a data-driven fashion on a detailed and inclusive pathway atlas based on a large multi-center DBS cohort (N = 129). Second, we showed that this symptom-network library was robust when subjected to cross-validations and outperformed a single tract model calculated on global UPDRS-III improvements. We replicated a qualitatively similar model based on an independent additional multi-center cohort (N = 93). Third, based on the generated model, we introduced an algorithm capable of suggesting personalized and symptom-specific DBS stimulation parameters, which could similarly be validated in out-of-sample datasets and was prospectively tested in five patients. Using monopolar review data acquired in patients with segmented electrodes, we were able to demonstrate symptom-specificity of the algorithm. Namely, a model tuned to estimate bradykinesia outcome performed better in estimating bradykinesia compared to rigidity outcomes, and vice versa.

Our results support the notion that different networks may correlate with improvements of cardinal symptom categories in Parkinson's Disease. Our results may segregate the basal ganglia thalamocortical motor loop by symptoms arranged along a rostro-caudal gradient within the sensorimotor-premotor functional zone of the STN. In doing so, they extend a recently published model of the "human dysfunctome"[18,26], which aims at describing circuits that may become dysfunctional within based on brain disorders, and hence become responsive to functional suppression by DBS. In comparison

to the original report, this present refinement of the "Parkinsonian loop" breaks down individual symptoms along the general circuit which was identified by a global clinical motor score (UPDRS-III) in the study by Hollunder et al. While interspersed on a subthalamic level, each of the symptom-specific tracts predominantly originated from different cortical regions. Further, tremor connections included cerebellar projections while axial symptoms included connections to the PPN region in the brainstem.

It is important to clarify at this point that our results do not suggest that one symptom domain can be modulated independently or exclusively by a specific set of streamlines. There were considerable overlaps between connections, especially on the cortical level and along the indirect (pallidosubthalamic) projections. On the other hand, projection zones of hyperdirect (cortical) input to the STN seemed segregated. At first glance, this could appear to be contradictory to clinical experience: Indeed, the same DBS setting typically modulates many symptoms at once, seemingly with similar intensity. However, this experience does not conflict with our results: the identified tracts reside very close to one another, spanning across a region of millimeters within the sensorimotor functional zone of the STN level. As Fig. 7A shows, a single well-placed electrode may produce a stimulation volume that modulates all identified tracts (and hence symptoms), simultaneously. However, Fig. 7B shows potential use of the tract model with a modern 16-contact segmented electrode (such as the Boston Scientific model Cartesia X). Using Multiple Independent Current Control technology, distinct stimulation volumes may be generated along the same electrode, each with different amplitudes and frequencies[40]. In the hypothetical example shown in Fig. 7, one could steer a first volume at high frequency (180 Hz) to the tremor streamlines and a second at low frequency (25 Hz) to the axial & gait streamlines to treat the two symptoms as optimally, as possible.

Hence, we argue that our results could potentially become clinically relevant: First, segmented electrodes allow for increasingly refined steering of the stimulation field. This leads to an explosion of the parameter space where imaging-guided algorithms will become indispensable[41,42]. With imaging methods and electrode localizations becoming ever more precise, we are poised to use symptom-response multi-tract models such as the present one to fine-tune stimulation settings depending on the symptom profile of each patient. Second, while a single stimulation site of a well-placed electrode may cover the majority of symptom-tracts we identified, it may still matter where the focus of the electric field resides.

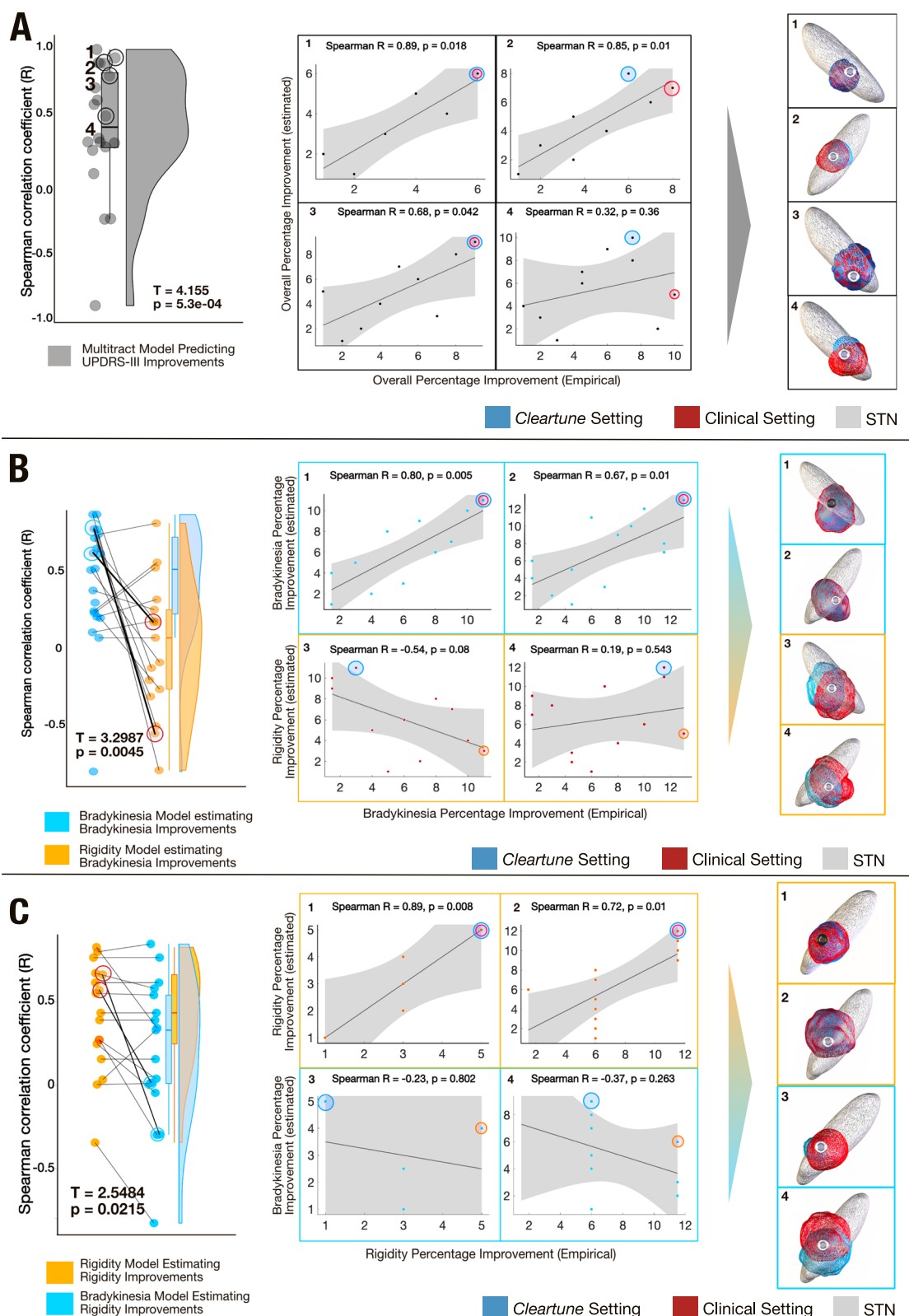

To develop this further, we introduced an algorithm, *Cleartune*, capable of suggesting symptom-specific DBS parameters based on the multi-tract model. *Cleartune* allows to set weights of symptoms that are both prevalent and burdensome for an individual patient. For instance, one could run the algorithm with a symptom profile of a typical tremor-dominant patient with high weighting of tremor and lower weighting of bradykinetic-rigid symptoms, which would favor settings that maximally target the tremor connections from primary motor cortex and cerebellum (Supplementary Movie 1).

Finally, the current approach of personalization (originally proposed in a perspective article[27]) is worth discussing. Namely, it is natural to associate the concept of "connectome based personalization of DBS" with the idea of scanning patients and analyzing *patient-specific* tract anatomy using diffusion-MRI based tractography[43–45]. While this approach is certainly promising and our present concept does not

**Fig. 6 | Retrospective validation on monopolar review dataset. A** The left panel illustrates a raincloud plot where each data point represents a Spearman's correlation coefficient between estimated and empirical UPDRS-III improvements for settings in one of the 20 electrodes. A one sided t-test is significant, illustrating that Spearman's rho is positive across most electrodes ($T = 4.15$, $p = 5.3e-04$, Average $R = 0.41 \pm 0.44$). All correlation plots are shown in Fig. S32. The right panel gives four representative examples. A red eclipse is used to represent the stimulation contact that renders the highest improvement for a given electrode, while the contact chosen by the model is marked with a blue eclipse, corresponding stimulation fields are shown for the example electrodes. **B, C** To assess symptom-specificity of the model, the analysis was repeated, this time maximally weighting either bradykinesia or rigidity symptoms, respectively. Correlations across settings

in the 20 electrodes were almost all positive when the model was used to estimate improvements in the correct symptom, but significantly dropped when used to estimate improvements in the respective other symptom. In each panel, two representative examples of correct vs. incorrect symptom pairings are given. In both (**B**) and (**C**), the T value is derived from a paired t-test between the Spearman's rho for estimated improvements in the correct symptom (for instance, when the model, trained on bradykinesia improvement estimated empirical bradykinesia improvement, $T = 3.2987$, $p = 0.0045$) vs estimated improvements in the incorrect symptom (for instance, when the model trained on bradykinesia improvements estimated rigidity improvement, $T = 2.5484$, $p = 0.02$). The shaded area of the correlation plots signifies the 95% confidence interval on the slope of the line. Source data available as source data file.

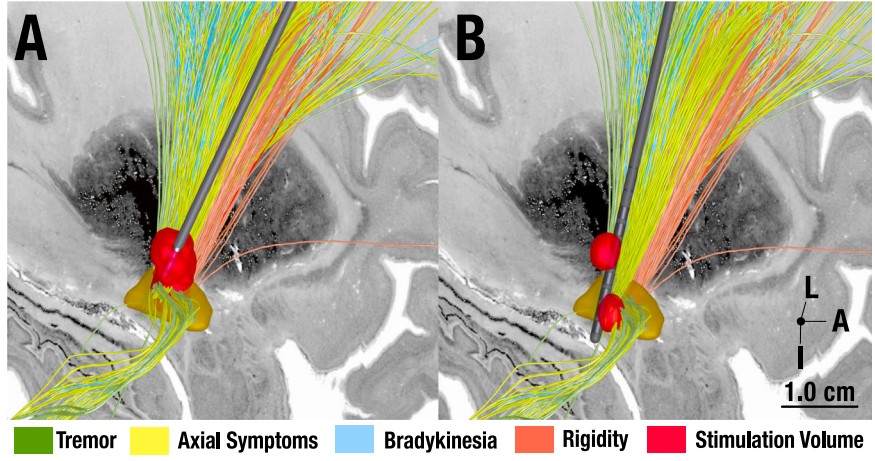

**Fig. 7 | Hypothetical future use of symptom-tract model. A** DBS Today. A well-placed, standard omnidirectional Medtronic 3389 electrode is shown with a single stimulation volume that equally covers all symptom-specific tracts. **B** DBS in the future. A hypothetical future concept with a modern electrode (Boston Scientific Cartesia X electrode with 15 directional and one omnidirectional contact) is shown.

With some devices, it is possible to steer multiple stimulation volumes toward individual tracts. In our example, one stimulation could target tremor streamlines (potentially with a high frequency of ~180 Hz). A second volume would focus on the axial/gait streamlines connecting to the PPN region (potentially with a low frequency of ~25 Hz). L = Lateral, A = Anterior, I = Inferior.

oppose (rather complements[46]) it, here, we propose an alternative approach to personalize therapy. Namely, we propose to first use *normative* connectivity data to identify and define symptom-specific tracts on a group level. Second, we spatially register patient data with the resulting multi-tract model to analyze how a single patient's electrode maps to it. Third, the actual personalization of the approach takes place on the level of symptoms, using a concept we termed *network blending*, in the past[27,46]. The concept is to blend – or weight – the identified symptom networks to derive an optimal stimulation target for the symptom profile prevalent in an individual patient. To speculate further, in the future, brain sensing combined with machine learning might provide immediate feedback to the DBS system that could automatically inform *Cleartune* to switch network targets based on the individual need of the patient at the time a symptom – such as tremor under stress – emerges[47]. One day, this could open new horizons to an integration of adaptive DBS technology and symptom associated connectomics, towards an individualized precision medicine approach to DBS in real-time.

## Limitations

Several limitations apply to this study. First, our main model applies normative tractograms instead of patient-specific tractography data to isolate symptom-specific networks. The reasons to focus on normative datasets are manifold: It is challenging, if not impossible, to reconstruct thin bundles such as the ansa lenticularis, the comb fibers or the striatopallidofugal bundle based on clinical imaging data since these are thin structures that traverse through gray matter and orthogonally to the internal capsule[48–50] (also see Supplementary

Methods; Section S3. However, even in normative data, these structures may not be identifiable with submillimeter precision. Practical reasons preclude us from generating large cohorts with individualized dMRI data given the cost and logistics involved. Typical reports of patient studies that have been based on individualized dMRI data range in the order of $N < 30$[12,29,51,52], while studies that use normative tractograms were often able to pool across larger numbers of patients[16,19,24,26] (for a review see Ref. [53]). Studies that carried out direct head-to-head comparisons found similar results when using patient-specific vs. normative data[29,52]. Finally, patients suffer from a movement disorder that leads to higher movement artifacts than in healthy controls and most patients are unable to lie still in the scanner for longer periods of time that would allow for research scan protocols[51]. Here, we created an atlas that was directly compared to anatomical data from Klingler dissections and textbook results (see Supplementary Methods; Section S3). While we believe this to be the only viable way to compare tractography results to "ground-truth" data, the comparison is indirect in nature and is based on visual inspection by anatomists and neurosurgeons. Furthermore, the identified tracts represent group averages, and it is currently impossible to match them to the exact tracts present in the individual patient. Despite this, the use of normative connectomes is inherently limited and does not include patient-specific variability of white-matter tracts. Relatedly, the use of normative tractograms includes the necessity to register patient and atlas data, which is inherently prone to inaccuracies. In other words, a patient scan can never be perfectly aligned with an atlas, despite all efforts. This leads to inaccuracies of the model and, as a function of that, to its predictive

power, i.e., it biases our results toward non-significance. Relatedly, DBS electrode reconstructions should be seen as models that inherently include an amount of uncertainty. To this end, we tested robustness of the model to uncertainty in lead localizations by repeating analyses after adding a spatial jitter to the stimulation volumes. Also, we applied a modern pipeline specifically built for the process, which includes multispectral normalizations[21] using a protocol that reaches the accuracy of expert raters[54] and that were further manually refined, when necessary, using the WarpDrive tool included in Lead-DBS[55]. Further, brain shift corrections[56] and a phantom-validated automated electrode localization algorithm were applied[57]. Using this setup, it has recently been shown feasible to obtain accurate and largely observer-independent reconstructions of DBS electrodes in both native and standard space[58]. Next, our model only considers improvement scores and currently ignores side-effects. As the prospective application shows, this is a clear limitation of the algorithm that limits its utility in clinical practice. While side-effect data were not available for this retrospective multi-center cohort, this limitation warrants additional steps to improve the model (i.e. to include tracts of avoidance that are associated with capsular effects, speech problems or cognitive/affective disturbances[23]). Next, the bioelectrical model employed here is simple compared to other methods[59,60] and has not been directly validated using electrophysiological data. Namely, while the forward solution provided by the SimBio/FieldTrip pipeline[37] as employed here, solves the static formulation of Laplace's equation to estimate the electric field in an established fashion (as widely used in the EEG literature), our process ends there and we calculate statistics based on electric field magnitude. Using the electric field magnitude, rather than a binary metric, such as the stimulation volume allows to partially account for the uncertainty in axonal parameters[61]. Our reasoning behind choosing this simpler and more probabilistic approach, which does not assume sharp borders of the stimulation field, has been described at length elsewhere[62]. However, it is key to mention that more elaborate biophysical modelling pipelines have combined volume conductor models with axonal cable models (placed orthogonally to the lead[22] or along pathways[60]) to probe in more deterministic fashion whether axons would fire action potentials in response to the DBS pulse. Even such models ignore the fact that GABAergic vs. Glutamatergic axons respond differently to DBS (the former fire along while the latter deplete readily[63]). In addition, concepts that model axons require many assumptions about the fiber type (mixed, myelinated and unmyelinated axons), axon diameters, degree of myelination, degree of arborization of both dendritic and axonal terminals, number of nodes of Ranvier to include into the model, conductivity of axonal, interstitial vs. myelin components, degree of microstructural anisotropy, heterogeneity and dispersivity of tissue conductivity, capacitive properties, and others. Despite these assumptions, more elaborate models are often deemed more "biophysically plausible" than the simpler approach applied here. To this end, we replicated our main results using a more elaborate pipeline that has been developed by a different team[38], which calculated pathway activation models, that, when subjected to fiber filtering, produced comparable results. Next, it is possible to stimulate a patient with many different parameter settings (or different contacts) and get good/similar clinical results. This matter makes demonstration of clinical utility of both out-of-sample estimates of improvements and the *Cleartune* algorithm difficult. This task is even more complicated in the present monopolar review cohort ($N = 20$), where only the three segments of a given contact level were compared (which are even closer to one another than different contact levels would be). While present results seem promising and *Cleartune* was able to suggest the clinically chosen contact despite the aforementioned difficulty, this general limitation still applies to any form of image-guided programming. Finally, correlations between model estimates and empirical

improvements are moderate. Crucially, our model was capable of estimating ranks of improvements within a given cohort, rather than absolute improvement values in individual patients. We point the reader to our modelling considerations section S6 (supplementary methods) for additional thoughts on this matter. In brief, many factors beyond electrode placement influence clinical outcomes following DBS. Critically, however, stimulation location is a key variable that *can be influenced* by doctors, while other factors (such as age, disease-subtype, etc) cannot. This isolates the variable of stimulation placement as a key one to improve patient care. Hence, despite the model not being able to predict improvements accurately, we argue that identifying optimal targets for given symptoms, as done here, is still key to move forward.

In conclusion, we created a model capable of explaining significant variance in symptom-specific effects following subthalamic DBS in PD. This model extends and refines our prior definition of the "human dysfunctome"[26]. Moreover, we introduce an algorithm capable of leveraging this model to suggest symptom-specific stimulation parameters for DBS programming, which may ultimately improve clinical outcomes and patient satisfaction.

## Methods

### Ethics declaration
The study was carried out in accordance with the Declaration of Helsinki and was approved by the institutional review board of Charité–Universitätsmedizin (retrospective analyses) and the institutional review board of University Würzburg (prospective analyses). All patients consented to the study and the sharing of their data for this study.

### Anatomical tract atlas
To carry out DBS fiber filtering based on electric fields estimated and symptom improvements across the cohort of patients, we first established a streamline atlas using various sources of information. This work is based on two published streamline atlases[28,50] that were extended to include a more exhaustive set of tracts in and around the subthalamic region. The process involved diffusion MRI based tractography on a group average template, using manually defined regions of interest, inclusion of published resources, comparisons of results with the anatomical literature, cadaveric dissection studies, histology and ex-vivo imaging. Supplementary methods: section S3 details methods and results that led to the resulting "DBS tractography atlas version 2."

### Patient cohort and imaging
232 patients who underwent STN-DBS for Parkinson's Disease (PD) were retrospectively included in this study, and 5 additional patients were enrolled prospectively. From the retrospective arm, 129 patients formed a discovery cohort (51 of whom were treated in Berlin, 43 in Würzburg and 35 in Amsterdam; patient characteristics and demographic data are provided in Table 1). Two retrospective validation cohorts consisted of 93 (52 from Würzburg, 41 from Beijing), and 10 patients, respectively. All patients in the discovery cohort were bilaterally implanted with two quadripolar DBS electrodes (model 3389; Medtronic, Minneapolis, MN). Both the validation cohorts, I and II were implanted with Vercise Cartesia electrodes; Boston Scientific, Marlborough, MA). For the prospective arm, we applied DBS settings suggested by *Cleartune* to five patients with PD who were implanted with Boston Scientific Vercise Cartesia electrodes at the University Hospital Würzburg.

Percentage improvements measured by the motor part of the Unified Parkinson's Disease Rating Scale (UPDRS-III) were calculated based on the difference between preoperative and postoperative scores divided by preoperative scores as a measure of global treatment outcome. We similarly calculated improvements of UPDRS-III items

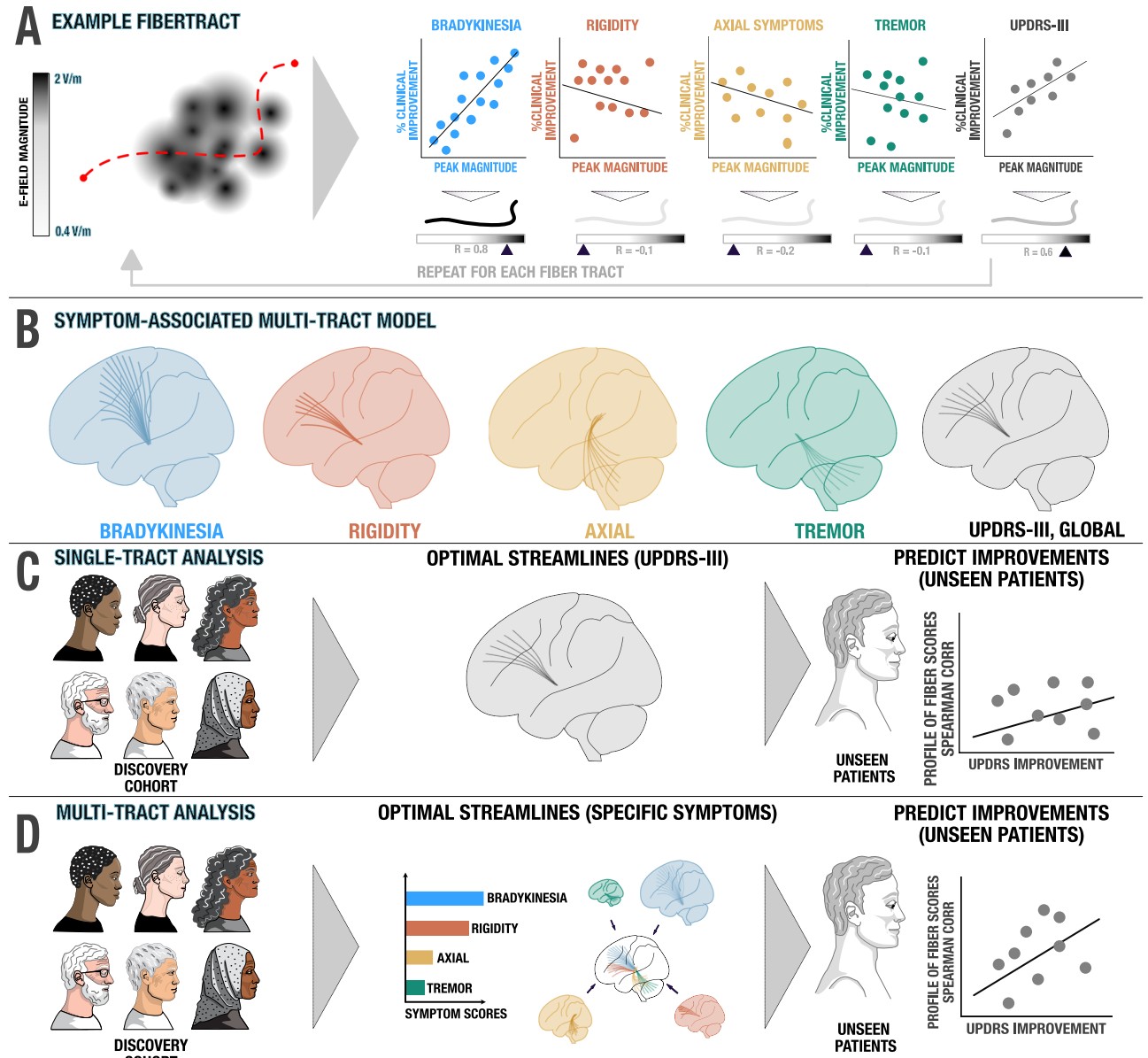

**Fig. 8 | Methods for calculating and cross-validating the multi-tract model / symptom network library. A** An example fiber tract from the pathway atlas is shown (red dashed line). For each E-field that it passes, the peak magnitude is recorded and correlated with symptom improvements. For instance, the example tract was strongly activated by E-fields that led to improvements in bradykinesia (blue scatter plot), leading to general UPDRS-III improvement (gray scatter plot). The tract is tagged by five Spearman rank correlation coefficients, one for each symptom domain, and one for global motor improvement. **B** This process is repeated across all fiber tracts in the pathway atlas to create the symptom network library. Tracts can be filtered and visualized based on the correlation coefficients (and significance values) they obtained. **C** The single tract model (coding for global motor improvements) is cross-validated by estimating motor improvements in left-out patients based on their activation of the tract. **D** A more elaborate symptom-specific tract model repeats this procedure four times (for each symptom tract) and weight estimates by baseline scores of each symptom.

that represented four major motor symptoms in PD: bradykinesia (items 23,24,25,26 which measure finger tapping, hand movement, rapid alternating and leg agility [& MDS items Toe tapping where available]), rigidity (items 22 which measure rigidity of the neck, arm & leg), tremor (items 20 and 21 of the UPDRS [& MDS tremor items: MDS postural tremor, MDS kinetic tremor, MDS tremor rest lip/jaw and constancy of rest tremor where available]), and axial symptoms (items 18, 19, 27, 28, 29, 30 which measure speech, facial, posture, postural stability, and gait [& MDS items Freezing Of Gait where available]). Gait scores (items 29 and 30) were further singled-out in a sub-analysis, as was a combination of all axial symptoms without these gait items. In our analysis, tremor sub-scores at the baseline had a high standard deviation (4.58 ± 5.24 points), when compared to the other subscore

values (Supplementary Table S5). Patients with tremor scores below two points at baseline, and those patients who improved to 100% relative improvement, were excluded from tremor analyses. Multispectral preoperative MRI scans were acquired during clinical routine to define patient-specific anatomical targets. Post-operatively, patients either underwent CT ($N = 184$) or MRI scanning ($N = 53$) to localize electrodes.

## DBS electrode localization and estimation of stimulation volume

DBS electrodes were localized using Lead-DBS software[21,62] following the revised protocol of version 3[22]. In brief, this included linear co-registration of post- and preoperative images using Advanced

Normalization Tools (ANTs)[63]. Co-registered images were then normalized into the ICBM 2009b Nonlinear Asymmetric ("MNI") template space using the ANTs SyN approach with the Effective: Low Variance + subcortical refinement protocol as implemented in Lead-DBS[21]. The results of each pre-processing step were visually inspected and refined if necessary. Normalization errors in particular were revised using the WarpDrive module available in Lead-DBS[55]. Following pre-processing, DBS electrodes were localized using the phantom-validated PaCER approach[57] for postoperative CT or the TRAC/CORE algorithm or manual localization algorithm for postoperative MRI data[62].

To estimate the stimulation volume, we calculated electric field magnitudes around the electrode (E-Fields). This was done based on a four-compartment mesh distinguishing gray and white matter, electrode contacts, and insulated parts. Gray matter regions were defined by the DISTAL atlas[64]. An adapted version of the FieldTrip-SimBio pipeline[37] was then used to solve the static formulation of Laplace's equation on a discretized domain represented by the tetrahedral four-compartment mesh. Since two fields (from the two electrodes implanted in a given patient) code for one improvement score, following the same approach as in our prior studies[20,25], electric fields were mirrored to the respective other side and both used to account for the same improvement value when running mass-univariate correlations during fiber filtering (below).

### Multi-tract implementation of DBS fiber filtering

We build upon the DBS fiber filtering concept introduced in[29] and extended in[16] to isolate tracts associated with changes across multiple motor symptom domains (Fig. 8). In the first step, we used this method to build a symptom-response multi-tract model (or "symptom network library") that associates streamlines with improvements of clinical subscores (tremor, bradykinesia, rigidity and axial symptoms). Stimulation of these four tract sets correlated with improvements in respective symptoms. To be included into the library, each tract had to pass through low number of E-fields (>0.5%) at a rather high peak intensity of >1.5 V/mm. This constraint was set up since we wanted to exclude tracts that were not strongly modulated by any stimulation field at all (which in theory could still obtain high correlation values if sub-threshold intensities correlated with clinical improvements). Changing the arbitrarily chosen values (>0.5% E-fields and >1.5 V/mm) e.g., to >2 and >4 V/mm did not qualitatively alter results. For each tract, Spearman's rank correlations were then calculated for each symptom group separately, by correlating the respective sub-score with the peak amplitude of each patient's E-field a given streamline passed through. This mass-univariate approach leads to a high number of rank correlation coefficients, which were thresholded at a $p$ value < 0.05 after correction for multiple comparisons using the false-discovery rate (FDR).

### Estimating clinical improvements based on the multi-tract model

Top 1,500 positive fibers, and top 500 negative fibers, each respectively associated with improvement and worsening of clinical symptoms were used to estimate clinical improvements (in a k-fold cross-validation design) by overlaying E-fields of left-out patients with respective symptom tracts. Here, the k (number of folds) was set to 10 since this is a standard choice in the machine learning field[65]. In each iteration, fit between the set of tracts (calculated across k-1 sets of patients) and E-fields (from the left out set) were quantified by spatially correlating the R-values of the tract landscape with the E-Field magnitudes. For each E-field, this led to a correlation coefficient for each of the symptoms. Together, these fiber scores coded for a patient-specific blend of symptom improvements. Fiber scores were further mapped to the percentage improvements in each symptom using a linear model applied to the respective training cohort. The estimated

improvement for each symptom was averaged in weighted fashion, where each weight was the normalized baseline score for the respective symptom. To exclude the possibility of strong correlations just due to the specific fold assignments, we iterated the process of 10-fold cross validation 1000 times, and each iteration resulted in a random assignment of patients into the folds. Different fold designs, i.e., k = 7 and 5, were probed, as well.

### Multi-tract implementation of OSS-DBS

OSS-DBS is an open-source toolbox for deep brain stimulation modeling based on a highly detailed volume conductor coupled with axon-cable models[38], allowing to compute pathway activations as described in Ref. 66. In this study, we used ICBM 2009b Nonlinear Asymmetric space ("MNI") for the brain segmentation to be consistent with the methodology employed in the main analysis. This brain segmentation was used to describe the electric conductivity distribution in the vicinity of the electrode. For specific frequency and tissue dependent values see[66]. Furthermore, normative diffusion data[67] were used to incorporate brain tissue anisotropy, which is largely present along white matter pathways. The electric field problem was then solved for the given stimulation protocols following the Fourier Finite Element Method[68] using the quasistatic formulation of Laplace's equation. The resulting distribution of the electric potential in time and space (along fibers of DBS Tractography Atlas, V2) described the extracellular membrane potential that was used to solve the cable equation for the widely employed mammalian axon model described in Ref. 69. Lengths of the axon models were adjusted to the lengths of the corresponding fibers, and the diameters were set to 3.0 μm, which is a compromise among the values reported in Ref. 70–72. If the model responded with an action potential, it was considered "activated". After computing such states for all fibers across all stimulation protocols, we conducted a two-sample T-test considering fibers activated in at least 5% of stimulations. The two sample T-test compared clinical improvements for the cases where the fiber was "activated" against improvements where it was not, analogous to the method employed in Ref. 16, but based on the biophysical axon model instead of the stimulation volume.

### An algorithm to suggest DBS programming parameters based on the multi-tract model

We introduce an algorithm capable of suggesting DBS stimulation parameters based on the multi-tract model (Fig. 9). This algorithm, termed *Cleartune*, attempts to solve the optimization problem to create a simulation volume such that the baseline weighted activation of the symptom pathways led to maximum estimated relative UPDRS − III improvement. Given that solving the underlying simulation model is a computationally intensive process, we employed a surrogate optimizer (*surrogateopt*, MATLAB v2022b) to solve the optimization problem. The optimizer was built to accept the current (mA) at each contact of the electrode as input. At present, the optimizer is only able to operate with cathodic stimulations, thus, all assigned currents were negative. The maximum allowed amplitude at each activated contact was set to −4 mA and the total current across all contacts at each iteration was set to −5 mA. These constraints were specified as linear inequalities, along with the starting point (−3 mA, 3rd contact switched on) to initialize the algorithm. We defined a lower bound for each contact as 0.1 times of the current amplitude. Currents below this threshold were set to zero to avoid unnecessarily complex settings. Further details about the optimizer are given in Supplementary Methods; Section S7. To evaluate each setting, the same outcome estimation concept as described above was applied. Hence, for each simulation setting generated by the optimizer, symptom improvements were estimated, which drove the objective function to find optimal settings. To avoid solutions where stimulation amplitudes are excessively high or low (and, as a consequence, insufficient symptom improvement or the occurrence of

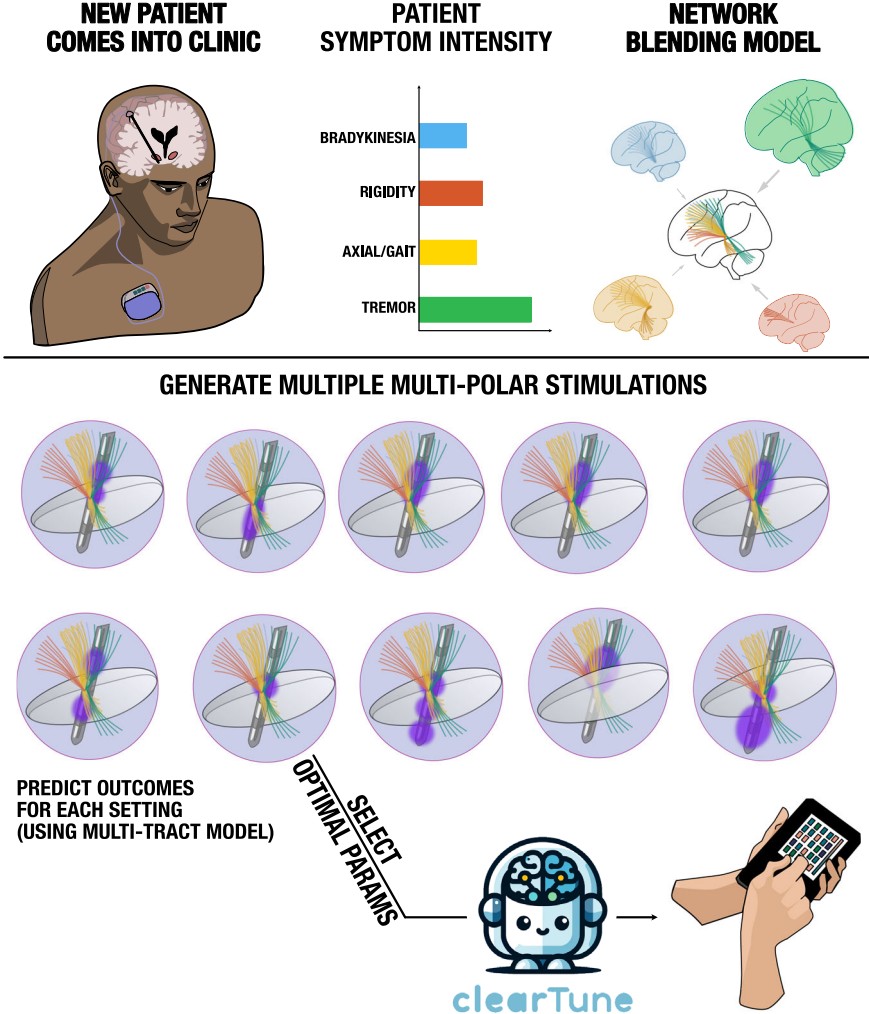

**Fig. 9 | Methodological overview of the *Cleartune* algorithm.** In this example, a novel (hypothetical) patient is treated based on *Cleartune*, who appears to be tremor-dominant. Cleartune performs network blending (Fig. 2) taking into consideration the patient's symptom profile, which directly leads to strong weights for tremor and rigidity tracts. After localization of DBS electrodes for the new patient, E-fields for various permutations and combinations of the contacts are simulated using a surrogate optimizer, and their impact on each symptom network is calculated. These impacts are weighted by the symptom profile, leading to an estimated improvement score for each solution. Finally, the solution leading to the best estimate is selected and reported to the clinician, who may consider programming the solution into the DBS pulse generator.

side-effects), we added the following penalty function to the objective function, as specified in the following equation.

$$Fval = S_{ff} + ((A - A_o)^2 * \lambda) \qquad (1)$$

*Where, Fval* is the evaluated objective function, $S_{ff}$ is the fiber score value for a given solution, $A$ is the total amplitude across all contacts in the present evaluation, and $A_o$ is the standard amplitude of 3 mA at which penalty value = 0. $\lambda$ is a penalty factor set to 0.02. Therefore, the equation applied the greatest penalty to current values that deviated furthest from −3 mA.

### Statistics & reproducibility

The sample size for the present analysis was contingent on data availability. In view of these natural restrictions on available sample sizes, we carried out sensitivity analysis for the discovery cohort using the G* power software[73]. We restricted sensitivity analysis to the discovery cohort ($N = 129$) because the primary findings of our study build upon the results from this cohort, and the rest of the data ($N = 113$) were used for validation. All analyses were carried out using a two tailed correlation test (Spearman's rho). Sensitivity tests revealed that for a sample size of 129 patients, our analysis had a power greater than 87% to detect effect sizes of 0.27−0.5, which are reported as optimal for neurological disorders in previous publications[19] when underlying an $\alpha$ of 0.05.

Our study design is largely based on previous publications[16,19], and is described in detail in the method section. The primary objective here was to develop a symptom-response model for STN-DBS in PD patients, and to establish whether this model could significantly estimate the improvement of patients which we could best evaluate by using two-tailed correlation tests. We used the non-parametric ordinal method (Spearman's rho) as a metric for our correlation analysis, since we were primarily interested in whether our model would be able to predict *ranks* of improvements and that it was capable to account for significant amounts of variance in clinical improvements of unseen patients. In other words, the aim was not an accurate prediction of clinical outcomes, but a comparison of stimulation sites and parameters among each other. No data were excluded from analysis. Given that most of our analysis were carried out on retrospective data, no randomization steps were included in most parts with the exception of the feasibility trial with

*Cleartune*, where patients were randomized and blinded to the application of *Cleartune* or Standard of Care (SoC).

To ensure that our model was well validated and to improve the chances of reproducibility, we performed two out-of-sample validations. One, with a data structure similar to the original model (validation cohort I) and another, with a repeated measures dataset (validation cohort II). The code base used for all experiments as well the connectome atlases used are made openly available within Lead DBS software[22].

### Reporting summary

Further information on research design is available in the Nature Portfolio Reporting Summary linked to this article.

## Data availability

Cohort-wise demographic and means clinical outcomes are made available in Table 1. Patient imaging data cannot be openly shared due to data sharing and privacy regulations. However, they can be made available upon request to the corresponding primary investigators who acquired the data. The corresponding author and the principal investigator (NR and AH) commit to returning data requests within a time frame of 30 days. The DBS Tractography Atlas, version 2.1, which was developed for the present study can be openly downloaded (https://github.com/netstim/DBS-Tractography-Atlas.git). The template for the development of the DBS Tractography atlas is open via DSI-Studio (HCP-1,065; https://brain.labsolver.org/hcp_template.html; fiber orientation maps at 1-mm resolution). DISTAL atlas, v 1.1[64] which was used for visualization of the basal ganglia nuclei is openly available in the Lead DBS knowledge base and comes pre-installed in Lead DBS software. Source data are provided with this paper.

## Code availability

All code used to analyze the dataset is openly available within Lead-DBS fiber filtering software (https://github.com/leaddbs/leaddbs).

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

## Acknowledgements

We would like to thank the patients and their families for their participation, which directly led to this study being possible. WJN is funded by the European Union (ERC, ReinforceBG, project 101077060), Deutsche Forschungsgemeinschaft (DFG, German Research Foundation) – Project-ID 424778381 – TRR 295 and the Bundesministerium für Bildung und Forschung (BMBF, project FKZ01GQ1802). AH was supported by the German Research Foundation (Deutsche Forschungsgemeinschaft, 424778381 – TRR 295), Deutsches Zentrum für Luft- und Raumfahrt (DynaSti grant within the EU Joint Programme Neurodegenerative Disease Research, JPND), the National Institutes of Health (R01 13478451, 1R01NS127892-01, 2R01 MH113929 & UM1NS132358) as well as the New Venture Fund (FFOR Seed Grant). We acknowledge support of funding by the Open Access Publication Fund of Charité – Universitätsmedizin Berlin.

## Author contributions

N.R. & A.H. – Designed and implemented software necessary for analysis, performed data analysis, and wrote the manuscript. A.H. conceptualized and supervised the study; H.F. – validated the anatomical tract atlas, wrote, reviewed, and edited the manuscript; T.D. & K.B. – designed and implemented the software necessary for validation, reviewed and edited the manuscript; M.R., P.N. & F.L. – provided data for and performed the feasibility trial, and edited the manuscript; P.Z. – Assisted in data analysis; C.N. and B.H. – edited the manuscript and provided feedback on analysis; X.X., Z.L., C.Y. – provided data for validation of the study; G.P.S., S.K., A.K., C.K., G.S., V.M. E.M. – provided data for validation of the tract atlas; M.B., A.A.K., R.M.d.B., J.V., V.J.O. – provided data for validation and reviewed the manuscript; P.R., J.N., N.L. – provided supervision and reviewed the manuscript; M.D.F. – reviewed and edited the manuscript.

## Funding

## Competing interests

N.R., B.H. and A.H. serve as inventors on patent application by Charité – University Medicine Berlin that covers multitract fiberfiltering and the cleartune algorithm introduced in this work. The application has been submitted on July 21, 2023, with the patent office of Luxembourg (application #LU103178). A.H. reports lecture fees from Boston Scientific and is a consultant for FxNeuromodulation and Abbott unrelated to present work. W.J.N. received honoraria unrelated to present work from Medtronic that is a manufacturer of deep brain stimulation devices. E.M. declares the following funding sources and employment: Boston Scientific Corp: Advisory Board Member, Research Support, Varian Medical Systems: Clinical Trial Funding, Advisory Board Member, Speaker's Bureau; Vigil Neuroscience, Inc: Clinical Trial Funding. A.A.K. reports personal fees from Medtronic and Boston Scientific. The remaining authors declare no competing interest.

## Additional information

[1]Movement Disorder and Neuromodulation Unit, Department of Neurology, Charité-Universitätsmedizin Berlin, corporate member of Freie Universität Berlin and Humboldt-Universität zu Berlin, Berlin, Germany. [2]Center for Brain Circuit Therapeutics Department of Neurology Brigham & Women's Hospital, Harvard Medical School, Boston, MA, USA. [3]University of Würzburg, Faculty of Medicine, Josef-Schneider-Str. 2, 97080 Würzburg, Germany. [4]Department of Neurology, University of Cologne, Cologne, Germany. [5]Department of Neurology, University Clinic of Würzburg, Josef-Schneider-Str. 11, 97080 Würzburg, Germany. [6]Einstein Center Digital Future, Berlin 10117, Germany. [7]Brain Simulation Section, Department of Neurology, Charité University Medicine Berlin and Berlin Institute of Health, Berlin 10117, Germany. [8]Department of Neurology, Amsterdam University Medical Center, Amsterdam, The Netherlands. [9]Department of Neurosurgery, Chinese PLA General Hospital, Beijing 100853, China. [10]Department of Neurosurgery, Hainan Hospital of Chinese PLA General Hospital, Sanya, Hainan 572000, China. [11]Department of Neurosurgery, The National Key Clinic Specialty, Shenzhen Key Laboratory of Neurosurgery, the First Affiliated Hospital of Shenzhen University, Shenzhen Second People's Hospital, Shenzhen 518035, China. [12]Bernstein center for Computational Neuroscience Berlin, Berlin 10117, Germany. [13]Section of Neurosurgery, Dartmouth Hitchcock Medical Center, Lebanon, NH 03756, USA. [14]Department of Neurosurgery, National and Kapodistrian University of Athens Medical School, Evangelismos General Hospital, Athens, Greece. [15]Centre for Spinal Studies and Surgery, Queen's Medical Centre, Nottingham University Hospitals NHS Trust, Nottingham, UK. [16]Division of Neurosurgery, Toronto Western Hospital, University Health Network, Toronto, ON, Canada. [17]Neurosurgical Division, Hospital Beneficência Portuguesa de São Paulo, São Paulo, Brazil. [18]Department of Neurosurgery, Mayo Clinic, Florida, USA. [19]Movement Disorders and Neuromodulation Unit, DOMMO Clinic, São Paulo, Brazil. [20]Harvard Medical School, Boston, MA 02114, USA. [21]Brain Modulation Lab, Department of Neurosurgery, Massachusetts General Hospital, Boston, MA 02114, USA. [22]Department of Radiology, Mayo Clinic Florida, Jacksonville, FL, USA. ✉e-mail: nrajamani@bwh.harvard.edu

