## [Peer Review File · Nature Communications]

Deep Brain Stimulation of Symptom-Specific Networks in Parkinson's DiseaseEditorial Note: Parts of this Peer Review File have been redacted as indicated to either remove third-party material where no permission to publish could be obtained, or to remove any confidential information from the file.

Reviewers' comments:

Reviewer #1 (Remarks to the Author):

The work of Rajamani et al. presents a statistical tour-de-force of the latest iteration of Lead-DBS analyses to the study of subthalamic deep brain stimulation (DBS) for the treatment of Parkinson's disease (PD). The study has several strengths in both concept and practice. These include an evaluation of the hypothesis that activation of specific pathways can be related to the modulation of specific symptoms, and the analysis of a large cohort of subjects from multiple institutions. However, the study also suffers from several fundamental flaws that make the results highly questionable, and those reservations are only reinforced by the especially weak correlations. In addition, the retrospective nature of the analyses is a major limitation. Overall, it is unclear how this paper, with the countless other papers from this group, using these ~same datasets and ~same methods, are continuing to provide any new knowledge or insight to the field.

Specific Comments:

1) Patient data. A primary strength of the study is the relatively large N. However, that number is somewhat misleading, as only a single DBS parameter setting and corresponding outcome measure is available for each subject. As such, the actual density of data is quite weak. A far more valuable situation would be if many different settings, at many different electrode contacts, were tested and evaluated for each subject. In turn, a basic flaw of this study is the highly unrealistic assumption that all subjects have a "common" response to stimulation in the same anatomical location of the brain atlas. This is almost certainly false, but there is no way to account/represent this variability with this clinical dataset.

2) Atlas brain model. The Horn group has successfully popularized the concept of using a "normative" connectome to study DBS and there are many scientific and statistical advantages to this approach. However, while the "v2 tract atlas" used in this study appears to be an interesting update, it is not something that has been reviewed/legitimized/validated in the literature, by either anatomists or imaging people. In fact, it appears to be a "Frankenstein" of many different models/methods. As such, its value/utility/appropriateness is unknown. Such things should be vetted in the literature first and then used for correlation analyses second.

3) DBS model. Any issues related to "patient-specific" vs. "atlas" analyses aside, the methods for representing "DBS" in this study are terribly flawed. While the Lead-DBS literature insists that they have "validated" their methods, the simple reality is that they have not. To my knowledge there has never been a published study dedicated to explicitly comparing the biophysical models used in Lead-DBS to any established standards or electrophysiological measurements (for either the E-field model or the neural response predictor function). The only comparisons that are available in the literature suggest that the simulations performed in Lead-DBS are grossly inaccurate. In addition, the analyses performed in this study make the ridiculous assumption that the DBS lead location is precisely known for each subject, and that there is zero variability associated with that component of the model. Given the clinical datasets used in this study, the reality is that ~2mm of anatomical uncertainty is associated with each data point in Figure 1. That anatomical uncertainty translates into far greater variability and uncertainty in the pathway activation predictions than accounted for in this study.

4) Given the weak data and methodology used in this study, poor correlation coefficients are too be expected. Nonetheless, R^2 values this bad are not real results, but instead the correlations are more likely to be just noise. Either way, their relevance to clinical DBS programming algorithm development or mechanistic understanding is minimal.

5) A single example, testing a single prediction of a model, is not a “prospective” study. The provided example is also far from compelling, as clinical experience has long known that you can stimulate a subject with many different parameter settings (or different contacts) and get good/similar clinical results. The “Cleartune” algorithm sounds promising, but similar to the “v2 tract atlas”, this tool needs to be properly vetted in a dedicated publication. In addition, the “innovation” of this tool, and the study in general, is greatly overstated. Prospective clinical testing of model-based DBS parameter selection has been going on for more than a decade, and there are already commercial products with demonstrated success. Similarly, there is nothing new/novel about the concepts of “connectomic” DBS programming for PD.

Reviewer #2 (Remarks to the Author):

The aim of this study was to investigate deep brain stimulation on PD patients

The paper is well written and clear.

I have only some minor comments.

- The authors did not give any details about diffusion tensor imaging in the three centres. Did the authors conduct sequence harmonization?

- In the same line, did the authors investigate center effect?

- The algorithm “Cleartune” is an interesting perspective and could have a clinical value. However, Cleartune appears as a prognostic black box. It is not clear which data were used to set up the algorithm, and what kind of methods is used to predict. Did the authors use the same retrospective data for learning? How the prediction is calculated? The readers needs more details about this tool and a validation prospective study.

This work showed a symptom network library and proposed a tool to suggest monopolar stimulation settings. The first part is very interesting and perfectly documented. Unfortunately, even if Cleartune appears promising, validation needs a prospective clinical study (more than one patient). May be the authors could more focus only on the first part.

Reviewer #3 (Remarks to the Author):

The ability to predict exact locations of optimal STN deep brain stimulation to improve the unique symptoms of individual patients is an important challenge in the field. This manuscript seeks to personalize deep-brain stimulation programming by examining the association of specific symptom improvement (tremor, bradykinesia, axial improvement) with stimulation of specific network targets that connect to various brain regions. The goal is to create a three-dimensional symptom-circuit model in standard stereotactic space and to predict optimal therapy by stimulating multiple contacts on a single electrode. The study builds on previously published work. At the same time, the claimed benefits lack validation and therefore appear somewhat overstated. Inclusion of a single patient with improved results when tested for a 24 hour period with algorithmic recommendations is unconvincing.

Major Comments:

1. Active contacts are visualized after atlas-based co-registration with Lead-DBS software. Figure 1 shows the striking heterogeneity of contact locations in the STN region with some contacts located far from traditional sites of stimulation. This distribution seems highly unlikely, particularly given that 2 mm shifts in stimulation result in dramatic changes in therapeutic efficacy. It suggests that the apparent variability of active contact location may reflect differences introduced through the atlas-fitting process. Validation is needed. Where postoperative MRI data is available, the matching of atlas-based localization to the MR visualized STN (or co-registration of post-operative CT and pre-operative MRI) needs to be demonstrated.

2. It is not clear how the symptom-network library was generated. An existing “DBS tractography atlas”, appears to be manually adjusted with a “more exhaustive” set of connections. The placement of expected or anticipated tracts is confusing, since the claimed streamline resolution far exceeds that of standard DTI imaging. In the discussion, the authors point this out, but do not address how the issue was resolved, allowing tight differentiation despite blurring associated with atlas-based fitting. Please clarify how the locations of these highly defined tracts were determined and validated.

3. Further, it further appears that the data were used to define the tracts and then the proposed tracts were used to interpret the same data, creating a potential circularity of logic (despite leave-one-out analysis). While using populations to refine tracts seems reasonable. At the same time, each of these should produce an average (or median) and a distribution. No such spread is shown. Spatial variability introduced by fitting to the atlas is not well quantified and its impact on the library generation is not explored.

3. Rather strong conclusions are drawn from relatively weak localization correlations, and the raw data for these correlations are not shown. For example, correlation coefficients of 0.23 and 0.18 would correspond to explaining only a tiny fraction of the variance in outcomes. This manuscript, and cited work, would be greatly strengthened with a model that explains clinical outcomes to a significant degree and then determines the portion attributable to active contact location.

4. The Cleartune algorithm seeks to optimize contact position by maximizing stimulation over symptom-

specific tracts. If successful, such an algorithm would be helpful to adapt programming as the disease progresses. The authors argue that where programming agreed with Cleartune, outcomes were better than those where programming differed. However, there are two striking issues that arise:

(1) Based on the model, the authors claim that an average of 21% improvement would be expected by using Cleartune. The amalgamation of matching and adjacent contact into a single group is questionable. If the algorithm is in fact correct, then one would expect matching > adjacent > different. These categories should be separated and should show a clear trend, if this approach is valid.

(2) In addition, there should be some explanation why, even with monopolar stimulation, that the expert neurologists in these centers were unable to identify the optimal sites of stimulation, and fell short by an astounding 21% on average. Again, this seems very unlikely, and likely overestimates the potential benefit of the algorithmic approach. A notable shortcoming of the study is that undesired side effects of stimulation are ignored. Hence, the failure to stimulate the modeled optimal site may simply be that stimulation at the site was accompanied by undesirable side effects.

5. The prospective application of Cleartune in a single patient is anecdotal and contributes little. At a minimum, these three centers could examine 10-20 patients prospectively to help validate the methodology. For each patient, individualized imaging to show the location of the active contact with respect to the MR visualized STN also should be presented to confirm the accuracy of the atlas fitting. In addition, the study should confirm the accuracy of the streamline predictions with individual data (to the degree that DTI resolution allows).

In summary, the manuscript asks important questions and pursues an interesting quantitative modeling and atlas-based approach. At the same time, the work lacks critical and necessary validation steps (confirmed accuracy of the atlas-based fit, confirmed location of symptom specific streamlines, an explanation of why programming performance was so poor, and appropriately controlled prospective data) to justify the authors claims of potentially strong benefits to patients.

Rajamani et al - Response to Reviewers
Nature communications, NCOMMS-23-08030

Points made by reviewers

Response by authors

Additions/Changes to the manuscript

Reviewer #1:

The work of Rajamani et al. presents a statistical tour-de-force of the latest iteration of Lead-DBS analyses to the study of subthalamic deep brain stimulation (DBS) for the treatment of Parkinson's disease (PD). The study has several strengths in both concept and practice. These include an evaluation of the hypothesis that activation of specific pathways can be related to the modulation of specific symptoms, and the analysis of a large cohort of subjects from multiple institutions. However, the study also suffers from several fundamental flaws that make the results highly questionable, and those reservations are only reinforced by the especially weak correlations. In addition, the retrospective nature of the analyses is a major limitation. Overall, it is unclear how this paper, with the countless other papers from this group, using these ~same datasets and ~same methods, are continuing to provide any new knowledge or insight to the field.

We would like to thank the reviewer for the thoughtful and critical comments that helped us drastically revise the manuscript. We added a large number of novel analyses and novel datasets to further strengthen the impact and rigor of our study. We hope that these additions may convince the reviewer that our manuscript may be a valuable contribution to the field.

Patient data. A primary strength of the study is the relatively large N. However, that number is somewhat misleading, as only a single DBS parameter setting, and corresponding outcome measure is available for each subject. As such, the actual density of data is quite weak. A far more valuable situation would be if many different settings, at many different electrode contacts, were tested and evaluated for each subject. In turn, a basic flaw of this study is the highly unrealistic assumption that all subjects have a “common” response to stimulation in the same anatomical location of the brain atlas. This is almost certainly false, but there is no way to account/represent this variability with this clinical dataset.

We have validated results based on the following additional datasets:

1. In a first step, we used the original model to estimate variance in outcomes within an entirely independent dataset of 93 patients from two centers for which a single setting per patient and long-term clinical outcomes were available. The dataset has the same limitations raised by the reviewer but may still demonstrate i) estimates made by the original model generalize to unseen data and ii) a comparable multi-tract model can be seen when re-calculating it on this entirely independent dataset.
2. In a second step, we used the original model to estimate variance in outcomes in a dataset that is structured following the suggestions by the reviewer. This dataset is particularly interesting since it tested improvements in the same ten patients (20 electrodes) using many settings along a segmented electrode. Within a prospective double-blind trial, the same segmented level had been tested in ring-mode (omnidirectional setting) and each segment alone, each using multiple amplitudes. First, predictions made for each setting positively correlated with empirical improvements for 19 of the 20 electrodes. Second, we could demonstrate that these estimates were symptom-specific: When tuning our model to estimate improvements for bradykinesia, it estimated bradykinesia improvements significantly better than rigidity improvements, and vice versa. Third, we used *Cleartune* to suggest the optimal contact settings for each of the 20 electrodes (solely informed by the N = 129 discovery cohort), which matched empirical choices quite well.

The following sections were added / changed:

"Symptom Associated Multi-Tract Model (Validation Cohorts)

To test generalizability of our model, next, we recalculated the same multi-tract model on an independent set of 93 patients from the University of Würzburg and Beijing (Validation cohort I). The result anatomically matched the original model. Namely, connections between M1 and the STN as well as cerebellar tracts associated with tremor improvements. Axial symptom improvements correlated with streamlines adjacently anteriorly followed by the ones that associated with rigidity improvements (SMA and prefrontal regions). Using network blending, we were able to predict UP-DRS-III improvements in this validation cohort purely based on the original model calculated from the discovery cohort. These predictions significantly correlated with empirical improvements in the test dataset ($R = 0.37$, $p < 0.001$, figure 5 C).— **results, pg.**

Figure 5. Retrospective validation on long term clinical outcome data. A) The fiber distribution of the original model as shown in previous figures, B) fiber distribution when recalculating the multi-tract model on the independent test dataset (N = 93). C) Prediction of UPDRS-III improvements in the test set based on the original symptom specific model.

"Cleartune – an algorithm to suggest stimulation parameters

In the next step, we created an algorithm capable of suggesting optimal stimulation settings by maximizing stimulation of a specific set of symptom tracts in novel patients. Termed *Cleartune*, this algorithm tests stimulation fields based on the entire parameter space of stimulation parameters and suggests the one that receives the highest predicted improvements. Video S1 visualizes the process of the algorithm testing parameters to maximize outcomes in the four symptom domains for a specific directional electrode. To test utility of the algorithm, it was first applied to all patients within the retrospective cohort. This led to an alternate set of stimulation volumes which could be compared to the ones applied in clinical practice using spatial correlations. Here, higher spatial correlations meant greater similarity between the clinically applied E-fields and the ones suggested by the algorithm. Higher similarities correlated with better UPDRS-III improvements ($R = 0.22$, $p = 0.001$). The same was true when repeating the analysis on the validation cohort I which the model had not seen ($R = 0.23$, $p = 0.03$). Intuitively, this finding may be understood as follows: In cases in which parameters suggested by *Cleartune* agreed with the clinical ones, improvement was higher than in the ones for which the two settings disagreed.

In a second step, we aimed at testing symptom-specificity of suggestions derived by *Cleartune*. To do so, we leveraged a unique dataset of 10 patients (20 electrodes; Validation cohort II), for which multiple settings had been tested in a prospective double-blinded clinical trial (N = 186)³⁰. These patients had been implanted with directional electrodes (Boston

Scientific Vercise Cartesia) and for the directional levels with best clinical response, each segment had been tested in increasing 1 mA steps until a side effect occurred or until reaching 5 mA. In addition, the omnidirectional setting (switching on all three segments) was tested in the same way. As above, we calculated predictions for each setting using the original model (from the N = 129 discovery cohort). In 17 of the 20 electrodes, predictions positively correlated with clinical improvements (all correlation plots with over six data points are shown in figure S32). Naturally, a one-sample t-test across these R-values was significant ($T = 4.155$, $p < 0.001$; figure. 6).

For each stimulation setting, bradykinesia and rigidity improvements were available separately. Only three of the ten cases had substantial tremor at baseline, so tremor could unfortunately not be analyzed. To test for symptom-specificity, we repeated the analysis two more times, each time maximally weighting either bradykinesia or rigidity in the multi-tract model. The model weighted for the correct symptom led to significantly higher correlations between predictions and empirical improvements across settings in each electrode for the correct vs. respective other symptom ($p < 0.05$ for both analyses; figure 6B and C)". – **results**

14 - 16

Fig 6. Retrospective validation on TWEED dataset. A) Left panel illustrates a raincloud plot where each data point represents the Spearman's correlation coefficient between predicted and empirical UPDRS-III improvements for settings in one of the 20 electrodes. All correlation plots are shown in figure S32. The right panel gives four representative examples. Here, a red eclipse is used to represent the stimulation contact that renders the highest improvement in a given patient, while the contact chosen by the model is marked with a blue eclipse, corresponding stimulation fields are shown for the example electrodes.

B and C) To assess symptom-specificity of the model, the analysis was repeated, this time maximally weighting either bradykinesia or rigidity symptoms, respectively. Correlations across settings in the 20 electrodes were almost all positive when the model was used to predict improvements in the correct symptom, but significantly dropped when used to predict improvements in the respective other symptom. In each panel, two representative examples of each correct vs. incorrect symptom pairings are given.

"We validate results on multiple additional datasets of various nature from different centers. Third, based on the generated model, we introduced an algorithm capable of suggesting personalized and symptom-specific DBS stimulation parameters, which could similarly be validated in out of sample datasets and prospectively tested in five patients. Using monopolar review data acquired in patients with segmented electrodes, we were able to demonstrate symptom-specificity of the algorithm. Namely, a model tuned to predict bradykinesia outcome performed better to predict bradykinesia compared to rigidity outcomes, and vice versa." – **discussion, pg.20**

Atlas brain model. The Horn group has successfully popularized the concept of using a “normative” connectome to study DBS and there are many scientific and statistical advantages to this approach. However, while the “v2 tract atlas” used in this study appears to be an interesting update, it is not something that has been reviewed/legitimized/validated in the literature, by either anatomists or imaging people. In fact, it appears to be a “Frankenstein” of many different models/methods. As such, its value/utility/appropriateness is unknown. Such things should be vetted in the literature first and then used for correlation analyses second.

We took this concern very seriously. We had put a lot of effort into documenting the anatomical thoughts and data that went into the creation of the original atlas and had originally planned to publish this separately. In this process, we consulted with multiple anatomists & neuroimaging experts. We now attach the extensive material that documents the comparison between streamlines in the atlas and anatomical ground-truth data to the supplement of the

present work. This led to the addition of co-authors that had helped us create the atlas. Since this documentation is bulky & extensive, we refrain from pasting it into the present document but refer to the supplementary material of the revised manuscript. We hope that the reviewer may appreciate the effort we undertook to create an anatomically plausible atlas model.

Beyond the supplementary material not pasted here (since extensive), the following sections were added:

" Symptom-Associated Multi-Tract Model (Discovery Cohort)

An extended version of the DBS tractography atlas¹⁷ was used to define anatomical connections from and to as well as passing the STN (see methods and supplementary section S2)". – **results, pg.8**

"Anatomical Tract Atlas

To carry out DBS fiber filtering based on electric fields estimated and symptom improvements across the cohort of patients, we first established a streamline atlas using various sources of information. This work is based on two published streamline atlases^{17,44} that were extended to include a more exhaustive set of tracts in and around the subthalamic region. The process involved diffusion MRI based tractography on a group average template, using manually defined regions of interest, inclusion of published resources, comparisons of results with the anatomical literature, cadaveric dissection studies, histology and ex-vivo imaging. Supplementary section S2 details methods and results that led to the resulting ‘DBS tractography atlas version 2’". – **methods, pg.25**

Supplementary methods, Section S2 – bulky material, not pasted here.

DBS model. Any issues related to “patient-specific” vs. “atlas” analyses aside, the methods for representing “DBS” in this study are terribly flawed. While the Lead-DBS literature insists that they have “validated” their methods, the simple reality is that they have not. To my knowledge there has never been a published study dedicated to explicitly comparing the biophysical models used in Lead-DBS to any established standards or electrophysiological measurements (for either the E-field model or the neural response predictor function). The only comparisons that are available in the literature suggest that the simulations performed in Lead-DBS are grossly inaccurate.

We appreciate this concern raised by the reviewer but also note that it is not straight-forward to address: The reviewer expresses vague criticism against the entire software-suite that was used. We must respectfully disagree with the notion that none of the Lead-DBS tools and methods have been validated. Instead, Lead-DBS is the only open-source tool in the field that is actively developed by multiple institutions, and with over 1,000 empowered studies represents by far the most often applied scientific software in the field of DBS imaging. Alternative software packages are either limited to use by collaborators of their authors (such as pyDBS and StimVision) or have with few exceptions mainly used by the authors themselves (such as DBSproc). Based on this, no other tool has undergone a similar scrutiny that comes with a true open-source package (which benefits from the “many eyes / many developers” principle). A large list of scientific colleagues in the field use and apply the software on a daily basis (<https://www.lead-dbs.org/about/publications/>) and bugs / issues are reported and fixed in a very active community of developers (<https://github.com/netstim/leaddbs>). The electrical model behind the software builds upon the SimBio / FieldTrip architecture which is one of the key standards in the field of EEG and builds upon an even larger open-source community. When it comes to validations, we are unaware of reports that show that these simulations are “grossly inaccurate” and would be very open for the reviewer pointing us to this specific literature, if possible.

Instead, we are aware of a long list of publications that have directly, or indirectly validated models derived by Lead-DBS. We attach a novel supplementary table to the manuscript which we also paste below. Since extensive, we refrain from going into details of each study, but are happy to elaborate more in case the reviewer feels appropriate. Critically, some of these validations are of indirect nature (such as predictions of clinical outcomes in unseen datasets). However, others are directly targeted to validate specific components of the pipeline (such as interrater and inter-modality comparisons of electrode placements or direct comparisons between LFP and imaging derived definitions of subcortical nuclei) or include prospective clinical trials that clinically applied Lead-DBS, successfully. In their sum, we feel that these studies may demonstrate accuracy and utility of the Lead-DBS toolbox, a table describing these studies is included in supplementary materials, table S2.

Table S2: Published reports in support of Lead-DBS methodology.

The table outlines studies which demonstrated the utility of the Lead-DBS toolbox, each validating specific aspects (or multiple aspects) of its processing pipeline. Critically, some of these validations are of indirect measure (such as predictions of clinical outcomes in unseen datasets). However, others are directly targeted to validate specific components of the pipeline (such as interrater and inter-modality comparisons of electrode placements or direct comparisons of LFP and imaging derived definitions of subcortical nuclei). In their sum, these studies may demonstrate accuracy and utility of the Lead-DBS toolbox. Abbreviations: CT: Computed Tomography, STN: subthalamic nucleus, GPi: internal pallidum, MER: Microelectrode recordings, MUA: Multiunit activity, VTA: volumes of tissue activated, PD: Parkinson’s disease, DYT: dystonia.

Validated Concept	Validated using...	Study	Notes
Electrode Localization	Phantom Validations	3	This study validated the automatic electrode detection algorithm applied in Lead-DBS using phantom data scanned in the CT.
		4	The study compared electrode localizations carried out by six raters after minimal training, showing an average difference in localizations between 0.52–0.75 mm.
	LFP-Recordings	5–7	These studies showed that peak beta-power magnitudes (and other LFP markers) localized to a common site within the STN across PD patients, which requires millimeter precision of electrode localizations.
		8	This study showed that theta-power magnitudes localized to a specific site within the GPi across DYT patients, which requires millimeter precision of electrode localizations.
		9	These studies showed that gamma-power magnitudes localized to a common site within the STN across PD patients that executed a movement task, which requires millimeter precision of electrode localizations.
	10	This study showed that high-spatial-resolution STN microelectrode electrophysiology recordings of PD patients (933 electrode trajectories) matched DBS electrode localizations obtained with Lead-DBS using imaging data.	

Validated Concept	Validated using...	Study	Notes
Directionality Detection		11	This study evidenced a strong accuracy in the position of electrode localized with Lead DBS (postoperative image reconstruction) and anatomical locations of intraoperative individual MERs (231 MERs, 144 in 34 STNs, 7 in 4 thalami, 5 in 4 ZIs, 34 in 10 SNs, 41 others) with an average difference in depth of the dorsal STN entry of 0.1 mm (standard deviation: 0.8 mm).
		12	This study showed the concordance between probabilistic electrode locations using Lead-DBS and intraoperative local field potential recordings in PD patients implanted in the STN with Vercise Cartesia directional electrodes (Boston Scientific).
	CT/MRI comparisons	4,13,14	These studies showed highly comparable results when localizing electrodes based on postoperative CTs vs. MRIs.
	Comparison to other software	14	This study compared DBS electrode reconstruction performed with Lead-DBS and Surgiplan and showed no significant difference in the relative distance of the electrode and the STN between the two methods (around 1mm coordinate difference).
	Phantom validations	15,16	These studies extensively validated the DiODE algorithm used for directionality detection using phantom and clinical datasets.
	Temporal Stability / Test-Retest Estimates	17	This study analyzed the temporal stability of directional DBS lead orientation using the DiODE algorithm implemented in Lead-DBS for 29 leads at 48 timepoints (up to 811 days). The mean difference of the orientation angles compared to the initial measurement was $-1.1 \pm 3.9^\circ$ (no significant difference), showing the constancy of the model over time and indirectly showing test-retest comparability of DiODE.
	Resolving marker ambiguity	(Dembek et al. 2021)	This study addressed the marker ambiguity of Boston Scientific directional leads. Provided sufficient CT quality and polar angles, ambiguity was resolved correctly in 100% of cases. Results were validated against stereotactic x-ray.

Validated Concept	Validated using...	Study	Notes
	Interuser reliability DiODe	(Henry et al. 2023)	This study investigated DiODe in data from two centers for both Abbott (DiODe not validated) and Boston Scientific directional leads. Intraclass correlation coefficients were >0.95 for both types of leads when using the automatic workflow but reduced to 0.88 for Abbott leads when using the manual workflow. Deviations between at least two users >30° were seen in 6.1 % of leads for Boston Scientific and 16 % of the Abbott leads. Images. Of note, none of the imaging matched the quality criteria regarding slice thickness for which DiODe has been validated.
Bioelectric Modeling	LFP-Recordings	¹⁸	The study showed a significant inverse correlation between % of the subthalamic nucleus stimulated (as modeled by Lead-DBS) and the %-change in beta burst durations.
	Comparison to alternate software	¹⁹	This study compares bioelectrical models calculated with the Lead-DBS pipeline to a more elaborate software (OSS-DBS, also used to validate findings in the present article).
	Clinical and side-effect thresholds	²⁰	This study created models based on overlaps of stimulation volumes calculated with Lead-DBS and two fiber bundles. Models indicated an activation of 50% of the hyperdirect pathway at effect threshold, and 4% of the corticospinal tract at capsular side effect threshold. Median suggestion errors for the effect threshold and side effect threshold were 1 and 1.5 mA, respectively.
Segmentation of deep nuclei	Comparison to manual expert segmentations	²¹	This study led to the current default settings of Lead-DBS which were capable of segmenting STN and GPi nuclei almost as accurately as expert segmentations (comparison to interrater DICE scores and surface distances).
		²²	External validation of the ²¹ study (above).
	MER/MUA	^{11,23}	This study demonstrated high agreement between definitions of the subthalamic nucleus by microelectrode recordings and anatomical segmentations carried out by Lead-DBS.
Brain Shift Correction Algorithm	MER/MUA	²³	This study compared the fit between microelectrode recordings before and after applying the brain shift correction implemented in Lead-DBS and showed significant increase in fit.

Validated Concept	Validated using...	Study	Notes
Sweetspot mapping	Synthetic Ground truth Datasets	24	This study compared various published concepts to carry out sweetspot mapping using a synthetic ground truth dataset. All concepts were implemented into Lead-DBS and the winning concept was chosen as the default parameter.
		25	This study compared various proposed concepts of sweetspot mapping as implemented in Lead-DBS and their utility to predict out-of-sample data in an STN-DBS cohort of patients with PD (N=95).
Behavioural Outcome Predictions	Behavioral tasks	26	This study used Lead-DBS derived models to explain variance in changes of reaction time and movement velocity in a behavioral task setting as a function of STN-DBS in 20 patients with PD. ~56-76% of variance in behavioral changes explained (out of sample data).
		27	This study used Lead-DBS derived models to explain variance in motor learning in a behavioral task setting as a function of STN-DBS in 20 patients with PD. 33% of variance in behavioral changes explained (out of sample data).
Clinical Outcome Predictions	Clinical scores	28	This study showed a significant correlation between clinical improvements (%-UPDRS-III) in PD-patients undergoing STN-DBS with i) electrode placement (distance to an optimal target coordinate), ii) VTA coverage of the STN as modeled by Lead-DBS and iii) structural connectivity to the supplementary motor area seeding from the modeled VTA
		29	This study estimated clinical improvements (% UPDRS-III in STN-DBS for PD) based on structural and functional connectivity seeding from the modeled VTA. An optimal connectivity profile was calculated on a first cohort and used to estimate outcomes in an unseen second cohort operated by a second surgeon at a different center.
		(Dem bek et al., 2019)	This study generated sweetspots for PD motor symptoms from monopolar review data and used these to predict outcomes in an external cohort of monopolar review data.

Validated Concept	Validated using...	Study	Notes
		25	This study demonstrated validity of predictive models on local, tract- and network-levels using Lead-DBS to estimate motor response in Parkinson's Disease (%-UPDRS-III).
		30	This study estimated clinical improvements (% TRS in VIM-DBS for ET) based on structural and functional connectivity seeding from the modeled VTA. The optimal connectivity profiles were used to estimate outcomes in unseen patients using a leave-one-out design. ~13-16% of variance in clinical improvements explained (out of sample data).
		31	This study estimated clinical side-effects (% BDI in STN-DBS for PD) based on structural connectivity seeding from the modeled VTA. A connectivity profile was associated with postoperative depression based on a first cohort and used to estimate outcomes in an unseen second cohort operated by a second surgeon at a different center (and vice-versa). A third test-cohort was used to further validate the model. ~10-33% of variance in clinical improvements explained (out of sample data).
		32,33	These studies established a tract that, when stimulated, would lead to improvements of OCD symptoms following DBS to either the ALIC or STN target. Results were cross-validated across targets, cohorts and centers. ~25-56% of variance in clinical improvements explained (out of sample data).
		34	External validation of the ³³ study (above).
		35	External validation of the ³³ study (above).
		36	External validation of the ³³ study (above).
		37	External validation of the ³³ study (above).
		38	Blinded validation of the ³³ study (above).
		39	This study used functional connectivity as input for a machine learning model and evidenced that connectome-based model is able to predict STN-DBS outcome in 50 patients with Parkinson's Disease.

Validated Concept	Validated using...	Study	Notes
Automatic programming	Clinical scores	40	This study showed that 6 neuroanatomical parameters computed by Lead-DBS (distance of each contact to the STN and the motor part of the STN, volume of the overlapping areas of the VTA and STN/ and motor part of the STN, and the number and ratio of fiber tracts through both the VTA and motor areas) were individually relevant to determine group differences between clinical optimal and nonoptimal outcomes. Additionally, the combined use of all 6 parameters suggested optimal contact selections with an accuracy of 73%.
		41	This study showed the relationship between the position of bilateral STN-DBS location of active contacts and clinical efficacy of the therapy on motor symptoms in 57 Parkinson's disease patients.
		42	This multicenter international study demonstrated the correlation between non-motor outcomes and DBS electrode location in 91 Parkinson's Disease patients.
		43	This study evidenced that automated data-driven algorithms (StimFit) predict stimulation parameters that lead to motor symptom control comparable to standard of care treatment in PD patients implanted in the STN.
		Nordström et al., 2022)	This study suggested stimulation parameters in silico based on a sweetspot for STN-DBS in PD. Retrospective comparison showed the method suggested the correct level in 56% of cases (25% chance level) and the best contact in 42% of cases (12.5% chance level).

In addition to this, we recalculated our main result (the multi-tract model) using an additional software package that was developed by a different team (at University of Rostock) which is also openly available and was hence available to us. The software, OSS-DBS, is capable of estimating pathway activations building upon the concept introduced by Gunalan et al. 2017 (from the reviewer's lab).

The resulting symptom-specific tract model shows the same general topography as our results calculated with the FieldTrip / SimBio pipeline. The following sections were added to document this effort:

“Third, we aimed to rule out that our results would be specific to the processing pipeline used for biophysical modelling (FieldTrip / SimBio pipeline 26 as adapted for Lead-DBS). Thus, we recom-puted results using the pathway activation modelling concept using a more elaborate pipeline that was independently created by a different team, OSS-DBS²⁷. The resulting model shared a similar topography with the one created by our default pipeline and performing a k-10 cross validation yielded significant correlation coefficients ($R_{\text{multitract}} = 0.40$, $p = 0.001$; $R_{\text{singletract}} = 0.34$, $p = 0.03$; figure S29).”

– **results, pg.12**

Multi-tract implementation of OSS-DBS

OSS-DBS is an open-source toolbox for deep brain stimulation modeling based on a highly detailed volume conductor coupled with axon-cable models⁶⁵, allowing to compute pathway activation as described in⁵⁸. In this study, we used ICBM 2009b Nonlinear Asymmetric space (“MNI”) for the brain segmentation to be consistent with the methodology employed in the main analysis. This brain segmentation was used to describe the electric conductivity distribution in the vicinity of the electrode. For specific frequency and tissue dependent values see⁶⁶. Furthermore, normative diffusion data⁶⁷ were used to incorporate brain tissue anisotropy which is largely present along large white matter tracts. The electric field problem was then solved for the given stimulation protocols following the Fourier Finite Element Method⁶⁸ using the quasistatic formulation of Laplace’s equation. The resulting distribution of the electric potential in time and space (along fibers of DBS Tractography Atlas, V2) described the extracellular membrane potential that was used to solve the cable equation for the widely employed mammalian axon model described in⁶⁹. Lengths of the axon models were adjusted to the lengths of the corresponding fibers, and the diameters were set to 3.0 μm , which is a compromise among the values reported in⁷⁰⁻⁷². If the model responded with an action potential, it was considered “activated”. After computing such states for all fibers across all stimulation protocols, we conducted a two-sample T-test considering fibers activated in at least 5% of stimulations. The two sample T-test compared clinical improvements for the cases where the fiber was “activated” against improvements where it was not, analogous to the method

employed in⁶⁴, but based on the biophysical axon model instead of the stimulation volume." – **methods, pg.28**

In addition, the analyses performed in this study make the ridiculous assumption that the DBS lead location is precisely known for each subject, and that there is zero variability associated with that component of the model.

Given the clinical datasets used in this study, the reality is that ~2mm of anatomical uncertainty is associated with each data point in Figure 1. That anatomical uncertainty translates into far greater variability and uncertainty in the pathway activation predictions than accounted for in this study.

We agree that modeling the effect of uncertainty in electrode placement is an exciting idea to empirically test how robust the multi-tract model would be as a function of electrode placement (and subsequently, stimulation volumes). The following sections were added to address this:

"Second, we tested how robust our results were regarding spatial inaccuracies of each stimulation site. To test this, we iteratively recalculated the symptom specific tract model 1,000 times, each time after spatially jittering each electrical field based on a 3D Gaussian distribution with 2 mm full width half maximum. The resulting models were highly similar to one another (and to the unjittered version) with an average mean spatial correlation of $R > 0.8$. Details and example visualizations of jittered models are shown in figure S28. Third, we aimed to rule out that our results would be specific to the processing pipeline used for biophysical modelling (FieldTrip / SimBio pipeline²⁶ as adapted for Lead-DBS)". – **results, pg.12**

Overall Model Similarity

Symptom Specific Similarity

Figure S28: To test robustness of model results as a function of spatial uncertainty in stimulation sites, we recalculated the model 1,000 times, each time after spatially jittering electric fields based on a 3D Gaussian distribution with 2 mm full width half maximum. Resulting models were highly similar (correlations of fiber weightings across fibers are shown in the figure, four example models are shown).

Correlation results. Given the weak data and methodology used in this study, poor correlation coefficients are to be expected. Nonetheless, R^2 values this bad are not real results, but instead the correlations are more likely to be just noise. Either way, their relevance to clinical DBS programming algorithm development or mechanistic understanding is minimal.

We agree with the reviewer, that at first sight, a low R^2 value may seem not clinically relevant. We outline our reasoning about the validity and significance of these findings in the newly added “Modelling Considerations” supplementary section:

S.5 Modelling Considerations

A natural question that may arise in the context of the present results is the validity of a model given the relatively low correlation coefficients (R), and consequentially, R^2 values. To discuss this question, we would like to raise the following points (Fig S35).

1. First, we need to ask how much variance DBS modeling should at best explain in the first place. Clinical improvements are not just based on electrode placement and stimulation volumes, but governed by many factors, such as disease subtype, age, sex, levodopa response, duration of disease, comorbid other conditions, etc. We estimate that these factors alone will explain ~50% of DBS response³⁴.
2. Noise in clinical scores (inter- and intra-rater test-retest reliability of the UPDRS-III) will account for another 15%.
3. Imaging resolution and electrode placement & modeling inaccuracy may explain another 15-20% of variance as mentioned by the reviewer above.
4. The use of multi-site datasets such as in the present study may add residuals that may explain another 5-10% of the variance.

Based on this assessment, we believe that ~10% explained variance (R values of ~0.3) may realistically be expected in a multi-site dataset such as the present one. However, we believe that such models are still useful because electrode placement / stimulation settings are the only factor that can be influenced, whereas other components such as patient age, disease type and disease onset are immutable. Therefore, identification of a robust, and highly optimal target that can only explain ~10% of variance would still remain an important finding in our field.

– **Supplementary Material, Section S5**

Figure S35. Modelling considerations. The pie chart shows the various factors that may influence the clinical outcomes of PD patients undergoing STN-DBS. Based on these competing factors, we concluded that the maximal amount of variance we should expect to be explainable by DBS models would be around ~10%.

In addition, we would like to note that all the nuisance variables that the reviewer mentions (in this point and above) will bias our results against significance – it is not the other way around. In other words, the assumption that “ R^2 values this bad are not real results, but instead the correlations are more likely to be just noise” does not match our knowledge about statistics.

As outlined above, we now confirm the results in two additional datasets that are completely independent from the discovery dataset. We hope that the reviewer may be more convinced by these results.

***Cleartune*. A single example, testing a single prediction of a model, is not a “prospective” study. The provided example is also far from compelling, as clinical experience has long known that you can stimulate a subject with many different parameter settings (or different contacts) and get good/similar clinical results. The “*Cleartune*” algorithm sounds promising, but similar to the “v2 tract atlas”, this tool needs to be properly vetted in a dedicated publication. In addition, the “innovation” of this tool, and the study in general, is greatly overstated. Prospective clinical testing of model-based DBS parameter selection has been going on for more than a decade, and there are already commercial products with demonstrated success. Similarly, there is nothing new/novel about the concepts of “connectomic” DBS programming for PD.**

We are unaware of commercial or scientific publications that used symptom-specific connections in an algorithm that would suggest stimulation parameters and see the innovation in this point. We still toned down the language regarding novelty.

We agree that the single case should not be overinterpreted but may demonstrate our first efforts to show a feasibility of applying *Cleartune* in clinical practice.

We have now added an additional set of clinical cases ($N = 5$) that were reprogrammed using *Cleartune* but still moved results to supplementary material to give them less weight. We further changed the language to state that this was done to show feasibility, not to prospectively validate the algorithm. As mentioned above, we additionally tested utility of *Cleartune* on an unseen retrospective dataset that applied multiple stimulation settings per patient using omnidirectional and directional settings.

The following sections were added / changed:

"Feasibility study to prospectively apply *Cleartune* in a clinical context

Finally, we prospectively applied DBS stimulation parameters suggested by *Cleartune* in a small set of five patients (study design shown in fig. S33). UPDRS-III scores were taken by raters that were blinded to which protocol was active and then compared between standard of care stimulation settings and the ones suggested by *Cleartune*. Figure S34 shows electrode localizations and the two stimulation protocols (*Cleartune* vs. Standard of Care; SoC) together with their tract overlaps from the multi-tract model. Detailed results are given in the supplementary material (supplementary section S4). In brief, from a baseline of 49.8 ± 22.1 UPDRS-III points, under *Cleartune* settings, scores improved by 34.4 ± 13.1 points ($73 \pm 11.8\%$). Under standard of care settings, scores improved by 31.8 ± 15.1 points ($65.4 \pm 12.1\%$). In four of the five patients, *Cleartune* settings led to a higher improvement than SoC settings. In the fifth patient, improvements were comparable (36 vs. 38 points improvement). While three of the five patients preferred *Cleartune* over SoC settings, in two patients, *Cleartune* settings led to side-effects (dyskinesia in patient 05 and dizziness in patient 04). This emphasizes that the current model was purely driven by improvements (and not by side-effects), which is a clear limitation for clinical applicability. Tracts of avoidance that code for side-effects should be added to the model in future attempts. Alternatively (and additionally), clinicians may reduce the stimulation amplitude suggested by *Cleartune* in case of side-effects (while keeping the remaining parameter choices unchanged). This would still reduce the parameter space and could hence help clinicians to come to a beneficial solution faster. While generally promising, given the low N, these results should not be overinterpreted. Rather, this trial was carried out to test feasibility of applying *Cleartune* in a clinical setting and to gather first experience in preparation for a proper prospective trial. As such, the trial was not powered to compare *Cleartune* vs. SoC settings (non-inferiority or superiority)". – **results, pg.20**

"Section S4. Feasibility study to prospectively apply *Cleartune* in a clinical context

To test feasibility of applying *Cleartune* in a clinical setting, a feasibility trial was carried out in a small sample of $n=5$ prospective patients. This trial was designed to include a randomization step, where the patient was blinded to the administration of *cleartune* vs. clinical settings. Clinical data, which included the pre-operative T1w, T2w, and post operative CT images was used to localize DBS electrodes in each patient. Baseline scores were taken in the stimulation and medication off states. The *Cleartune* algorithm was executed for each electrode separately, for 500 iterations each. This led to *Cleartune* settings, which were stored in the pulse generator as an additional program to the existing standard of care (SoC) setting. In the second week,

Cleartune settings or clinical settings were applied in randomized order, each for 24 hours. Resulting UPDRS-III scores were taken after 24 hours and the respective other program was switched on to be evaluated after another 24 hours. Figure S33 summarizes the trial design. Results are documented in table S3. In multiple cases, *Cleartune* suggested higher amplitudes than tolerable, and were hence reduced by the clinical team (without altering contact choices). Table S3 reports both suggested and programmed amplitudes. From a baseline of 49.8 ± 22.1 UPDRS-III points, under *Cleartune* settings, scores improved by 34.4 ± 13.1 points ($73 \pm 11.8\%$). Under standard of care settings, scores improved by 31.8 ± 15.1 points ($65.4 \pm 12.1\%$). In four of the five patients, *Cleartune* settings led to a higher improvement than SoC settings. In the fourth patient, improvements were comparable (36 vs. 38 points improvement). While three of the five patients preferred *Cleartune* over SoC settings, in two patients, *Cleartune* settings led to side-effects (dyskinesia in patient 05 and dizziness in patient 04). While generally promising, given the low N, these results should not be overinterpreted. Rather, this trial was carried out to test feasibility of applying *Cleartune* in a clinical setting and to gather first experience in preparation for a proper prospective trial. As such, the trial was not powered to compare *Cleartune* vs. SoC settings (non-inferiority or superiority).

Patient	Cleartune							Standard of care						
	Settings RH [%]	Settings LH [%]	Bradykin- -esia [%]	Rigidit- -y [%]	Axial [%]	Tremor [%]	Global Impr. [%]	Settings RH [%]	Settings LH [%]	Brady- kinesia	Rigi- dity [%]	Axi- al [%]	Trem- or [%]	Global Impr. [%]
Patient - 01	3: 44.1%; 7: 30.6%; 8: 25.3%, Amp: 3 mA [5 mA]	3: 100%, Amp: 2.1 mA [3 mA]	86.67	100	50	N/A	81.25	3: 50%; 4: 50%, Amp: 2.4 mA	3: 33.33%; 4: 33.33%; 5: 33.33%, Amp: 1.5 mA	53.33	87.5 0	37.5	N/A	62.50
Patient - 02	1: 7.5%; 4: 7.5%; 5: 10%; 7: 37.5%; 8: 37.5%, Amp: 3.5 mA [5 mA]	1: 100%, Amp: 3.3 mA	69.50	30	63.63	N/A	63.63	3: 50%; 4: 50%, Amp: 3.1 mA	3: 50%; 4: 50%; Amp: 3.2 mA	43.50	70	45.5	N/A	56.81
Patient - 03	1: 31.2%; 4: 6.63%; 7: 44.3%; 8: 17.9%, Amp: 3.5 mA [5 mA]	1: 60%; 4: 8.1%; 8: 32.0%, Amp: 3.0 mA [5 mA]	78.94	42.85	44.44	N/A	60.60	2: 60%; 5: 40%, Amp: 3.0 mA	2 to 7:15%, Amp: 2.5 mA	78.94	14.2 8	44.4	N/A	54.54
Patient - 04	4: 58%; 8: 42%, Amp: 5 mA	1: 22%; 8: 78%; Amp: 3.06 mA	50	6.67	71.43	100	55.29	2 - 4: 50%, 5 - 7: 50%,	1: 20%; 2,3,4: 80%, Amp: 3.0	59.40	20	60.7 1	100	57.64

Patient - 05	2: 41.7%, 3: 4.7%, 4: 6.05%, 5: 47.5%, Amp: 3.1 mA	1: 38.1%, 4: 57%, 5: 3.58%, Amp: 3.4 mA	100	100	50	100	90	Amp: 3.0 mA	mA	85.7	100	60	100	85
								2: 100%, Amp: 1.9 mA	3: 50%, 4:50%, Amp: 2.1 mA					

Figure S34. Feasibility study of *Cleartune*. The stimulation volume programmed by standard of care settings compared with the stimulation volume programmed by *Cleartune*. Fiber tractography in each patient is weighted by the strength of connection to the stimulation volume.

"Second, we showed that this symptom-network library is robust to cross-validations and outperforms a single tract model calculated on global UPDRS-III improvements. We validate results on multiple additional datasets of various nature from different centers. Third, based on the generated model, we introduce an algorithm capable of suggesting personalized and symptom-specific DBS stimulation parameters, which could similarly be validated in out of sample datasets and prospectively tested in five patients." – **discussion, pg.20**

Reviewer #2:

The aim of this study was to investigate deep brain stimulation on PD patients

The paper is well written and clear.

I have only some minor comments.

We would like to thank the reviewer for their positive evaluation of our manuscript.

The authors did not give any details about diffusion tensor imaging in the three centers.

Did the authors conduct sequence harmonization?

This may be a critical point and we extended the clarification about our use of normative data in the following paragraph that was added to the discussion of the manuscript:

First, our main model applies normative tractograms instead of patient-specific tractography data to isolate symptom-specific networks. The reasons to focus on normative datasets are manifold: It is hard, if not impossible, to reconstruct thin bundles such as the ansa lenticularis, the comb fibers or the striatopallidofugal bundle based on clinical imaging since these are thin structures that traverse through gray matter and/or orthogonal to the internal capsule⁴²⁻⁴⁴ (also see section S2). Practical reasons preclude us from generating large cohorts with individualized dMRI data given the cost and logistics involved. Typical reports of patient studies that have been based on individualized dMRI data range in the order of $N < 30$ ^{12,18,45,46}, while studies that use normative tractograms were often able to pool across larger numbers of patients^{28,47-50} (for a review see⁵¹). Finally, studies that carried out direct head-to-head comparisons found similar results when using patient-specific vs. normative data^{18,46}. – **discussion, pg.22**

In the same line, did the authors investigate center effect?

The reviewer raises an important question of how to account for DBS centers in statistical models. Indeed, we are uncertain of exactly how to deal with this: If surgeon A always implants more anteriorly than surgeon B, but outcomes from surgeon A are systematically better than the ones from surgeon B, then simply regressing out the difference in placement of electrodes across surgeons does not seem sensible. The fraction of noise we would indeed like to regress out would be the one possibly introduced by differences in imaging or clinical scoring. It is not straight-forward to disentangle the two.

We carried out the following analyses to maximize our understanding of the data and results further (changes to the text below):

Symptom Associated Multi-Tract Model (Validation Cohorts)

To test generalizability of our model, next, we recalculated the same multi-tract model on an independent set of 93 patients from the University of Würzburg and Beijing (Validation cohort I). The result anatomically matched the original model. Namely, connections between M1 and the STN as well as cerebellar tracts associated with tremor improvements. Axial symptom improvements correlated with streamlines adjacently anteriorly followed by the ones that associated with rigidity improvements (SMA and prefrontal regions). Using network blending, we were able to predict UP-DRS-III improvements in this validation cohort purely based on the original model calculated from the discovery cohort. These predictions significantly correlated with empirical improvements in the test dataset ($R = 0.37$, $p < 0.001$, figure 5 C).” – **results**, **pg. 14**

Figure 5. Retrospective validation on long term clinical outcome data. A) The fiber distribution of the original model as shown in previous figures, B) fiber distribution when recalculating the same model on the independent test dataset (N = 93). C) Prediction of UPDRS-III improvements in the test set based on the original symptom specific model.

“To control for subcohorts within the discovery cohort, we reran the original model and applied a mixed-effects model that controlled for dataset as a random effect. Results were similar and re-mained significant (R = 0.30, p < 0.001).” – **results, pg.12**

The algorithm “*Cleartune*” is an interesting perspective and could have a clinical value. However, *Cleartune* appears as a prognostic black box. It is not clear which data were used to set up the algorithm, and what kind of methods is used to predict. Did the authors use the same retrospective data for learning? How is the prediction calculated? The readers need more details about this tool and a validation prospective study

We apologize that we did not clarify the nature of *Cleartune* enough. Indeed, part of the confusion might have arisen from the fact that, in the original manuscript, *Cleartune* simply tested all monopolar solutions in a brute-force manner and made predictions for each one exactly in the same way as the original multi-tract model did (the methods of how this is done are outlined precisely). *Cleartune* then chose the contact with the best predictions.

In the revised manuscript, however, we have extended *Cleartune* in such a way that it is also able to suggest multipolar settings. This leads to an explosion of the parameter space, and hence, brute-force testing the entire parameter space is not feasible anymore. We now added a surrogate optimization algorithm and dedicated an entire supplementary section to outline what *Cleartune* does:

Supplementary methods, S6

Cleartune is a computational tool to optimize stimulation protocols trained on the Multitract model. The optimization problem for our study is formulated as

$$J = \operatorname{argmax} \sum_{s=1}^S \omega_s \hat{1}_s(J) \text{ such that } \begin{cases} \|J\|_1 \leq 5 \text{ mA} \\ -4 \text{ mA} \leq J_c \leq 0 \text{ for } c = 1 \text{ to } c = N \text{ contacts} \end{cases}$$

where S defines specific symptoms, ω_s is the symptom weight and \hat{i}_s is the predicted symptom improvement for stimulation J is composed of currents across each contact J_c . For conventional DBS electrodes with 4-8 contacts, the parameters space to be investigated is relatively large for the FEM based volume conductor model (≈ 1 min per sample). Therefore, we employ a surrogate optimizer (see `surrogateopt` in MATLAB) based on interpolation of radial basis functions through sparsely and randomly sampled parameter space. In brief, the algorithm performs the following steps.

ClearTune Algorithm

- | |
|--|
|  1. Set optimizer parameters, e.g. the maximum number of samples, the minimum number of random samples to create a surrogate model, the objective limit, etc. 2. Define current bounds (see Eq. 1) 3. Initiate \mathbf{J}_{init} 4. Solve the FEM problem for \mathbf{J}_{init}. If necessary, solve the additional random samples within the current bounds to create a surrogate model. 5. Investigate the parameter space using the surrogate model 6. Randomly sample around the incumbent (the best yet observed) a merit function that balances exploration of the parameter space and minimization of the surrogate. The sample with the smallest value is the adaptive sample. 7. Solve the FEM problem for the adaptive sample and refine the surrogate. 8. Update incumbent if the new global minimum (maximum) observed. 9. Update the sampling dispersion depending on the rate of success of adaptive samples against incumbents. 10. If converged, but above the objective limit and below the maximum number of samples, reset by discarding all adaptive points. |
|--|

Figure S36 Example of an optimization run in Cleartune. Only samples computed with the FEM model are shown. The algorithm is initialized to calculate the stimulation volume around the 3rd contact of the electrode. For the first 100 random samples the surrogate model is constructed. Once the model is constructed, adaptive sampling is performed (black dots) where the model searches for a local minimum phase. At the 200th iteration, the surrogate model is reset which implies that a new surrogate model is constructed. The purple points represent the best values since the previous surrogate reset. This cycle of adaptive sampling and random sampling is continued until the number of iterations equal the maximum functional evaluation set externally by the user or if the objective function reaches the desired value. In this case, the objective function evaluation has not reached, instead it completed the 500 iterations it was set to perform.

For the current application, we initialized the optimization protocol to calculate the stimulation volume at the 3rd electrode contact. However, this is a user defined parameter and therefore, the user has the ability to simulate a monopolar review. Next, 120 random samples are taken to define the surrogate model for the further adaptive sampling (Fig. S36). The maximum number of FEM samples is limited to 500, based on observation that most FEM calculations converge at around ~450 iteration mark. The objective limit is set to 0.9, which corresponds to 90% improvement. If the local convergence did not fulfil the objective limit criterion, the model is reset.

Reviewer #3:

The ability to predict exact locations of optimal STN deep brain stimulation to improve the unique symptoms of individual patients is an important challenge in the field. This manuscript seeks to personalize deep-brain stimulation programming by examining the association of specific symptom improvement (tremor, bradykinesia, axial improvement) with stimulation of specific network targets that connect to various brain regions. The goal is to create a three-dimensional symptom-circuit model in standard stereotactic space and to predict optimal therapy by stimulating multiple contacts on a single electrode. The study builds on previously published work. At the same time, the claimed benefits lack validation and therefore appear somewhat overstated. Inclusion of a single patient with improved results when tested for a 24-hour period with algorithmic recommendations is unconvincing.

We would like to thank the reviewer for their thoughtful comments that helped us in drastically revising the manuscript. What is key to the point raised here is that we now used the model to estimate improvements in two additional unseen cohorts (N = 103) and added additional prospective cases to further add credibility to the model (see below).

Active contacts are visualized after atlas-based co-registration with Lead-DBS software. Figure 1 shows the striking heterogeneity of contact locations in the STN region with some contacts located far from traditional sites of stimulation. This distribution seems highly unlikely, particularly given that 2 mm shifts in stimulation result in dramatic changes in therapeutic efficacy. It suggests that the apparent variability of active contact location may reflect differences introduced through the atlas-fitting process. Validation is needed. Where postoperative MRI data is available, the matching of atlas-based localization to the MR visualized STN (or co-registration of post-operative CT and pre-operative MRI) needs to be demonstrated.

We agree that it is helpful to validate these cases. We added a supplementary figure showing select cases that are outside of the normal range. Our group has localized around 3,000 patients from centers world-wide (including notable centers such as Harvard, UCL, UF, Stanford, Penn, etc.) and, indeed, the range of variance shown here is not above the norm.

Another effect we sometimes perceive ourselves is that 3D models psychologically amplify the magnitude of misplacements. Changing a planning trajectory in a surgical planning software by ~1-2 mm may not look as drastic as moving the respective 3D electrode by the same amount in these computerized models. Maybe, this can also be seen in the examples we provide in the added figure (below).

The following paragraph was added:

“Figure S1 shows native space imaging of example patients in synopsis with reconstructed electrodes.” – **results, pg. 6**

Figure S1: Comparison of imaging data with 3D DBS models of example cases with suboptimal electrode placements. Each row shows in order from left to right: A preoperative T2 (axial slice), the registered and brain shift corrected postoperative imaging (CT or MRI in

the last case). White lines connect corresponding dots between pre- and postoperative images to demonstrate co-registration. Next is a fused image of pre- and postoperative scans in which the postoperative image is starkly thresholded and shown in false colors to mainly show the electrode lead. An enlarged cutout of the relevant region is provided. The last two panels show the 3D electrode with active contacts marked in red and the visualization of active contacts as spheres (as presented in figure 1). In the first case, the left tip is seen on the slice and this electrode is placed too dorsally as correctly reconstructed in the 3D visualizations. In the second case, both electrodes are placed too anteriorly and similar to the first case, the left electrode is placed too dorsal. The tip of the electrode can be seen on the axial slice. Reconstructions correctly show this misplacement. In the third patient, the right electrode is placed accurately, but the left electrode is placed too medially and posteriorly. Three active contacts have been chosen by the clinical team. In the fourth case, the left electrode is placed too dorsally and cannot be seen in the selected axial slice. A contact that resides in the thalamus was chosen. The 3D reconstruction correctly captures both electrode placements. In the fifth example, a postoperative MRI shows the placement of electrodes in the zona incerta, which is correctly captured by the electrode reconstructions.

It is not clear how the symptom-network library was generated. An existing “DBS tractography atlas”, appears to be manually adjusted with a “more exhaustive” set of connections. The placement of expected or anticipated tracts is confusing since the claimed streamline resolution far exceeds that of standard DTI imaging. In the discussion, the authors point this out, but do not address how the issue was resolved, allowing tight differentiation despite blurring associated with atlas-based fitting. Please clarify how the locations of these highly defined tracts were determined and validated.

We are sorry that this wasn't clear enough, the method to create the atlas was based on the work of Erik Middlebrooks (Mayo Clinic) which used a diffusion dataset aggregated from >1,000 human connectome project subjects. We paste the relevant sections below and have refined them to add clarity as follows:

"Anatomical Tract Atlas

To carry out DBS fiber filtering based on electric fields estimated and symptom improvements across the cohort of patients, we first established a streamline atlas using various sources of information. This work is based on two published streamline atlases 17,44 that were extended

to include a more exhaustive set of tracts in and around the subthalamic region. The process involved diffusion MRI based tractography on a group average template, using manually defined regions of interest, inclusion of published resources, comparisons of results with the anatomical literature, cadaveric dissection studies, histology and ex-vivo imaging. Supplementary section S2 details methods and results that led to the resulting ‘DBS tractography atlas version 2’". – **methods, pg.24**

Also, we now include the extensive validation work that went into the creation of this atlas to the supplementary material (section S2). Since extensive, we refrain from pasting all figures and material here.

From this set of tracts (shown in its entirety in section S2 and listed in table S1), for each symptom, we filtered out the streamlines that correlated with changes in the symptom. This process is called DBS fiber filtering and was introduced earlier by our group (Li et al. 2020 Nature Communications). Namely, for each pair of electric field and streamline, we calculated the maximal magnitude of the electric field that the streamline passes through. We then rank-correlated these peak values with clinical symptom changes. Rank correlations were used since the values follow a skewed distribution and the exact relationships between field strength and tract activations are unclear. The resulting Spearman’s correlation coefficient is used to weight/tag each streamline. Finally, these weighted streamlines form our model. We visualize streamlines that correlated significantly after FDR-correction for multiple comparisons. Taken together, this result embodies the “symptom-associated multi-tract model” or “symptom-network library”. The following section describes this method with additional details in the manuscript:

Multi-Tract implementation of DBS fiber filtering

We build upon the DBS fiber filtering concept introduced in ¹⁸ and extended in ⁶⁴ to isolate tracts associated with changes across multiple motor symptom domains (figure 8). In the first step, we used this method to build a symptom-associated multi-tract model (or “symptom network library”) which associates streamlines with improvements of clinical subscores (tremor, bradykinesia, rigidity and axial symptoms). Activation of these four tract sets correlated with improvements in respective symptoms. To be included into the library, each tract had to pass through low number of E-fields (> 0.5 % of total number of E-fields) at a rather high peak intensity of >1.5 V/mm. This constraint was set up since we wanted to exclude

tracts that were not strongly modulated by any stimulation field at all (which in theory could still obtain high correlation values if sub-threshold intensities correlated with clinical improvements). Changing the arbitrarily chosen values (> 0.5 % E-fields and >1.5 V/mm) e.g., to >2 and > 4 V/mm did not qualitatively alter results. For each tract, Spearman's rank correlations were then calculated for each symptom group separately, by correlating the respective sub-score with the peak amplitude of each patient's E-field a given streamline passed through. This mass-univariate approach leads to a high number of rank correlation coefficients, which were thresholded at a p-value < 0.05 after correction for multiple comparisons using the false-discovery rate (FDR). - **methods, pg.26**

Further, it further appears that the data were used to define the tracts and then the proposed tracts were used to interpret the same data, creating a potential circularity of logic (despite leave-one-out analysis).

While the cross-validation design we originally applied in the $N = 129$ discovery cohort ensured avoidance of circularity, in addition, we now use the model to successfully estimate clinical outcomes in two out-of-sample datasets that were not seen by the model ($N = 103$). The following sections were added detailing results:

"Symptom Associated Multi-Tract Model (Validation Cohorts)

To test generalizability of our model, next, we recalculated the same multi-tract model on an independent set of 93 patients from the University of Würzburg and Beijing (Validation cohort I). The result anatomically matched the original model. Namely, connections between M1 and the STN as well as cerebellar tracts associated with tremor improvements. Axial symptom improvements correlated with streamlines adjacently anteriorly followed by the ones that associated with rigidity im-provements (SMA and prefrontal regions). Using network blending, we were able to predict UP-DRS-III improvements in this validation cohort purely based on the original model calculated from the discovery cohort. These predictions significantly correlated with empirical improvements in the test dataset ($R = 0.37$, $p < 0.001$, figure 5 C)." - **results, pg.14**

Figure 5. Retrospective validation on long term clinical outcome data. A) The fiber distribution of the original model as shown in previous figures, B) fiber distribution when recalculating the multi-tract model on the independent test dataset (N = 93). C) Prediction of UPDRS-III improvements in the test set based on the original symptom specific model.

***Clartune* – an algorithm to suggest stimulation parameters**

In the next step, we created an algorithm capable of suggesting optimal stimulation settings by maximizing stimulation of a specific set of symptom tracts in novel patients. Termed *Clartune*, this algorithm tests stimulation fields based on the entire parameter space of stimulation parameters and suggests the one that receives the highest predicted improvements. Video S1 visualizes the process of the algorithm testing parameters to maximize outcomes in the four symptom domains for a specific directional electrode. To test utility of the algorithm, it was first applied to all patients within the retrospective cohort. This led to an alternate set of stimulation volumes which could be compared to the ones applied in clinical practice using spatial correlations. Here, higher spatial correlations meant greater similarity between the clinically applied E-fields and the ones suggested by the algorithm. Higher similarities correlated with better UPDRS-III improvements ($R = 0.22$, $p = 0.001$). The same was true when repeating the analysis on the validation cohort I which the model had not seen ($R = 0.23$, $p = 0.03$). Intuitively, this finding may be understood as follows: In cases in which parameters suggested by *Clartune* agreed with the clinical ones, improvement was higher than in the ones for which the two settings disagreed.

In a second step, we aimed at testing symptom-specificity of suggestions derived by *Clartune*. To do so, we leveraged a unique dataset of 10 patients (20 electrodes; Validation cohort II), for which multiple settings had been tested in a prospective double-blinded clinical

trial (N = 186) 30. These patients had been implanted with directional electrodes (Boston Scientific Vercise Cartesia) and for the directional levels with best clinical response, each segment had been tested in increasing 1 mA steps until a side effect occurred or until reaching 5 mA. In addition, the omnidirectional setting (switching on all three segments) was tested in the same way. As above, we calculated predictions for each setting using the original model (from the N = 129 discovery cohort). In 17 of the 20 electrodes, predictions positively correlated with clinical improvements (all correlation plots with over six data points are shown in figure S32). Naturally, a one-sample t-test across these R-values was significant ($T = 4.155$, $p < 0.001$; figure. 6).

For each stimulation setting, bradykinesia and rigidity improvements were available separately. Only three of the ten cases had substantial tremor at baseline, so tremor could unfortunately not be analyzed. To test for symptom-specificity, we repeated the analysis two more times, each time maximally weighting either bradykinesia or rigidity in the multi-tract model. The model weighted for the correct symptom led to significantly higher correlations between predictions and empirical improvements across settings in each electrode for the correct vs. respective other symptom ($p < 0.05$ for both analyses; figure 6B and C).". –

results pg 14-16

Fig 6. Retrospective validation on TWEED dataset. A) Left panel illustrates a raincloud plot where each data point represents the Spearman's correlation coefficient between predicted and empirical UPDRS-III improvements for settings in one of the 20 electrodes. All correlation plots are shown in figure S32. The right panel gives four representative examples. Here, a red eclipse is used to represent the stimulation contact that renders the highest improvement in a given patient, while the contact chosen by the model is marked with a blue eclipse, corresponding stimulation fields are shown for the example electrodes.

B and C) To assess symptom-specificity of the model, the analysis was repeated, this time maximally weighting either bradykinesia or rigidity symptoms, respectively. Correlations across settings in the 20 electrodes were almost all positive when the model was used to predict improvements in the correct symptom, but significantly dropped when used to predict improvements in the respective other symptom. In each panel, two representative examples of each correct vs. incorrect symptom pairings are given.

"We replicated the same model based on a multi-center validation cohort (N = 93). Third, based on the generated model, we introduced an algorithm capable of suggesting personalized and symptom-specific DBS stimulation parameters, which could similarly be validated in out of sample datasets and prospectively tested in five patients. Using monopolar review data acquired in patients with segmented electrodes, we were able to demonstrate symptom-specificity of the algorithm. Namely, a model tuned to predict bradykinesia outcome performed better to predict bradykinesia compared to rigidity outcomes, and vice versa". – **discussion, pg.20**

While using populations to refine tracts seems reasonable. At the same time, each of these should produce an average (or median) and a distribution. No such spread is shown.

Indeed, we do not track in individuals and aggregate results but instead create tracts manually using a population-averaged high-resolution dataset. Unfortunately, the method does not allow us to infer average / median and distribution of results. This same method has been used to create normative atlases by experts in the field, e.g., in the following papers:

1. Middlebrooks EH, Domingo RA, Vivas-Buitrago T, Okromelidze L, Tsuboi T, Wong JK, Eisinger RS, Almeida L, Burns MR, Horn A, Uitti RJ, Wharen RE, Holanda VM, Grewal SS. Neuroimaging Advances in Deep Brain Stimulation: Review of Indications, Anatomy, and Brain Connectomics. *AJNR Am J Neuroradiol*. Published online August 13, 2020:1-11. doi:[10.3174/ajnr.A6693](https://doi.org/10.3174/ajnr.A6693)
2. Meola A, Comert A, Yeh FC, Sivakanthan S, Fernandez-Miranda JC. The nondecussating pathway of the dentatorubrothalamic tract in humans: human connectome-based

tractographic study and microdissection validation. *J Neurosurg*. Published online October 9, 2015:1-7. doi:[10.3171/2015.4.JNS142741](https://doi.org/10.3171/2015.4.JNS142741)

3. Yeh FC, Panesar S, Fernandes D, Meola A, Yoshino M, Fernandez-Miranda JC, Vettel JM, Verstynen T. Population-averaged atlas of the macroscale human structural connectome and its network topology. *NeuroImage*. 2018;178:57-68. doi:[10.1016/j.neuroimage.2018.05.027](https://doi.org/10.1016/j.neuroimage.2018.05.027)

4. Meola A, Yeh FC, Fellows-Mayle W, Weed J, Fernandez-Miranda JC. Human Connectome-Based Tractographic Atlas of the Brainstem Connections and Surgical Approaches. *Neurosurgery*. Published online February 2016:1-18. doi:[10.1227/NEU.0000000000001224](https://doi.org/10.1227/NEU.0000000000001224)

5. Yeh FC. Population-based tract-to-region connectome of the human brain and its hierarchical topology. *Nat Commun*. 2022;13(1):4933. doi:[10.1038/s41467-022-32595-4](https://doi.org/10.1038/s41467-022-32595-4)

We have added the following limitations section to discuss our rationale to use of normative tractograms:

"Several limitations apply to this study. First, our main model applies normative tractograms instead of patient-specific tractography data to isolate symptom-specific networks. The reasons to focus on normative datasets are manifold: It is hard, if not impossible, to reconstruct thin bundles such as the ansa lenticularis, the comb fibers or the striatopallidofugal bundle based on clinical imaging since these are thin structures that traverse through gray matter and/or orthogonal to the internal capsule⁴²⁻⁴⁴ (also see section S2). Practical reasons preclude us from generating large cohorts with individualized dMRI data given the cost and logistics involved. Typical reports of patient studies that have been based on individualized dMRI data range in the order of $N < 30$ ^{12,18,45,46}, while studies that use normative tractograms were often able to pool across larger numbers of patients^{28,47-50} (for a review see⁵¹). Finally, studies that carried out direct head-to-head comparisons found similar results when using patient-specific vs. normative data^{18,46}." – **Limitations, pg.23**

Spatial variability introduced by fitting to the atlas is not well quantified and its impact on the library generation is not explored.

We agree that the influence of the spatial variability on the symptom library is an important factor to test empirically. The following sections were added to do so:

Symptom-Associated Multi-Tract Model (Discovery Cohort)

Second, we tested how robust our results were regarding spatial inaccuracies of each stimulation site. To test this, we iteratively recalculated the symptom specific tract model 1,000 times, each time after spatially jittering each electrical field based on a 3D Gaussian distribution with 2 mm full width half maximum. The resulting models were highly similar to one another (and to theunjittered version) with an average mean spatial correlation of $R > 0.8$. Details and example visualizations of jittered models are shown in figure S28.— **results, pg.12**

Figure S28: To test robustness of model results as a function of spatial uncertainty in stimulation sites, we recalculated the model 1,000 times, each time after spatially jittering electric fields based on a 3D Gaussian distribution with 2 mm full width half maximum. Resulting models were highly similar (correlations of fiber weightings across fibers are shown in the figure, four example models are shown).

Rather strong conclusions are drawn from relatively weak localization correlations, and the raw data for these correlations are not shown. For example, correlation coefficients of 0.23 and 0.18 would correspond to explaining only a tiny fraction of the variance in outcomes. This manuscript, and cited work, would be greatly strengthened with a model that explains clinical outcomes to a significant degree and then determines the portion attributable to active contact location.

We agree that amount of explained variance is small. We have validated the results in additional datasets as mentioned above. Furthermore, we created a model with additional variables that we could think of which explained more variance (see below). However, as also discussed with reviewer one, we must emphasize that these additional variables are the ones we cannot change in clinical practice (while we can optimize electrode placements and stimulation settings). For instance, while the baseline UPDRS score explains a large amount of variance in outcomes, as medical care providers, we do not have any influence on this variable in our clinical practice. Instead, electrode placement and which networks to stimulate embody the key variables we can optimize.

The following changes were added:

Given the moderate strength of the correlation coefficients between the predicted improvement and empirical clinical improvements, we investigated whether a linear model considering other demo-graphic factors could explain additional variance. To do so, we fit a linear model that additionally included UPDRS-III baseline, patient age at surgery, sex, and levodopa equivalent dose (LEDD) reduction as covariates. This model explained 25.5% of the variance in clinical improvements ($R^2 = 0.26$, $p < 10^{-6}$). The predicted improvements of the multi-tract model remained a significant predictor ($t = 3.2$, $p < 0.0017$). UPDRS-III baseline scores ($t = 3.3$, $p = 0.001$) and sex also explained significant amounts of variance ($t = 3.0$, $p = 0.03$), while the other variables did not (LEDD reduction: $p = 0.43$, age: $p = 0.39$). Of note, none of these variables may be changed due to medical practice, with the sole exception of the electrode placement and stimulation settings, which renders the multi-tract model predictions (which are based on these factors) the critical anchor point with an opportunity to potentially improve patient care.– **results, pg.15**

Moreover, we now include the following section on modeling considerations, that argues that, while our results may not predict large amounts of variance, they may still be key and potentially important for clinical practice:

Section S5. Modelling Considerations

A natural question that may arise in the context of the present results is the validity of a model given the relatively low correlation coefficients (R), and consequentially, R² values. To discuss this question, we would like to raise the following points (Fig S35).

1. First, we need to ask how much variance DBS modeling should at best explain in the first place. Clinical improvements are not just based on electrode placement and stimulation volumes, but governed by many factors, such as disease subtype, age, sex, levodopa response, duration of disease, comorbid other conditions, etc. We estimate that these factors alone will explain ~50% of DBS response³⁴.
2. Noise in clinical scores (inter- and intra-rater test-retest reliability of the UPDRS-III) will account for another 15%³⁵.
3. Imaging resolution and electrode placement & modeling inaccuracy may explain another 15-20% of variance³⁶⁻³⁸.
4. The use of multi-site datasets such as in the present study may add residuals that may explain another 5-10% of the variance.

Based on this assessment, we believe that ~10% explained variance (R values of ~0.3) may realistically be expected in a multi-site dataset such as the present one. However, we believe that such models are still useful because electrode placement / stimulation settings are the only factor that can be influenced, whereas other components such as patient age, disease type and disease onset are immutable. Therefore, identification of a robust, and highly optimal target that can only explain ~10% of variance would still remain an important finding in our field.

Figure S35. Modeling considerations. The pie chart is illustrative of the various factors that influence the outcome of the patient undergoing DBS. With this, we could conclude that the maximum impact electrode positioning can have on the patient outcome is ~10%.

The *Cleartune* algorithm seeks to optimize contact position by maximizing stimulation over symptom-specific tracts. If successful, such an algorithm would be helpful to adapt programming as the disease progresses. The authors argue that where programming agreed with *Cleartune*, outcomes were better than those where programming differed. However, there are two striking issues that arise:

- (1) Based on the model, the authors claim that an average of 21% improvement would be expected by using *Cleartune*. The amalgamation of matching and adjacent contact into a single group is questionable. If the algorithm is in fact correct, then one would expect matching > adjacent > different. These categories should be separated and should show a clear trend if this approach is valid.

(2) In addition, there should be some explanation why, even with monopolar stimulation, that the expert neurologists in these centers were unable to identify the optimal sites of stimulation and fell short by an astounding 21% on average. Again, this seems very unlikely, and likely overestimates the potential benefit of the algorithmic approach. A notable shortcoming of the study is that undesired side effects of stimulation are ignored. Hence, the failure to stimulate the modeled optimal site may simply be that stimulation at the site was accompanied by undesirable side effects.

We apologize for this confusion. In fact, we had discussed this analysis with multiple colleagues and tried our best to word this as carefully as possible (e.g. “We must reiterate, however, that this analysis was entirely carried out in silico with the sole aim to quantify a potential room for improvement.”). It seems that we failed to communicate this analysis well and it was still interpreted as a claim of superiority of *Cleartune*. We have now removed this analysis to avoid similar confusion for readers.

In the revised version of the manuscript, we extended the capability of *Cleartune* to stimulate combinations of contacts by using a more efficient way to explore the parameter space (namely a surrogate-optimization algorithm). As such, the distinction between same vs. adjacent contacts is not straight-forward anymore. Rather, we correlate clinical vs. model-based electric fields and show that the more similar they are, the better clinical improvements were:

***Cleartune* – an algorithm to suggest stimulation parameters**

In the next step, we created an algorithm capable of suggesting optimal stimulation settings by maximizing stimulation of a specific set of symptom tracts in novel patients. Termed *Cleartune*, this algorithm tests stimulation fields based on the entire parameter space of stimulation parameters and suggests the one that receives the highest predicted improvements. Video S1 visualizes the process of the algorithm testing parameters to maximize outcomes in the four symptom domains for a specific directional electrode. To test utility of the algorithm, it was first applied to all patients within the retrospective cohort. This led to an alternate set of stimulation volumes which could be compared to the ones applied in clinical practice using spatial correlations. Here, higher spatial correlations meant greater similarity between the clinically applied E-fields and the ones suggested by the algorithm. Higher similarities correlated with better UPDRS-III improvements ($R = 0.22$, $p = 0.001$). The same was true

when repeating the analysis on the validation cohort I which the model had not seen ($R = 0.23$, $p = 0.03$). Intuitively, this finding may be understood as follows: In cases in which parameters suggested by *Cleartune* agreed with the clinical ones, improvement was higher than in the ones for which the two settings disagreed.

In a second step, we aimed at testing symptom-specificity of suggestions derived by *Cleartune*. To do so, we leveraged a unique dataset of 10 patients (20 electrodes; Validation cohort II), for which multiple settings had been tested in a prospective double-blinded clinical trial ($N = 186$)³⁰. These patients had been implanted with directional electrodes (Boston Scientific Vercise Cartesia) and for the directional levels with best clinical response, each segment had been tested in increasing 1 mA steps until a side effect occurred or until reaching 5 mA. In addition, the omnidirectional setting (switching on all three segments) was tested in the same way. As above, we calculated predictions for each setting using the original model (from the $N = 129$ discovery cohort). In 17 of the 20 electrodes, predictions positively correlated with clinical improvements (all correlation plots with over six data points are shown in figure S32).

Naturally, a one-sample t-test across these R-values was significant ($T = 4.155$, $p < 0.001$; figure. 6).

For each stimulation setting, bradykinesia and rigidity improvements were available separately. Only three of the ten cases had substantial tremor at baseline, so tremor could unfortunately not be analyzed. To test for symptom-specificity, we repeated the analysis two more times, each time maximally weighting either bradykinesia or rigidity in the multi-tract model. The model weighted for the correct symptom led to significantly higher correlations between predictions and empirical improvements across settings in each electrode for the correct vs. respective other symptom ($p < 0.05$ for both analyses; figure 6B and C)". – **results, pg.15-16**

"Third, based on the generated model, we introduce an algorithm capable of suggesting personalized and symptom-specific DBS stimulation parameters, which could similarly be validated in out of sample datasets and prospectively tested in five patients. – **discussion, pg.22**

Fig 6. Retrospective validation on TWEED dataset. A) Left panel illustrates a raincloud plot where each data point represents the Spearman's correlation coefficient between predicted and empirical UPDRS-III improvements for settings in one of the 20 electrodes. All correlation plots are shown in figure S32. The right panel gives four representative examples. Here, a red eclipse is used to represent the stimulation contact that renders the highest improvement in a given patient, while the contact chosen by the model is marked with a blue eclipse, corresponding stimulation fields are shown for the example electrodes.

B and C) To assess symptom-specificity of the model, the analysis was repeated, this time maximally weighting either bradykinesia or rigidity symptoms, respectively. Correlations across settings in the 20 electrodes were almost all positive when the model was used to predict improvements in the correct symptom, but significantly dropped when used to predict improvements in the respective other symptom. In each panel, two representative examples of each correct vs. incorrect symptom pairings are given.

The prospective application of *Cleartune* in a single patient is anecdotal and contributes little. At a minimum, these three centers could examine 10-20 patients prospectively to help validate the methodology. For each patient, individualized imaging to show the location of the active contact with respect to the MR visualized STN also should be presented to confirm the accuracy of the atlas fitting.

We agree that the single case should not be overinterpreted but may demonstrate our first efforts to show a feasibility of applying *Cleartune* in clinical practice.

Of note, prospective testing of DBS algorithms in larger cohorts constitutes a prospective trial that usually requires a different structure of the investigational team and underlying funding.

We were still able to add an additional set of clinical cases (N = 5) that were reprogrammed using *Cleartune* (the original patient had to be discarded since we updated the algorithm to allow more complex stimulation settings). We still mainly moved these results to supplementary material to give them less weight. We further changed the language to state that this was done to show feasibility, not to prospectively validate the algorithm. As mentioned above, we additionally tested utility of *Cleartune* on an unseen retrospective dataset that applied multiple stimulation settings per patient using omnidirectional and directional settings.

The following sections were added/changed:

Prospective application of *Cleartune*

Finally, we prospectively applied DBS stimulation parameters suggested by *Cleartune* in a small set of five patients (study design shown in fig. S33). UPDRS-III scores were taken by raters that were blinded to which protocol was active and then compared between standard of care stimulation settings and the ones suggested by *Cleartune*. Figure S34 shows electrode localizations and the two stimulation protocols (*Cleartune* vs. Standard of Care; SoC) together with their tract overlaps from the multi-tract model. Detailed results are given in the supplementary material (supplementary section S4). In brief, from a baseline of 49.8 ± 22.1 UPDRS-III points, under *Cleartune* settings, scores improved by 34.4 ± 13.1 points ($73 \pm 11.8\%$). Under standard of care settings, scores improved by 31.8 ± 15.1 points ($65.4 \pm 12.1\%$). In four of the five patients, *Cleartune* settings led to a higher improvement than SoC settings. In the fifth patient, improvements were comparable (36 vs. 38 points improvement). While three of the five patients preferred *Cleartune* over SoC settings, in two patients, *Cleartune* settings led to side-effects (dyskinesia in patient 05 and dizziness in patient 04). This emphasizes that the current model was purely driven by improvements (and not by side-effects), which is a clear limitation for clinical applicability. Tracts of avoidance that code for side-effects should be added to the model in future attempts. Alternatively (and additionally), clinicians may reduce the stimulation amplitude suggested by *Cleartune* in case of side-effects (while keeping the remaining parameter choices unchanged). This would still reduce the parameter space and could hence help clinicians to come to a beneficial solution faster. While generally promising, given the low N, these results should not be overinterpreted. Rather, this trial was carried out to test feasibility of applying *Cleartune* in a clinical setting and to gather first experience in preparation for a proper prospective trial. As such, the trial was not powered to compare *Cleartune* vs. SoC settings (non-inferiority or superiority).— **results, pg.18.**

Section S4. Feasibility trial for prospective application of *Cleartune* in a clinical setting.

To test feasibility of applying *Cleartune* in a clinical setting, a feasibility trial was carried out in a small sample of $n=5$ prospective patients. This trial was designed to include a randomization step, where the patient was blinded to the administration of *cleartune* vs. clinical settings.

Clinical data, which included the pre-operative T1w, T2w, and post operative CT images was used to localize DBS electrodes in each patient. Baseline scores were taken in the stimulation and medication off states. The *Cleartune* algorithm was executed for each electrode separately, for 500 iterations each. This led to *Cleartune* settings, which were stored in the pulse generator as an additional program to the existing standard of care (SoC) setting. In the second week, *Cleartune* settings or clinical settings were applied in randomized order, each for 24 hours. Resulting UPDRS-III scores were taken after 24 hours and the respective other program was switched on to be evaluated after another 24 hours. Figure S33 summarizes the trial design. Results are documented in table S3. In multiple cases, *Cleartune* suggested higher amplitudes than tolerable, and were hence reduced by the clinical team (without altering contact choices). Table S3 reports both suggested and programmed amplitudes. From a baseline of 49.8 ± 22.1 UPDRS-III points, under *Cleartune* settings, scores improved by 34.4 ± 13.1 points ($73 \pm 11.8\%$). Under standard of care settings, scores improved by 31.8 ± 15.1 points ($65.4 \pm 12.1\%$). In four of the five patients, *Cleartune* settings led to a higher improvement than SoC settings. In the fourth patient, improvements were comparable (36 vs. 38 points improvement). While three of the five patients preferred *Cleartune* over SoC settings, in two patients, *Cleartune* settings led to side-effects (dyskinesia in patient 05 and dizziness in patient 04). While generally promising, given the low N, these results should not be overinterpreted. Rather, this trial was carried out to test feasibility of applying *Cleartune* in a clinical setting and to gather first experience in preparation for a proper prospective trial. As such, the trial was not powered to compare *Cleartune* vs. SoC settings (non-inferiority or superiority).

Table S3. Results of feasibility study in n=5 patients.

Patient	Cleartune							Standard of care						
	Settings RH [%]	Settings LH [%]	Bradykin- -esia [%]	Rigidit- -y [%]	Axial [%]	Tremor [%]	Global Impr. [%]	Settings RH [%]	Settings LH [%]	Brady- kinesia	Rigi- dity [%]	Axi- al [%]	Trem- or [%]	Global Impr. [%]
Patient - 01	3: 44.1%; 7: 30.6%; 8: 25.3%, Amp: 3 mA [5 mA]	3: 100%, Amp: 2.1 mA [3 mA]	86.67	100	50	N/A	81.25	3: 50%; 4: 50%, Amp: 2.4 mA	3: 33.33%; 4: 33.33%; 5: 33.33%, Amp: 1.5 mA	53.33	87.5 0	37.5	N/A	62.50
Patient - 02	1: 7.5%; 4: 7.5%; 5: 10%; 7: 37.5%; 8: 37.5%, Amp: 3.5 mA [5 mA]	1: 100%, Amp: 3.3 mA	69.50	30	63.63	N/A	63.63	3: 50%; 4: 50%, Amp: 3.1 mA	3: 50%; 4: 50%; Amp: 3.2 mA	43.50	70	45.5	N/A	56.81

Patient - 03	1: 31.2%; 4: 6.63%; 7: 44.3%; 8: 17.9%, Amp: 3.5 mA [5 mA]	1: 60%; 4: 8.1%; 8: 32.0%, Amp: 3.0 mA [5 mA]	78.94	42.85	44.44	N/A	60.60	2: 60%; 5: 40%, Amp: 3.0 mA	2 to 7:15%, Amp: 2.5 mA	78.94	14.28	44.4	N/A	54.54
Patient - 04	4: 58%; 8: 42%, Amp: 5 mA	1: 22%; 8: 78%; Amp: 3.06 mA	50	6.67	71.43	100	55.29	2 - 4: 50%, 5 - 7: 50%, Amp: 3.0 mA	1: 20%; 2,3,4: 80%, Amp: 3.0 mA	59.40	20	60.71	100	57.64
Patient - 05	2: 41.7%, 3: 4.7%, 4: 6.05%, 5: 47.5%, Amp: 3.1 mA	1: 38.1%, 4: 57%, 5: 3.58%, Amp: 3.4 mA	100	100	50	100	90	2: 100%, Amp: 1.9 mA	3: 50%, 4:50%, Amp: 2.1 mA	85.7	100	60	100	85

CLINICAL SETTING

CLEARTUNE SETTING

■ TREMOR
 ■ BRADYKINESIA
 ■ RIGIDITY
 ■ AXIAL SYMPTOMS
 ■ STN
 ■ STIMULATION VOLUME

Figure S34. Feasibility study of *ClearTune*. The stimulation volume programmed by standard of care settings compared with the stimulation volume programmed by *ClearTune*. Fibers tractography in each patient is weighted by the strength of connection to the stimulation volume.

In addition, the study should confirm the accuracy of the streamline predictions with individual data (to the degree that DTI resolution allows).

We agree that this is a key point. However, DBS fiber filtering by design requires the same streamlines to be tested across patients (since we correlate electrical field magnitudes with clinical improvements for the same streamline across patients). So, by design, the technique is not applicable to patient-specific tractography. A similar (but still quite different) approach is termed DBS network mapping (following the approach of Horn et al. 2017 *Annals of Neurology*), which may be potentially comparable by proxy. However, based on the imaging resolution and noise in the data, we would really be testing the limits of both the approach and the data. While we were able to compare results in qualitative sense in a sample of 20 patients from which individual tractography was available (a subset of the newly added validation cohort 1), we for now only paste the results for the reviewer below. We believe that already, our manuscript is quite complex and adding this analysis could go beyond the original goal of the study. If the reviewer feels strongly that this should be included, however, we are of course happy to do so. This being said, we were excited to see the same general pattern across normative and individualized tractography, bearing in mind that the two approaches differed.

The following section and reviewer figure were not added to the manuscript, but could be, if the reviewer deems important:

Comparison with individualized tractography data

By design, DBS fiberfiltering can only be calculated on normative tract data, since the same exact streamline needs to be tested in each patient to calculate streamline-based statistics. Nonetheless, we wanted to test how results of the multi-tract model would compare on patient-specific tractography data, at least in a qualitative fashion. To do so, electrodes were localized in a third independent retrospective validation cohort in which patient-specific diffusion MRI data had been acquired, and a similar technique, DBS network mapping²⁷ was applied for each symptom cluster separately. As in the normative results, tracts that associated with

bradykinesia and rigidity improvements traversed to the STN from its lateral aspect (internal capsule), while tracts associated with tremor and axial improvements traversed from the medial aspect to the nucleus. As normative results, patient specific tractography results also suggested a stronger role for indirect connections from the pallidum to contribute to rigidity improvements. A visual head-to-head comparison between normative and patient-specific results is shown in reviewer figure 1.

Reviewer figure 1. Comparison of DBS fiber filtering results as defined by the multi-tract model (fig. 2) and DBS network mapping carried out on an independent cohort of $N = 20$ patients in which patient-specific tractography data was available (Retrospective validation cohort 3, Table 1). Enlarged panel shows patient-specific results in synopsis. In both results, we see tracts associated with bradykinesia and rigidity improvements to come from the lateral internal capsule, while tracts associated with tremor and axial improvements come from the medial aspect of the nucleus. Also note that both normative and patient specific tractography converged on a stronger role for indirect connections from the pallidum to contribute to rigidity improvements.

REVIEWER COMMENTS

Reviewer #2 (Remarks to the Author):

Authors have addressed all my comments convincingly.

Reviewer #3 (Remarks to the Author):

Rajamani (Revised)

Thank you for the opportunity to review this revised manuscript. The authors have provided a lengthy rebuttal of the previous reviewer comments, adding additional data and discussion. The manuscript is expanded and some aspects are clarified. At the same time, while seeking to refute earlier commentary, the authors have not directly addressed the core methodological concerns and seem, in their enthusiasm for this approach, to overstate the robustness of their findings.

As noted, this is a statistical tour-de-force of Lead-DBS analyses. The overarching claim is that Lead-DBS can be used to personalize STN DBS stimulation parameters by mapping implanted leads to a standardized atlas that includes labelled DTI streamlines. The premises are (1) that there are symptom specific tracts which respond favorably to DBS stimulation; and (2) that STN DBS for PD may be tailored by directing stimulation based on these individual symptoms. The authors further claim that (3) an atlas of circuit pathways accurately represents the relative locations of individual symptom-associated circuits in standard stereotactic space; and (4) that individual patient data (pre-op MRI, post-op CT, symptom characteristics) can be fed into an algorithm that outputs optimal stimulation parameters based on these internal models. These claims are explored further below.

There are numerous assumptions underlying these claims. First (A) that individual anatomy in the STN region can be reliably and accurately warped into standard stereotactic space; and (B) that the result of such warping will not only be affine but will also accurately map to an atlas with sub-millimeter precision. Similarly, (C) that the relevant tracts connecting cortical and subcortical brain regions can be identified with submillimeter precision in a MR-tractography atlas; and that (D) these anatomical relationships will be maintained after atlas-fitting at the individual patient level. In this and previous work, this influential group has not included clear statements of these and other assumptions, leading to broad claims that seem “too good to be true.” Wherever this manuscript is published, the assumptions and an assessment of their weaknesses and strength should be clearly stated and the statistical shortcomings, particularly significant confusion between accuracy and precision, must be addressed.

That said, the challenge of DBS targeting and programming is significant and important for the field, and the authors are to be commended for their efforts and for creating an easily accessible set of tools that have been broadly adopted (1000 studies!). Soberingly, the history of science and medicine is filled with

fundamentally flawed (or limited) approaches that are enthusiastically supported, and transiently and widely adopted, before abandonment. Hence, it is critical that influential authors of potentially impactful manuscripts like this group directly acknowledge and (where possible) address potential shortcomings, rather than dismiss criticisms with appeals to authority or citing broad adoption. Readers should not only understand the potential but also the limitations of this work, or many patients will suffer. In this manuscript, the limitations are greatly obscured.

A basic flaw of this study is the highly unrealistic assumption that all subjects have a “common” response to stimulation in the same anatomical location of the brain atlas.

The authors have responded with additional analysis of 93 patients and testing of 10 additional patients (20 electrodes) which showed positive correlation with empirical improvement. These results are unsurprising. They show that the patient population has, on average, average anatomy. The authors gloss over the fact that even state-of-art atlases cannot capture the diversity of unique, individual brains within the accuracy required for DBS stimulation (2 mm). This is due to the fact, well recognized by anatomists (See work by PP Mitra or S Haber) that at the mesoscopic scale (0.1 to 10 mm) there is a transition from a macroscopic level, where a stable, species-typical neural architecture is observed, to a finer scale where individual variation is prominent. This is readily apparent, though consistently ignored and de-emphasized, throughout the manuscript (as but one example, in Reviewer figure 1, even though fiber tracts are projected onto the same anatomic MRI, the differences between the individual and population means are striking).

The atlas has not been reviewed/legitimized/validated in the literature, by either anatomists or imaging people, but is a “Frankenstein” amalgamation.

The authors have responded with a favorable comparison of streamlines in the atlas and “anatomical ground-truth data”. But what are “ground-truth” data? In Table S1, the authors repeatedly cite “Expert Neuroanatomist’s Definition” or popular atlases, and include multiple photos in the supplemental material. Fair enough. However, to validate this approach, the authors not only need to know the location of, for example, the pallidothalamic tracts not only on average (assuming the exemplar used to create an atlas is representative of this) but also the variability in this location for the specific individual. This is a currently impossible task. An additional problem here is that what is known about cortico-subthalamic projections, which are proposed to be clinically relevant in this groups work (and very well may be) does not accord with the experience of clinical DBS. For example, it is known (see for example Parent and Hazrati 1995) that cortico-subthalamic projects are not sharply differentiated, and that Area 4 projections are dorsolateral while premotor (Area 6,8,9) projections are ventromedial. In LeadDBS, the medio-lateral dimension is de-emphasized. If the authors wish to claim, as they do, that tremor control arises from stimulation to Area 4 projections and rigidity/bradykinesia from premotor/prefrontal projections, then there are several vexing questions to answer: 1) Why does the atlas (Figure 2) show sharply delineated tracts all along the lateral aspect of the STN?; 2) How does stimulation at single electrode contacts at the ZI/dorsal surface of STN, at amplitudes that produce a 2-3 mm VTA routinely eliminate all cardinal motor symptoms of PD?; and 3) why does ventral STN stimulation (where they show these tracts to converge) routinely fail to produce benefit?

There has never been a published study dedicated to explicitly comparing the biophysical models used in Lead-DBS to any established standards or electrophysiological measurements.

The authors surprisingly respond with a table of publications in which the Lead-DBS package has been used. Is the logic in this response that 1000 studies cannot be wrong? A more credible response would be to acknowledge the limitations of the VTA modeling presented by the authors. In fact, the authors do not, and cannot, know the local anisotropic conductance of tissue around the DBS lead in any individual let alone on average. These models, at their best, rely upon atlases that are derived from classical studies of white and grey matter, or more recently, estimates based on low resolution DTI. This is not to say that isotropic modeling that produces a spherical VTA has no value. However, the size of the VTA is unknown within a millimeter, does not have a sharp border, and is exceedingly unlikely to affect heterogenous fiber tracts in a uniform way. Again, without a candid approach to discussing limitations clearly and explicitly in this and previous LeadDBS papers, and therefore a clearer understanding among non-specialist clinicians, overstated conclusions will continue to permeate the DBS literature.

Ridiculous assumption that the DBS lead location is precisely known for each subject, and that there is zero variability associated with that component of the model.

The authors respond that “modeling the effect of uncertainty in electrode placement is an exciting idea” and recalculate the symptom specific tract model. This response does not address the concern. As illustrated in Figure 1, there are examples at each of the centers (though most pronounced in Wurzburg and Beijing) where stimulation, according to Lead DBS projections, is very far from any location expected to have therapeutic benefit, let alone to allow for the excellent results reported in Table 1. The authors, in this and previous publications, have not addressed this limitation. In this reviewer’s experience, comparison of the Lead DBS mapping to actual patient high-resolution pre-operative MRI and post-operative CT (after pneumocephalus has resolved) often distorts the location of the active contact multiple millimeters from its original location.

The authors seek to address this issue with examples of misplaced leads in Figure S1. The result would be more convincing if (1) Axial T2 and co-registered CT were shown at a higher magnification that shows the anatomy more clearly; (2) Axial images were selected at the level of the STN and not mid-red Nucleus (which is below the STN midpoint); (3) if corresponding coronal MRI images that distinguish the STN and SNR were shown, and then these were compared to both axial and coronal Lead DBS atlas images at the level of the active contact(s). Better still, the authors should quantify the inaccuracy of atlas-fitting for randomly selected DBS patients with successful outcomes from a pool of those whose anatomy deviates from the mean (e.g. wide 3rd ventricle, narrow and broad STN width, anterior and posterior displacement of the STN midpoint). The word “ridiculous” may be too strong, but there are major incorrect simplifications and assumptions that are glossed over, increasing confusion in the field.

In response to comments about the effects of changing jitter in the localization of the DBS lead, the authors respond (Figure S28) that shifts of 2 mm result in highly correlated estimates of symptom specific tract stimulation (once again, using correlation as the statistical measure). But this in fact proves the point of the criticism—longstanding clinical experience demonstrates that 2 mm shifts in the electrode very significantly impacts clinical efficacy (which is why intraoperative testing is performed and

why MER tracts are spaced 2 mm apart). The claim by the authors that 2 mm shifts do not significantly change symptoms specific tract stimulation underscores that this analysis, with all of its assumptions, is simply unable to explain the variability in clinical outcomes with lead location that are seen in practice. That is a real problem for the overall premise of the manuscript.

Relevance of correlations to clinical DBS programming algorithm development or mechanistic understanding is minimal.

The authors respond that higher correlations are not expected due to additional sources of variability in outcomes. Here again, the authors may have not completely understood or addressed the concern. While there clearly are multiple sources of variability in DBS outcomes, and that the magnitude of these contributions may be guessed at, the authors have in their data, a number which normalizes out many of these variables—the levodopa response. ON/OFF Meds could be compared to OFF MEDS ON/OFF DBS—data that is gathered in most reputable centers. One would expect the quality of lead placement to highly correlate with the amount and variability of Levodopa response (perhaps excluding non-responsive tremor and dyskinesia scores).

What is left unaddressed by the response is the low correspondence of model estimates and clinical outcomes. The authors claim (see Figure 5) that the “Original model predicts outcomes in validation cohort.” But does it? The percentage of prediction falls in a narrow band of 0.4 to 0.6 percent improvement while the empirical data range from 0 to 0.8. The authors seek to justify the conclusion with a (low) correlation of $R = 0.37$. This conflates accuracy and precision, which should be examined separately. In this and essentially all results in this work, the means are close (i.e. the result may be accurate, though should be reported as mean/SEM of both distributions), but precision, which determines how much this approach can be applied to individuals, is clearly extremely low. There are multiple ways to present such a comparison, including the coefficient of determination (accuracy), the mean squared error (precision) or a Bland-Altman plot (both). This and previous studies from this group lack this sort of rigorous analysis to quantify the usefulness of LeadDBS.

Clartune validation: You can stimulate a subject with many different parameter settings (or different contacts) and get good/similar clinical results.

The authors have expanded the analysis from a single patient and have now performed and included a study of 5 patients. The logic here is that the system appears to work in a few cases. What is not known is how much these 5 patients differed from the mean anatomy represented in the atlas. The additional data are helpful, of course, but do not substitute for the rigorous statistical analysis described above which is needed to support the author’s very strong claims.

Additional More Minor Methodological Concerns and Comments:

The above commentary notwithstanding, the authors may argue that the LeadDBS software provides a tool to improve symptom-specific outcomes that empirically works (at least on average across a broad patient population). In this regard, it would be similar to atlas-based approaches dating Schaltenbrand–Wahren in 1977. The challenge, then and now, has not been one of population-based accuracy but of precision at the level of individual patients. Despite the volume of papers published by this influential

group, the problem of precision remains vexing.

Some additional and more minor comments/questions include:

1) The abstract does not clearly differentiate this work with past findings and makes broad claims. To minimize misunderstanding, the abstract should reflect the methods and findings of the current work. Similarly, the title should be more specific and telegraphic for the content of the manuscript.

2) Figures 2-5 and 7 do not show scale bars and it is not clear how the anatomic image background relates to the streamlines shown. This should be added/clarified.

3) Figure 1: Scale bars and orientation information are missing. The relationship of the imaged cortex to the displayed STN is unclear.

4) Figure 2: How are indirect pathway streamlines between STN and pallidum identified? In addition, it appears that all relevant tracts lie lateral to the STN, but most effective stimulation is medial—how are fiber tracts identified within and medial to the STN, which are the areas of effective clinical stimulation (and also where it is difficult if not impossible to perform accurate streamline tracing?)

5) Figure 3: The very broad representation of tremor-associated fibers in Figure 3A is difficult to reconcile with the very narrow representation in Figure 2, subsequent figures, and Reviewer Figure 1. Under the premises of the LeadDBS, it would suggest that tremor should be effectively treated across broad swathes of the posterior/anterior, superficial and deep STN. However, this is not the case. How are the associated fibers tuned?

6) Figure 4: In the figure, the patient selected for display is one of very few that happens to have an extremely accurate multi-tract and single-tract prediction. The cross-validation panel shows that the multi-tract analysis had very poor performance for individuals who had lowest quartile clinical outcomes. In general, the model predicted 0.4 to 0.6 improvement, although the actual response was -.2 to 0.4. This is exactly what would be expected, even if the atlas is correct, for patients whose lead location is distorted by atlas fitting.

7) Figure 5: As commented above, here also the precision of prediction is poor.

8) Figure 6: Here the axes values are presented at a scale that is unreadably small. There should be identical values on both axes in all panels. When this is done, it will be clear that, as in other figures, the predictions fall in a narrow range, while the empirical findings are very broad, indicating a lack of precision of the Lead DBS approach.

9) In bilateral stimulation, how are the effects of each stimulation side accounted for and incorporated into the model? Are symptom responses lateralized in the analysis? While all patients underwent bilateral STN DBS and bilateral stimulation, the effects of each side are not differentiated or methods explained in the figures or methods.

10) The LeadDBS algorithms ignore side effects. However, how is stimulation from primary motor cortex to STN distinguished from fibers that would produce motor side effects? Does application of the authors' methods allow these therapeutic and side-effect producing tracts to be distinguished? Is there any data that predicts motor side effects? This seems an important component towards validation of the overall approach.

Again, this is important work by an influential group that seeks to address very significant issues in the field. I am grateful for the opportunity to review this very detailed and extensive collection of studies, which will no doubt be published in a high-impact journal. Upon review, I believe that DBS patients and the field (again 1000 studies) will most benefit if the limitations, assumptions, and potential shortcomings of the approach are now clearly and transparently reported by the authors, so that the capabilities of Lead DBS are not overstated and misunderstood.

Points made by original reviewer 01

Points made by reviewer 03

Response by authors

Additions/Changes to the manuscript

Reviewer #3:

Thank you for the opportunity to review this revised manuscript. The authors have provided a lengthy rebuttal of the previous reviewer comments, adding additional data and discussion. The manuscript is expanded and some aspects are clarified. At the same time, while seeking to refute earlier commentary, the authors have not directly addressed the core methodological concerns and seem, in their enthusiasm for this approach, to overstate the robustness of their findings.

We would like to thank the reviewer for the very thoughtful and critical comments that helped us revise the manuscript further. We would also like to additionally thank the reviewer for also overseeing responses we made to original reviewer #01's concerns. We apologize that we were not yet able to fully convince the reviewer about the robustness of our findings and do our best to respond to remaining points. We feel that beyond a few specific additional analyses that were requested (and which we were able to gladly carry out), one main theme of the criticism seems to be in overstating claims. We now add an extensive limitations section and additionally toned down the language throughout the manuscript (see below).

A second main theme of the criticism seems to point to general concern about the use of the Lead-DBS software. The manuscript already features a replication of main results carried out with an independent software, and we now include a five-page supplementary figure that shows better views of individual cases reconstructed with Lead-DBS.

The following changes were made to tone down the language and to discuss limitations:

1. Use of the word 'predict': We apologize for this general overstatement and have changed any occurrence of the word to e.g. 'account for', 'estimate', or 'explain significant amounts of variance in'. Indeed, the reviewer is correct that our models are not capable of accurately predicting improvement values in unseen data. Rather, if at all, the model seems to be able to

predict the ranks of improvements to some degree, or, as now stated, to account for variance in improvements of unseen data. In this regard, we would like to beg the reviewer to consider the state of the art in the field: As the reviewer mentions, no DBS imaging model is currently capable of accurately predicting individual outcomes across cohorts and centers. As the reviewer notes below, this is what we all dream of since years. But given our analysis made in and around figure S35, we are less optimistic that this is generally possible. There are many sources that contribute to the variance in outcomes above and beyond stimulation location. This does not mean, that we, as a field, are not interested in the optimal stimulation location, however. Hence, we argue that a location or network that is capable of predicting the ranks within an unseen cohort is still key to identify (as done here).

2.Limitations section: We have added a more extensive and candid discussion of limitations:

“Despite this, the use of normative connectomes is inherently limited and does not include patient-specific variability of white-matter tracts. Relatedly, the use of normative tractograms includes the necessity to register patient and atlas data, which is inherently prone to inaccuracies. In other words, a patient scan can never be perfectly aligned with an atlas, despite all efforts. This leads to inaccuracies of the model and, as a function of that, to its predictive power, i.e. it biases our results toward non-significance.” – limitations, p. 24

“Next, the bioelectrical model employed here is simple compared to other methods^{59,60} and has not been directly validated using electrophysiological data. Namely, while the forward solution provided by the SimBio/FieldTrip pipeline²⁶ as employed here, solves the static formulation of the Laplace equation to estimate the electric field in an established fashion (as widely used in the EEG literature), our process ends there and we calculate statistics directly on level of this field. Our reasoning behind choosing this simpler and more probabilistic approach, which does not assume sharp borders of the stimulation field, has been described at length elsewhere^{61,62}. However, it is key to mention that more elaborate biophysical modelling pipelines have combined the volume conductor models with axonal cable models (placed orthogonally to the lead⁶³ or along pathways⁵⁹ to probe in more deterministic fashion whether axons would fire additional action potentials due to the DBS pulse. Even such models ignore the fact that GABAergic vs. Glutamatergic axons respond differently to DBS (the former fire along while the latter deplete readily⁶⁴). In addition, concepts that model axons require to pose many assumptions, such as fiber type (mixed, myelinated and unmyelinated axons), axon diameters, degree of myelinisation, degree of arborization of both dendritic and axonal terminals, number

of nodes of Ranvier to include into the model, conductivity of axonal, interstitial vs. myelin components, degree of microstructural anisotropy, heterogeneity and dispersivity of tissue conductivity, specific properties of the encapsulation layer, capacitive properties, and others. Still, more elaborate models are often deemed more biophysically plausible to the simpler approach applied here. To this end, we replicated our main results using a more elaborate pipeline that has been developed by a different team ²⁷, which calculated pathway activation models, that, when subjected to fiber filtering, produced comparable results.” – limitations, p. 25

“Next, it is possible stimulate a patient with many different parameter settings (or different contacts) and get good/similar clinical results. This matter makes demonstration of clinical utility of both out-of-sample estimates of improvements and the *Cleartune* algorithm difficult. This task is even more complicated in the present monopolar review cohort (N = 20), where only the three segments of a given contact level were compared (which are even closer to one another than different contact levels). While results seem promising and *Cleartune* was able to suggest the clinically chosen contact above chance, this general limitation still applies to any form of image guided programming.” – limitations, p. 24

“Finally, correlations between model estimates and empirical improvements are moderate. Crucially, if at all, our model is capable of estimating ranks of improvements within a given cohort, rather than absolute improvement values in individual patients. We point the reader to our modelling considerations section S5 for additional thoughts on this matter. In brief, many factors beyond electrode placement influence clinical outcomes following DBS. Critically, however, stimulation location is a key variable that can be influenced by doctors, while other factors (such as age, disease-subtype, etc) cannot. This isolates the variable of stimulation placement as a key one to improve patient care. Hence, despite the model not being able to predict improvements accurately, we argue that identifying ideal targets for given symptoms, as done here, is, while limited, still key to move forward.” – limitations, p. 26

3. Toned down language throughout the manuscript. We made numerous changes in the wording and are unable to paste all examples below. The following are examples but represent typical sentences which made claims that were now toned down in the revised manuscript:

“Here, we pursue two goals: i) creating a circuit model in stereotactic standard space for four cardinal motor symptom categories (tremor, bradykinesia, rigidity, and axial symptoms) and ii) derive and apply an algorithm that uses the model to suggest optimal stimulation parameters as a function of the baseline symptom severity profile in each patient.” – introduction, p. 5

“While three of the five patients preferred *Cleartune* over SoC settings, in two patients, *Cleartune* settings led to side-effects (dyskinesia in patient 05 and dizziness in patient 04). This emphasizes that the current model was purely driven by improvements (and not by side-effects), which is a clear limitation for clinical applicability. [...] While generally promising, given the low N, these results should not be overinterpreted. Rather, this trial was carried out to test feasibility of applying *Cleartune* in a clinical setting and to gather first experience in preparation for a proper prospective trial. As such, the trial was not powered to compare *Cleartune* vs. SoC settings (non-inferiority or superiority).” – results, p. 19-20

“It is important to clarify at this point that our results do not suggest that one symptom domain can be modulated *independently* by a specific set of streamlines. There were considerable overlaps between connections, most especially on a cortical level and along the indirect (pallidosubthalamic) projections. On the other hand, projection zones of hyperdirect (cortical) input to the STN seemed quite segregated.” – discussion, p. 21

As noted, this is a statistical tour-de-force of Lead-DBS analyses. The overarching claim is that Lead-DBS can be used to personalize STN DBS stimulation parameters by mapping implanted leads to a standardized atlas that includes labelled DTI streamlines. The premises are (1) that there are symptom specific tracts which respond favorably to DBS stimulation; and (2) that STN DBS for PD may be tailored by directing stimulation based on these individual symptoms. The authors further claim that (3) an atlas of circuit pathways accurately represents the relative locations of individual symptom-associated circuits in standard stereotactic space; and (4) that individual patient data (pre-op MRI, post-op CT, symptom characteristics) can be fed into an algorithm that outputs optimal stimulation parameters based on these internal models. These claims are explored further below.

There are numerous assumptions underlying these claims. First (A) that individual anatomy in the STN region can be reliably and accurately warped into standard

stereotactic space; and (B) that the result of such warping will not only be affine but will also accurately map to an atlas with sub-millimeter precision. Similarly, (C) that the relevant tracts connecting cortical and subcortical brain regions can be identified with submillimeter precision in a MR-tractography atlas; and that (D) these anatomical relationships will be maintained after atlas-fitting at the individual patient level. In this and previous work, this influential group has not included clear statements of these and other assumptions, leading to broad claims that seem “too good to be true.” Wherever this manuscript is published, the assumptions and an assessment of their weaknesses and strength should be clearly stated and the statistical shortcomings, particularly significant confusion between accuracy and precision, must be addressed.

We agree with the reviewer regarding these limitations and have added the following paragraphs to the limitations section:

“First, our main model applies normative tractograms instead of patient-specific tractography data to isolate symptom-associated networks. The reasons to focus on normative datasets are manifold: It is hard, if not impossible, to reconstruct thin bundles such as the ansa lenticularis, the comb fibers or the striatopallidofugal bundle based on clinical imaging since these are thin structures that traverse through gray matter and/or orthogonally to the internal capsule⁴²⁻⁴⁴ (also see section S2). However, even in normative data, these structures may not be identifiable with submillimeter precision.” – limitations, p. 23

“Despite this, the use of normative connectomes is inherently limited and does not include patient-specific variability of white-matter tracts. Relatedly, the use of normative tractograms includes the necessity to register patient and atlas data, which is inherently prone to inaccuracies. In other words, a patient scan can never be perfectly aligned with an atlas, despite all efforts. This leads to inaccuracies of the model and, as a function of that, to its predictive power, i.e. it biases our results toward non-significance. Relatedly, DBS electrode reconstructions should be seen as models that inherently include an amount of uncertainty.” – limitations, p. 24

That said, the challenge of DBS targeting and programming is significant and important for the field, and the authors are to be commended for their efforts and for creating an easily accessible set of tools that have been broadly adopted (1000 studies!). Soberingly,

the history of science and medicine is filled with fundamentally flawed (or limited) approaches that are enthusiastically supported, and transiently and widely adopted, before abandonment. Hence, it is critical that influential authors of potentially impactful manuscripts like this group directly acknowledge and (where possible) address potential shortcomings, rather than dismiss criticisms with appeals to authority or citing broad adoption. Readers should not only understand the potential but also the limitations of this work, or many patients will suffer. In this manuscript, the limitations are greatly obscured.

We are very sorry if our reasoning came across as dismissive (or appealing to authority). Original reviewer 01 had pointed out that Lead-DBS models were ‘grossly inaccurate’, which we had to politely disagree with. To address those comments, we aimed at pointing the reviewer to a large body of studies that validated specific components of the software (table S2).

We now include a more thorough discussion of the limitations of the biophysical models created within the SimBio/FieldTrip pipeline (as employed within Lead-DBS):

“Next, the bioelectrical model employed here is simple compared to other methods^{59,60} and has not been directly validated using electrophysiological data. Namely, while the forward solution provided by the SimBio/FieldTrip pipeline²⁶ as employed here, solves the static formulation of the Laplace equation to estimate the electric field in an established fashion (as widely used in the EEG literature), our process ends there and we calculate statistics directly on level of this field. Our reasoning behind choosing this simpler and more probabilistic approach, which does not assume sharp borders of the stimulation field, has been described at length elsewhere^{61,62}. However, it is key to mention that more elaborate biophysical modelling pipelines have combined the volume conductor models with axonal cable models (placed orthogonally to the lead⁶³ or along pathways⁵⁹ to probe in more deterministic fashion whether axons would fire additional action potentials due to the DBS pulse. To this end, we recalculate main findings of our study with a deterministic solution that includes axon cable models and show that results remain similar. Even such models ignore the fact that GABAergic vs. Glutamatergic axons respond differently to DBS (the former fire along while the latter deplete readily⁶⁴). In addition, concepts that model axons require to pose many assumptions, such as fiber type (mixed, myelinated and unmyelinated axons), axon diameters, degree of myelination, degree of

arborization of both dendritic and axonal terminals, number of nodes of Ranvier to include into the model, conductivity of axonal, interstitial vs. myelin components, degree of microstructural anisotropy, heterogeneity and dispersivity of tissue conductivity, specific properties of the encapsulation layer, capacitive properties, and others. Still, more elaborate models are often deemed more biophysically plausible to the simpler approach applied here. To this end, we replicated our main results using a more elaborate pipeline that has been developed by a different team²⁷, which calculated pathway activation models, that, when subjected to fiber filtering, produced comparable results.” – limitations, p. 24-25

A basic flaw of this study is the highly unrealistic assumption that all subjects have a “common” response to stimulation in the same anatomical location of the brain atlas.

The authors have responded with additional analysis of 93 patients and testing of 10 additional patients (20 electrodes) which showed positive correlation with empirical improvement. These results are unsurprising. They show that the patient population has, on average, average anatomy.

While we understand where the reviewer is coming from, we would beg to politely appeal to this point. When the reviewer writes that these results are unsurprising, a feeling of dismissal emerges, and we are somewhat surprised that the reviewer glosses over these validations as if they were obvious and contributed little, but instead focuses their review in many accounts on the fact that such a model is not theoretically possible (or ‘too good to be true’). Indeed, few if even no prior publications have created a model on N = 129 DBS patients and validated results on an independent large cohort of 93 patients.

Second, we believe that especially the additional N = 20 electrodes dataset (which features many stimulation settings per patient as requested by original reviewer #01) shows that in the majority of electrodes, a positive relationship between estimates and actual clinical responses is seen. What may have gotten lost in the bulk of results is that these settings were all tested on the same contact level (but on different segments). In our view, this shows that the model is indeed capable of resolving minute differences in stimulation settings in individual patients. Critically, this held true when correcting for stimulation amplitudes and this relationship was symptom-specific (i.e., it worked for global outcomes, but also for an adapted algorithm that focused on rigidity vs. bradykinesia). To the best of our knowledge, no comparable models have been demonstrated in the field.

We added the following paragraph that further discusses these findings:

“These patients had been implanted with directional electrodes (Boston Scientific Vercise Cartesia) and for the directional levels with best clinical response, each segment had been tested in increasing 1 mA steps until a side effect occurred or until reaching 5 mA. In addition, the omnidirectional setting (switching on all three segments) was tested in the same way. As above, we calculated predictions for each setting using the original model (from the N = 129 discovery cohort). In 17 of the 20 electrodes, rank estimates positively correlated with clinical improvements (all correlation plots with over six data points are shown in figure S32). Naturally, a one-sample t-test across these R-values was significant ($T = 4.155$, $p < 0.001$; figure. 6).” – results, p. 16

The authors gloss over the fact that even state-of-art atlases cannot capture the diversity of unique, individual brains within the accuracy required for DBS stimulation (2 mm). This is due to the fact, well recognized by anatomists (See work by PP Mitra or S Haber) that at the mesoscopic scale (0.1 to 10 mm) there is a transition from a macroscopic level, where a stable, species-typical neural architecture is observed, to a finer scale where individual variation is prominent. This is readily apparent, though consistently ignored and de-emphasized, throughout the manuscript (as but one example, in Reviewer figure 1, even though fiber tracts are projected onto the same anatomic MRI, the differences between the individual and population means are striking).

Regarding individual dMRI results: While these often differ substantially, they also do so when scanning the same brain twice. In other words, if single subject dMRI results show differences, these may be i) based on true differences in anatomy and/or ii) based on noise. This is not relevant for most dMRI test-retest studies since they work with large tracts such as the superolateral fascicle or the internal capsule. Indeed, test-retest studies in the field of DBS show that, for instance, the impact of the MRI machine is larger than the impact of the individual brain (10.3174/ajnr.A3140), and that even when scanning the same subject multiple times on the same machine, large deviations of results in the order of ~2 mm may occur (10.3171/2016.4.JNS1624). Hence, while tracts created with tractography look impressive, they do not necessarily represent the anatomical truth (also see Irontract challenge work by S Haber or the famous Maier-Hein study in Nature Comms).

Regarding the discussion of patient-specific vs. normative tractograms, we very much agree with the reviewer: There is some degree of similarity between each brain's anatomy (if not, no brain atlas would ever be helpful), and this similarity can be augmented further when precisely co-registering brains nonlinearly. Despite these efforts, a substantial degree of diversity across brains will always remain. Unfortunately, we have no means of measuring this diversity accurately. Indeed, the reviewer makes this point ('assumption C') above, where they doubt whether diffusion tractography may lend itself to define tracts with submillimeter precision.

As a bottom line: We agree (and have agreed in all other queries raised above) that these methods have their limits. The only way we see to test whether the models are meaningful *despite* these limitations is to empirically test whether i) the associated correlations are significant after corrections for multiple comparisons (which the correlation coefficients of our tract models are), whether ii) they hold when subjected to cross-validations (which our tract models do) and iii) whether they can estimate clinical improvements in unseen (out-of-sample) data (which they do in two unseen test datasets and five prospective patient cases).

We hope that the additional sections of the now extensive limitations section, our attempts to tone down language throughout the manuscript further, and responses to multiple similar points in this letter will convince the reviewer that our aim is not to gloss over inaccuracies. To directly address this point further, we added the following limitation statement that further highlights inaccuracies that are prone to the concept of spatial normalizations and the limitations in the use of normative tractograms:

“Despite this, the use of normative connectomes is inherently limited and does not include patient-specific variability of white-matter tracts. Relatedly, the use of normative tractograms includes the necessity to register patient and atlas data, which is inherently prone to inaccuracies. In other words, a patient scan can never be perfectly aligned with an atlas, despite all efforts. This leads to inaccuracies of the model and, as a function of that, to its predictive power, i.e. it biases our results toward non-significance.” – limitations, p. 24

The atlas has not been reviewed/legitimized/validated in the literature, by either anatomists or imaging people, but is a “Frankenstein” amalgamation.

The authors have responded with a favorable comparison of streamlines in the atlas and “anatomical ground-truth data”. But what are “ground-truth” data?

We have responded to this original query by including the extensive backlog and anatomical co-authors that had helped us create the atlas in the first place. Originally, we had planned to publish this work separately. While our manuscript did not use the term ‘ground-truth’ data, we refer to the head-to-head comparisons between tract visualizations and Klingler dissections, dark-field microscopy data and text-book results, which were carried out by anatomists, neurosurgeons and neuroradiologists (see list of coauthors). We do not believe that substantially better ways to validate tractography results exist but very much agree that ‘ground truth’ is a broad term. We have now added the following point to clarify limitations of the ‘ground truth’ data into our limitations section:

“Here, we created an atlas that was directly compared to anatomical data from Klingler dissections and textbook results (see section S2). While we believe this to be the only viable way to compare tractography results to ‘ground-truth’ data, it is indirect in nature and bases on visual comparisons by anatomists and neurosurgeons as the co-authors of the present work that were involved in this part of this work.” – limitations, p. 24

In Table S1, the authors repeatedly cite “Expert Neuroanatomist’s Definition” or popular atlases, and include multiple photos in the supplemental material. Fair enough. However, to validate this approach, the authors not only need to know the location of, for example, the pallidothalamic tracts not only on average (assuming the exemplar used to create an atlas is representative of this) but also the variability in this location for the specific individual. This is a currently impossible task.

We very much agree with the assumption that this is a currently impossible task and add this to the limitations section, as well:

“Furthermore, the identified tracts represent group averages, and it is currently impossible to match them to the exact tracts present in the individual patient.” – discussion, p. 24

Regarding the term ‘Expert neuroanatomist definition’, we now include the initials of Erik Middlebrooks, who created these respective tracts. Critically, however, they were further

assessed by Vanessa Milanese who has a strong track record in Klingler's dissections and neurosurgery, as well as by the team in Greece (GPS, SK, AK) together with the core authors of the study (HM, CN, AH), all with a track record of publishing about subcortical anatomy.

An additional problem here is that what is known about cortico-subthalamic projections, which are proposed to be clinically relevant in this groups work (and very well may be) does not accord with the experience of clinical DBS. For example, it is known (see for example Parent and Hazrati 1995) that cortico-subthalamic projects are not sharply differentiated, and that Area 4 projections are dorsolateral while premotor (Area 6,8,9) projections are ventromedial. In LeadDBS, the medio-lateral dimension is de-emphasized. If the authors wish to claim, as they do, that tremor control arises from stimulation to Area 4 projections and rigidity/bradykinesia from premotor/prefrontal projections, then there are several vexing questions to answer: 1) Why does the atlas (Figure 2) show sharply delineated tracts all along the lateral aspect of the STN?;

We are aware of the work by Parent and Hazrati 1995 (10.1016/0165-0173(94)00008-d), but, potentially, the reviewer may have misconceived the idea that premotor areas project to the STN from its ventromedial border. The authors write:

“The premotor cortex (areas 8, 9 and 6) also projects to the primate subthalamic nucleus. These projections terminate principally in the ventromedial sectors of the nucleus, the projection from area 6 being the most ventral and medial.”

While the authors make a statement about the terminal fields of these projections within the STN, in this article, they do not mention from which side/aspect of the nucleus the projections enter it. Indeed, as e.g. beautifully seen in the work by the single axon tracing work from the same group from 2018 (10.1007/s00429-018-1726-x), cortical projections from the STN enter the nucleus mainly from its lateral aspect (the one facing the capsule) but then traverse through the nucleus, also into its ventromedial aspects. The aforementioned beautiful atlas published by the McIntyre group (10.1016/0165-0173(94)00008-d), which Martin Parent co-authored, also maps projections all hyperdirect projections this way (the side facing the internal capsule, see e.g. figure 4 in the article by Petersen et al.). We further confirmed this notion in personal communication with both Dr. Parent (below).

[REDACTED]

Reviewer Figure 1: Response to our inquiry from Prof. Martin Parent.

We hope that these thoughts could clarify the matter but are happy to discuss further, if needed.

2) How does stimulation at single electrode contacts at the ZI/dorsal surface of STN, at amplitudes that produce a 2-3 mm VTA routinely eliminate all cardinal motor symptoms of PD?;

This is a very important point, and indeed based on clinical practice, we also anticipated that tracts could not be segregated since a single VTA would mediate all four symptoms at the same time. Upon further thought, however, we realized that a typical VTA would easily encompass all four tract systems. To preconceive this criticism, we had included a paragraph of discussion and a corresponding figure into the manuscript, which we again paste below for convenience. In the figure, panel A is particularly important to this discussion, which shows that a single well-placed electrode can very well modulate all four fiber systems (which, as the reviewer mentions, fits clinical experience):

Figure 7. Hypothetical future use of symptom-tract model. A) A well-placed, standard omnidirectional (Medtronic 3389) electrode is shown with a single stimulation volume that equally covers all symptom-specific tracts. B) A hypothetical future concept with a modern electrode (Boston Scientific Cartesia X electrode with 15 directional contacts and one omnidirectional contact) is shown. With some devices, it is possible to steer multiple stimulation volumes toward individual tracts. In our example, one volume could target tremor streamlines (potentially with a high frequency of 180 Hz). A second volume would focus on the axial/gait streamlines connecting to the PPN region (potentially with a low frequency of 25 Hz). S = Superior, A = Anterior, I = Inferior.

“It is important to clarify at this point that our results do not suggest that one symptom domain can be modulated *independently* by a specific set of streamlines. There were considerable overlaps between connections, most especially on a cortical level and along the indirect (pallidusubthalamic) projections. On the other hand, projection zones of hyperdirect (cortical) input to the STN seemed quite segregated. At first glance, this could seem contradictory to clinical experience: Indeed, the same DBS setting typically modulates many symptoms at once, seemingly with similar intensity. However, this notion does not conflict with our results: the identified tracts reside very close to one another, spanning across a region of millimeters within the sensorimotor functional zone of the STN level. As figure 7A shows, a single well-placed electrode may produce a stimulation volume that modulates all identified tracts (and hence symptoms), simultaneously. However, figure 7B shows potential use of the tract model with a modern 16-contact segmented electrode (such as the Boston Scientific model Cartesia X). Using Multiple Independent Current Control (MICC) technology, distinct stimulation volumes may be generated along the same electrode, each with different amplitudes and frequencies³¹. In the hypothetical example shown in figure 7, one could steer a first volume at high frequency (180

Hz) to the tremor streamlines and a second at low frequency (25 Hz) to the axial & gait streamlines to treat the two symptoms as optimally, as possible.” – discussion, p. 21-22

and 3) why does ventral STN stimulation (where they show these tracts to converge) routinely fail to produce benefit?

In clinical practice, ‘ventral STN DBS’ is sometimes conflated with anteromedial STN DBS (associative / limbic domains). For this reason, we are uncertain which typical scenario the reviewer exactly refers to. First, the reviewer could refer to a misplaced electrode that is too anteromedial (often referred to as too ventral in clinical practice, see panel A in the reviewer figure below), i.e., in the associative-limbic domain of the nucleus. This would be a position used for treatment of OCD, but in PD, stimulation at this site is at times associated with hypomania and other cognitive-affective side effects. If the reviewer refers to this scenario, our tracts do not converge here. Alternatively, the reviewer could refer to ventral contacts of a generally well-placed electrode (black dot in panel B). While we are not as certain that such a stimulation will always fail to produce benefit, the *Cleartune* algorithm presented here would usually pick such contacts when selecting the best contact to improve axial symptoms and gait. This matches the experience of our clinical co-authors: Indeed, such contacts often fail to produce apparent benefit especially for tremor, which could be explained by the tremor tracts (green) traversing off at that level but are often activated or at least probed when gait problems arise further down the line. Multiple papers by different groups point to an optimal general location that is pretty much in the center of the premotor/motor STN (white dot in both figure panels). This location would very much be in line with the site of most convergence of our tract model. To mention some reports by various groups that independently identified this ‘optimal general location’ using various methods:

1. Bordeaux group using AC-PC coordinates (10.1007/s00701-013-1782-1),
2. Amsterdam group using stereotactic landmarks (10.1136/jnnp-2017-316907),
3. London group using Suretune software, (10.1016/j.neuroimage.2017.07.012),
4. our own group using Lead-DBS (10.1101/2020.01.14.904615; for a review see 10.1097/WCO.0000000000000679).

Reviewer Figure 2: Different definitions of “ventral” stimulation sometimes used in clinical practice. Panel A shows an extreme position that would stimulate the limbic portion of the STN. STN-DBS electrodes implanted for OCD, for instance, go into this direction. Panel B shows the location that inferior contacts of a well-placed STN-DBS electrode for PD would typically fall into.

For now, we refrain from making changes to the manuscript related to this point for reasons of brevity and reading flow. We are happy to include these ideas if the reviewer deems important to include.

There has never been a published study dedicated to explicitly comparing the biophysical models used in Lead-DBS to any established standards or electrophysiological measurements.

While some studies exist that compared Lead-DBS models and localizations with electrophysiological markers (see our table S2), we assume the reviewer speaks of EMG studies that measure capsular effects, or evoked potentials. While we are aware of two ongoing studies of this nature with very promising results, it is true that no such study has been published. The same is true, for instance, for the models included in the FDA approved software solutions of SureTune (Medtronic) or GuideXT (BSci), or comparable research software (e.g. DBSproc).

We very much agree with the reviewer that the lack of direct head-to-head trials should be declared and added the following limitations section about this:

“Next, the bioelectrical model employed here is simple compared to other methods^{59,60} and has not been directly validated using electrophysiological data.” – limitations, page 25

The authors surprisingly respond with a table of publications in which the Lead-DBS package has been used. Is the logic in this response that 1000 studies cannot be wrong? A more credible response would be to acknowledge the limitations of the VTA modeling presented by the authors.

Our logic in mentioning this was not that 1,000 studies cannot be wrong. But the assumption that all this research is wrong, and all authors that used the software are essentially unable to see ‘its many flaws’ would sound similarly off to us. We do believe that heavy usage of a tool comes with a certain degree of scrutiny, especially if the code is open source and the user base includes sophisticated authors in the field. Many key experts use the tool on a daily basis and have adapted it in ways that show their deep understanding of the underlying methods (for instance multiple reports by the Mayo Jacksonville group, the adaptation of Lead-DBS to macaque brains by the Neurospin group; <https://www.science.org/doi/abs/10.1126/sciadv.abl5547>, the adaptation to Swine model by the Miami group; [10.1016/j.brs.2021.02.017](https://doi.org/10.1016/j.brs.2021.02.017) or the additions to the code by Enrico Opri for Lead-DBS to support unilateral cases). These are a few of many examples where ‘users’ just by the scope of their work demonstrated deep understanding of the methods and strong adaptations of the work. The tool is developed by many institutions world-wide and has gone through code review and extensive usage by key experts in the field (e.g., Till Dembek, Erik Middlebrooks, Kai Miller, Andreas Nowacki, or the methods-heavy groups of Mark Richardson & Philip Starr just to name a few). Numerous groups have their own forks and work on their own extensions (e.g., https://github.com/Brain-Modulation-Lab/ECOG_localization, https://github.com/oprienrico/leaddbs_dev).

Hence, the notion that all these groups simply do not see the ‘basic flaws’ and all make ‘highly unrealistic assumptions’ (quotes by original reviewer 01) seems somewhat biased to us, and we did not see a better way of responding to this strong criticism than to list evidence that many parts of the Lead-DBS pipeline have indeed been validated in various ways.

Table S2 does not feature a list of studies which have merely used Lead-DBS. As the reviewer mentions, this list would be much longer. Each of the listed study supports specific aspects of the tool. We added this table in direct response to the claim by reviewer 1, that “The only

comparisons that are available in the literature suggest that the simulations performed in Lead-DBS are grossly inaccurate.”, which this table demonstrates to be incorrect.

On top, we also recalculated the main model using pathway activations calculated by the much more sophisticated and independently developed software OSS-DBS. This method surpasses the state of the art of biophysical modelling (including work of the McIntyre group) in the following ways:

- 1) Patient-specific heterogeneous anisotropic volume conductor models are used. Note that anisotropic conductivity is highly relevant in the context of white matter tracts.
- 2) Tissue dispersion is accounted for by the Fourier Finite Element Method
- 3) Pathway Activation Modeling to quantify axonal responses to extracellular stimulation (in contrast to VTA / E-field or Driving Force based approximations) are used.

Nonetheless, we agree with the reviewer that a more thorough discussion of limitations of our E-field model can be helpful to many readers and added the following to the revised version of the manuscript:

“Next, the bioelectrical model employed here is simple compared to other methods^{59,60} and has not been directly validated using electrophysiological data. Namely, while the forward solution provided by the SimBio/FieldTrip pipeline²⁶ as employed here, solves the static formulation of the Laplace equation to estimate the electric field in an established fashion (as widely used in the EEG literature), our process ends there and we calculate statistics directly on level of this field. Our reasoning behind choosing this simpler and more probabilistic approach, which does not assume sharp borders of the stimulation field, has been described at length elsewhere^{61,62}. However, it is key to mention that more elaborate biophysical modelling pipelines have combined the volume conductor models with axonal cable models (placed orthogonally to the lead⁶³ or along pathways⁵⁹ to probe in more deterministic fashion whether axons would fire additional action potentials due to the DBS pulse. Even such models ignore the fact that GABAergic vs. Glutamatergic axons respond differently to DBS (the former fire along while the latter deplete readily⁶⁴). In addition, concepts that model axons require to pose many assumptions, such as fiber type (mixed, myelinated and unmyelinated axons), axon diameters, degree of myelination, degree of arborization of both dendritic and axonal terminals, number of nodes of Ranvier to include into the model, conductivity of axonal, interstitial vs. myelin components, degree of microstructural anisotropy, heterogeneity and dispersivity of tissue

conductivity, specific properties of the encapsulation layer, capacitive properties, and others. Still, more elaborate models are often deemed more biophysically plausible than the simpler approach applied here. To this end, we replicated our main results using a more elaborate pipeline that has been developed by a different team ²⁷, which calculated pathway activation models, that, when subjected to fiber filtering, produced comparable results.” – limitations, p. 25

In fact, the authors do not, and cannot, know the local anisotropic conductance of tissue around the DBS lead in any individual let alone on average. These models, at their best, rely upon atlases that are derived from classical studies of white and grey matter, or more recently, estimates based on low resolution DTI. This is not to say that isotropic modeling that produces a spherical VTA has no value. However, the size of the VTA is unknown within a millimeter, does not have a sharp border, and is exceedingly unlikely to affect heterogenous fiber tracts in a uniform way. Again, without a candid approach to discussing limitations clearly and explicitly in this and previous LeadDBS papers, and therefore a clearer understanding among non-specialist clinicians, overstated conclusions will continue to permeate the DBS literature.

We very much agree with the reviewer. For precisely this reason, we do not model VTAs in the present study, but apply statistics on the electric fields directly (i.e. our model does not assume sharp borders). We believe that modeling physics (i.e., the field) is already complicated, but may be more readily feasible than modeling biology (which requires manifold assumptions in the axonal, neuronal, glial and surrounding tissue properties, see above). The following (also see above point) has been amended to discuss this in our limitations section:

“Our reasoning behind choosing this simpler and more probabilistic approach, which does not assume sharp borders of the stimulation field, has been described at length elsewhere ^{61,62}. However, it is key to mention that more elaborate biophysical modelling pipelines have combined the volume conductor models with axonal cable models (placed orthogonally to the lead⁶³ or along pathways ⁵⁹ to probe in more deterministic fashion whether axons would fire additional action potentials due to the DBS pulse. Even such models ignore the fact that GABAergic vs. Glutamatergic axons respond differently to DBS (the former fire along while the latter deplete readily⁶⁴). In addition, concepts that model axons require to pose many assumptions, such as fiber type (mixed, myelinated and unmyelinated axons), axon diameters,

degree of myelination, degree of arborization of both dendritic and axonal terminals, number of nodes of Ranvier to include into the model, conductivity of axonal, interstitial vs. myelin components, degree of microstructural anisotropy, heterogeneity and dispersivity of tissue conductivity, specific properties of the encapsulation layer, capacitive properties, and others. Still, more elaborate models are often deemed more biophysically plausible than the simpler approach applied here.” – limitations, p. 25

Ridiculous assumption that the DBS lead location is precisely known for each subject, and that there is zero variability associated with that component of the model.

The authors respond that “modeling the effect of uncertainty in electrode placement is an exciting idea” and recalculate the symptom specific tract model. This response does not address the concern. As illustrated in Figure 1, there are examples at each of the centers (though most pronounced in Wurzburg and Beijing) where stimulation, according to Lead DBS projections, is very far from any location expected to have therapeutic benefit, let alone to allow for the excellent results reported in Table 1. The authors, in this and previous publications, have not addressed this limitation. In this reviewer’s experience, comparison of the Lead DBS mapping to actual patient high-resolution pre-operative MRI and post-operative CT (after pneumocephalus has resolved) often distorts the location of the active contact multiple millimeters from its original location. The authors seek to address this issue with examples of misplaced leads in Figure S1. The result would be more convincing if (1) Axial T2 and co-registered CT were shown at a higher magnification that shows the anatomy more clearly; (2) Axial images were selected at the level of the STN and not mid-red Nucleus (which is below the STN midpoint); (3) if corresponding coronal MRI images that distinguish the STN and SNR were shown, and then these were compared to both axial and coronal Lead DBS atlas images at the level of the active contact(s). Better still, the authors should quantify the inaccuracy of atlas-fitting for randomly selected DBS patients with successful outcomes from a pool of those whose anatomy deviates from the mean (e.g. wide 3rd ventricle, narrow and broad STN width, anterior and posterior displacement of the STN midpoint). The word “ridiculous” may be too strong, but there are major incorrect simplifications and assumptions that are glossed over, increasing confusion in the field.

We have added the requested figures showing registrations of the atlas and electrode localizations in more detail for the same selection of patients as we used before (now one full-page figure per patient in the supplements):

Section S1. Variability in Electrode Placement

Pt. Example W 04

Pt. Example W 04

Figure S1. Comparison of imaging data with 3D DBS models of example cases with suboptimal electrode placements (continued on the next pages). Each figure shows in order from top to bottom: A preoperative T2 (axial & coronal sections) with thresholded registered postoperative CT superimposed for the cases of

postoperative CT usage, the same views superimposing the segmented subthalamic nucleus (based on the diffeomorphic transform following spatial normalization and WarpDrive correction), and a 3D reconstruction of the same data (Lead-DBS output). The left columns shows the respective views in native (AC/PC registered) space, the right columns show the same data after transform into MNI space.

Pt. Example W 20

Pt. Example W 20

Figure S1 (continued).

Pt. Example A 03

Pt. Example A 03

Figure S1 (continued).

Pt. Example A 04

Pt. Example A 04

Figure S1 (continued).

Pt. Example B 20

Pt. Example B 20

Figure S1 (continued).

Regarding a quantitative comparison between manual segmentations and Lead-DBS reconstructions, this has been done extensively, before (by us: 10.1016/j.neuroimage.2018.09.061 and by others: 10.1016/j.nicl.2020.102271). However, critically, these papers quantified DICE coefficients and surface distances using the automated pipeline (ANTs multispectral effective low variance + subcortical refinement protocol used here). After the automated results, we apply manual refinements using the WarpDrive tool (10.1016/j.media.2023.103041). This tool essentially fuses the process of manual segmentations and spatial normalizations, i.e., its results become as good as the human eye (and the underlying imaging data). Indeed, since the introduction of this method, our laboratory would use WarpDrive to manually segment STN nuclei rather than drawing in the structure slice by slice. Hence, the comparison between normalizations and segmentations would be between WarpDrive and itself and is hence not sensible to carry out. We point the reviewer to a WarpDrive demo video that shows how we segmented/normalized the STN in the present manuscript (and all recent papers from our group): <https://youtu.be/EgtN168LFUI>. As requested by the reviewer, we added an additional figure showing this manually adjusted fit of the STN in a series of patients with atypical anatomy.

In response to comments about the effects of changing jitter in the localization of the DBS lead, the authors respond (Figure S28) that shifts of 2 mm result in highly correlated estimates of symptom specific tract stimulation (once again, using correlation as the statistical measure). But this in fact proves the point of the criticism—longstanding clinical experience demonstrates that 2 mm shifts in the electrode very significantly impacts clinical efficacy (which is why intraoperative testing is performed and why MER tracts are spaced 2 mm apart). The claim by the authors that 2 mm shifts do not significantly change symptoms specific tract stimulation underscores that this analysis, with all of its assumptions, is simply unable to explain the variability in clinical outcomes with lead location that are seen in practice. That is a real problem for the overall premise of the manuscript.

We agree that 2 mm shifts are significant and have shown numerous times that Lead-DBS can resolve such differences (e.g., in the main methods papers of the toolbox, or multiple other papers, see table S2).

However, it is critical to emphasize that the 2 mm shift was not applied to all leads at the same time and not in the same direction (note that the mean displacement along axes was ~0 mm). Rather, in this analysis, we introduced 3D Gaussian noise onto the group level analysis. This could demonstrate that the effect size was large enough to rediscover the same tracts when noise was added to the placements of active contacts. Stated differently, the model was robust to noise in the exact electrode placements on a group level, which was the correct analysis to carry out to empirically test the assumption laid out by the original reviewer 1. We hope that this helps in better understanding this control analysis. We added the following paragraph to clarify the analysis further:

“To test this, we iteratively recalculated the symptom associated tract model 1,000 times, each time after spatially jittering each electrical field based on a 3D Gaussian distribution with 2 mm full width half maximum. Critically, this introduced random noise to the electrode placements on a group level (not all electrodes were moved in the same direction).” – results, p. 12

Relevance of correlations to clinical DBS programming algorithm development or mechanistic understanding is minimal.

The authors respond that higher correlations are not expected due to additional sources of variability in outcomes. Here again, the authors may have not completely understood or addressed the concern. While there clearly are multiple sources of variability in DBS outcomes, and that the magnitude of these contributions may be guessed at, the authors have in their data, a number which normalizes out many of these variables—the levodopa response. ON/OFF Meds could be compared to OFF MEDS ON/OFF DBS—data that is gathered in most reputable centers. One would expect the quality of lead placement to highly correlate with the amount and variability of Levodopa response (perhaps excluding non-responsive tremor and dyskinesia scores).

This is a good idea, but unfortunately the ON/OFF data for these cases is not available to us. The goal of the study is to find the optimal stimulation target (defined as a set of streamlines) for specific symptoms. We are unsure how one could use the medication effect to help this analysis further.

Furthermore, while acquired in many centers, multiple studies have shown that the Levodopa response is, in fact, not a valid predictor of DBS response. As Zaidel et al. illustrate

(10.1002/mds.23294), the common concept that Levodopa response correlates with DBS response, may build on the fact that both share a common variable (the OFF/off UPDRS-III score). Given that both are a factor of this variable, they will strongly correlate by design. In fact, in all the cohorts we have analyzed so far, the UPDRS-III DBS/Med OFF score alone is an excellent predictor for the improvement. This is not surprising, since a variable A (baseline) will most often correlate to some degree with a variable such as A-B (total improvement) or (A-B)/A (relative improvement).

Even if the levodopa response were a good predictor (as commonly assumed in the field), it would not control for numerous of the nuisance factors we listed in figure S35 (e.g. Inter- and intrarater reliability of scores, day-to-day variance, onset age, age, comorbidities in the affective realm, brain atrophy, PD phenotype, etc). For data on this, please check our covariance structure analyses of outcome variables in one of the highest quality studies in the field (Earlystim), e.g., as outlined in the supplementary material of (10.1002/mds.28952), or the excellent long-term predictor analysis carried out by the Moro group (10.1002/ana.25994). Complex covariance relationships resulting from these and similar studies stand opposed to the notion that levodopa challenge data could control for most nuisance variables.

Instead, based on our own clinical experience, we are not surprised that electrode placement can only explain ~10% of variance in outcomes – especially when they are measured using a single long-term score taken by different raters across cohorts and centers.

However, stimulation placement is the only variable we can really influence as care providers (we cannot change the patient age, PD subtype or levodopa responsiveness, etc.). For this reason, in the present manuscript, our goal was not to create a maximally predictive model, but to define the optimal target for specific symptoms.

Adding covariates to the model (as requested by the same reviewer in the last round) did help to explain more variance but a maximally predictive model was not the focus of our study. We paste the paragraph with additional covariates again below for convenience:

“Given the moderate strength of the correlation coefficients between the estimated improvement and empirical clinical improvements, we investigated whether a linear model considering other demographic factors could explain additional variance. To do so, we fit a linear model that additionally included UPDRS-III baseline, patient age at surgery, sex, and levodopa equivalent

dose (LEDD) reduction as covariates. This model explained 25.5% of the variance in clinical improvements ($R^2 = 0.26$, $p < 10^{-6}$). The estimated improvements of the multi-tract model remained a significant regressor ($t = 3.2$, $p < 0.0017$). UPDRS-III baseline scores ($t = 3.3$, $p = 0.001$) and sex also explained significant amounts of variance ($t = 3.0$, $p = 0.03$), while the other variables did not (LEDD reduction: $p = 0.43$, age: $p = 0.39$). Of note, none of these variables may be influenced due to medical practice, with the sole exception of the electrode placement and stimulation settings, which renders the multi-tract model estimates (which are based on these factors) the critical anchor point with an opportunity to potentially improve patient care.” – results, p. 14-15

What is left unaddressed by the response is the low correspondence of model estimates and clinical outcomes. The authors claim (see Figure 5) that the “Original model predicts outcomes in validation cohort.” But does it? The percentage of prediction falls in a narrow band of 0.4 to 0.6 percent improvement while the empirical data range from 0 to 0.8. The authors seek to justify the conclusion with a (low) correlation of $R = 0.37$. This conflates accuracy and precision, which should be examined separately. In this and essentially all results in this work, the means are close (i.e. the result may be accurate, though should be reported as mean/SEM of both distributions), but precision, which determines how much this approach can be applied to individuals, is clearly extremely low. There are multiple ways to present such a comparison, including the coefficient of determination (accuracy), the mean squared error (precision) or a Bland-Altman plot (both). This and previous studies from this group lack this sort of rigorous analysis to quantify the usefulness of LeadDBS.

This is a very valid point, and we very much agree and apologize for our general overstatement to use the word ‘predict’ in the manuscript. As mentioned in the introduction, we have changed any occurrence of the word to e.g. ‘account for’, ‘estimate’, or ‘explain significant amounts of variance in’. While definitions of the word ‘predict’ drastically vary, and while the coefficient of determination (below) is positive in our test-validation (and our results fulfill all criteria to claim evidence of prediction as recommended by Poldrack et al., 10.1001/jamapsychiatry.2019.3671), we are happy to remove any claims of predictions since, as mentioned above, this was not the focus of our study. We agree with the reviewer that, to stay on the conservative side, claims of prediction should not be made given our results. Rather, if at all, the model seems to be able to predict variance in the ranks of improvements to

some degree, or, as now stated, to account for variance in improvements of unseen data. We added the following paragraph to clarify this (and have removed any claim for prediction):

“Finally, correlations between model estimates and empirical improvements are moderate. Crucially, if at all, our model is capable of estimating ranks of improvements within a given cohort, rather than absolute improvement values in individual patients.” – limitations, p. 26.

We also added the RMSE, and R^2 value for the estimates of our model on the test data ($N = 93$) in figure 5 (RMSE = 0.22, $R^2 = 0.07$).

Figure 5. Retrospective validation on long term clinical outcome data. A) The fiber distribution of the original model as shown in previous figures, B) fiber distribution when recalculating the same model on the independent test dataset ($N = 93$). C) Prediction of UPDRS-III improvements in the test set based on the original symptom associated model. S = Superior, A = Anterior, I = Inferior.

Clartune validation: You can stimulate a subject with many different parameter settings (or different contacts) and get good/similar clinical results.

This is very true, which makes the idea of an algorithm that would choose the same contacts as clinicians such a hard problem. This task is even more complicated in the Tweed cohort ($N = 20$), where only the three segments of a given contact level were compared (which are even closer to one another than different contact levels). We were excited that *Clartune* chose the correct contact way above chance even in this challenging cohort. One advantage of the cohort is, that it is cleaner, since it does not build upon retrospective long-term outcomes, but on

monopolar review data (i.e., many limitations discussed above may not apply). As the original reviewer #01 had suggested, such a cohort may represent the better test dataset to validate results, which is why we have included it in the last round of revisions.

We have added the following paragraph to discuss this issue further:

“Next, it is possible stimulate a patient with many different parameter settings (or different contacts) and get good/similar clinical results. This matter makes demonstration of clinical utility of both out-of-sample estimates of improvements and the *Cleartune* algorithm difficult. This task is even more complicated in the present monopolar review cohort (N = 20), where only the three segments of a given contact level were compared (which are even closer to one another than different contact levels). While results seem promising and *Cleartune* was able to suggest the clinically chosen contact above chance, this general limitation still applies to any form of image guided programming.” – Limitations, p. 25-26

The authors have expanded the analysis from a single patient and have now performed and included a study of 5 patients. The logic here is that the system appears to work in a few cases. What is not known is how much these 5 patients differed from the mean anatomy represented in the atlas. The additional data are helpful, of course, but do not substitute for the rigorous statistical analysis described above which is needed to support the author’s very strong claims.

We appreciate the reviewer’s concerns. Already in the last revision, we removed all claims that this prospective application would ‘validate’ our results. Rather, we mentioned that the application showed practical feasibility that the approach could be applied in a clinical setting, i.e., these results at best consist of preparations for a proper clinical trial. We had even moved most of these results into the supplementary material, to further deemphasize them. For convenience, we paste the most relevant passages that currently describe the results of the five prospective cases (but these are unchanged from the last version of the manuscript).

“In four of the five patients, *Cleartune* settings led to a higher improvement than SoC settings. In the fifth patient, improvements were comparable (36 vs. 38 points improvement). While three of the five patients preferred *Cleartune* over SoC settings, in two patients, *Cleartune* settings led to side-effects (dyskinesia in patient 05 and dizziness in patient 04). This

emphasizes that the current model was purely driven by improvements (and not by side-effects), which is a clear limitation for clinical applicability. Tracts of avoidance that code for side-effects should be added to the model in future attempts. Alternatively (and additionally), clinicians may reduce the stimulation amplitude suggested by *Cleartune* in case of side-effects (while keeping the remaining parameter choices unchanged).” – results, p. 19

“While generally promising, given the low N, these results should not be overinterpreted. Rather, this trial was carried out to test feasibility of applying *Cleartune* in a clinical setting and to gather first experience in preparation for a proper prospective trial. As such, the trial was not powered to compare *Cleartune* vs. SoC settings (non-inferiority or superiority).” – results, p. 19

“Section S4. Feasibility trial for prospective application of Cleartune in a clinical setting.

To test feasibility of applying *Cleartune* in a clinical setting, a feasibility trial was carried out in a small sample of n=5 prospective patients. This trial was designed to include a randomization step, where the patient was blinded to the administration of *Cleartune* vs. clinical settings. Clinical data, which included the pre-operative T1w, T2w, and post operative CT images was used to localize DBS electrodes in each patient. Baseline scores were taken in the stimulation and medication off states. The *Cleartune* algorithm was executed for each electrode separately, for 500 iterations each. This led to *Cleartune* settings, which were stored in the pulse generator as an additional program to the existing standard of care (SoC) setting. In the second week, *Cleartune* settings or clinical settings were applied in randomized order, each for 24 hours. Resulting UPDRS-III scores were taken after 24 hours and the respective other program was switched on to be evaluated after another 24 hours. Figure S33 summarizes the trial design. Results are documented in table S3. In multiple cases, *Cleartune* suggested higher amplitudes than tolerable, and were hence reduced by the clinical team (without altering contact choices). Table S3 reports both suggested and programmed amplitudes. From a baseline of 49.8 ± 22.1 UPDRS-III points, under *Cleartune* settings, scores improved by 34.4 ± 13.1 points ($73 \pm 11.8\%$). Under standard of care settings, scores improved by 31.8 ± 15.1 points ($65.4 \pm 12.1\%$). In four of the five patients, *Cleartune* settings led to a higher improvement than SoC settings. In the fourth patient, improvements were comparable (36 vs. 38 points improvement). While three of the five patients preferred *Cleartune* over SoC settings, in two patients, *Cleartune* settings led to side-effects (dyskinesia in patient 05 and dizziness in patient 04). While generally promising, given the low N, these results should not be overinterpreted. Rather, this trial was carried out to test feasibility of applying *Cleartune* in a clinical setting and to gather

first experience in preparation for a proper prospective trial. As such, the trial was not powered to compare *Cleartune* vs. SoC settings (non-inferiority or superiority).” – supplementary material, p. 40

Additional More Minor Methodological Concerns and Comments:

The above commentary notwithstanding, the authors may argue that the LeadDBS software provides a tool to improve symptom-specific outcomes that empirically works (at least on average across a broad patient population). In this regard, it would be similar to atlas-based approaches dating Schaltenbrand–Wahren in 1977. The challenge, then and now, has not been one of population-based accuracy but of precision at the level of individual patients. Despite the volume of papers published by this influential group, the problem of precision remains vexing.

We agree that our work consists in an atlas and are in fact honored by the comparison with the work by Schaltenbrand, (Bailey) and Wahren. In fact, our introduction points to the even earlier work by Rudolf Hassler which, qualitatively, already established many of the present findings, before:

“The notion that different symptoms of PD map to different brain regions or networks is not new^{9,10}. For instance, in seminal work by the Freiburg school of stereotaxy based on 560 ablation cases between 1950 and 1958, Hassler et al. concluded that optimal control of tremor involved lesioning a loop between cerebellum (and Mollaret triangle), the posterior nucleus ventrooralis and primary motor cortex⁹. In contrast, optimal control of bradykinesia and rigidity involved lesioning connections from pallidum to the anterior nucleus ventrooralis and a subregion of the supplementary motor area (defined by the Vogt/Hassler/Brodmann school as area 6α). Using DBS, Akram et al., among others, in addition confirmed improvements in bradykinesia and rigidity to be related to connections from premotor areas and prefrontal cortex^{11,12}. Aside from DBS or lesion data but using functional MRI, Helmich et al. associated Parkinsonian rest and, likely, action tremor with the cerebellothalamocortical circuit^{13,14,15}. ” – introduction, p. 4 (*unchanged from previous version*).

In our view, the work is innovative in comparison to textbook atlases in the following ways: We explore how well a 3D atlas that is available in stereotactic space and can be deformed to individual patients may be used to estimate outcomes in novel patient cohorts, we introduce a

surrogate optimizer for automated parameter suggestion, and apply it to estimating ranks of optimal contacts in unseen data. Another potential strength could be seen in the large multicentric cohorts studied here, which may make results more robust than the ones from smaller trials. The resulting atlas quantifies symptom-associated improvements in the space of streamlines.

As mentioned, we have largely toned-down claims throughout the manuscript and have extended a now very long and extensive limitations section to cover a more detailed narrative on study limitations. We hope that the reviewer may still find at least some aspects of our manuscript interesting enough to warrant publication.

Some additional and more minor comments/questions include:

1) The abstract does not clearly differentiate this work with past findings and makes broad claims. To minimize misunderstanding, the abstract should reflect the methods and findings of the current work. Similarly, the title should be more specific and telegraphic for the content of the manuscript.

We very much agree. As the reviewer has published in nature outlets before, they may appreciate that the length of the abstract (200 words) and title (15 words) is strongly limited. Given the extensive nature of the manuscript with a novel atlas, analyses on three cohorts of different nature and the introduction of a novel algorithm to suggest stimulation settings, we were still bound to keep the abstract somewhat general without the ability to mention the content more in a more detailed way. It now reads:

“Deep Brain Stimulation (DBS) can improve tremor, rigidity, and bradykinesia in patients with Parkinson’s disease (PD), but optimally improving each symptom may require stimulation of different white matter tracts. Here, we study a large cohort of DBS patients ($N = 237$ from five centers) to identify tracts associated with improvements in each motor symptom. Tremor improvements were associated with stimulation of tracts connected to primary motor cortex and the cerebellum. In contrast, axial symptoms associated with stimulation of tracts connected to the supplementary motor cortex and brainstem. Bradykinesia and rigidity improvements associated with stimulation of tracts connected to supplementary motor and premotor cortices, respectively. By introducing a novel algorithm that leverages these findings to suggest optimal stimulation parameters, we illustrate that these symptom-associated tracts may bear potential to

ultimately be useful in *personalizing* DBS parameters based on the symptoms most bothersome in an individual patient. Going forward, this concept may pave the way toward connectome-based personalized DBS.” – abstract, p. 3

2) Figures 2-5 and 7 do not show scale bars and it is not clear how the anatomic image background relates to the streamlines shown. This should be added/clarified.

We thank the reviewer for their comment and have added the orientation and scale information for these figures. The revised versions are pasted below:

Figure 2: Symptom-network library. Views A-C from medial. A) symptom associated tracts shown in a sagittal view from medially and magnified at the level of the STN (orange, insets, one rotated by 180 degrees, i.e., shown from laterally). Symptom associated tracts follow a rostrocaudal gradient with tremor most occipital, followed by bradykinesia, axial symptoms, and rigidity. All shown tracts significantly correlated with symptom improvements after correcting for multiple comparisons ($p < 0.05$). Note that tracts are in proximity to one another, making it

possible to modulate all of them with a single well-placed electrode (matching clinical experience). B) Symptom-associated tracts visualized separately at the STN level with the other tracts grayed out for spatial comparison. Insets represent permutation tests and 10-fold cross validation results and for each symptom tract. C) Segregation of symptoms within indirect pathway streamlines between STN and pallidum, following a similar rostrocaudal gradient. D) Cortical origins of hyperdirect projections. Streamlines associated with tremor improvements originated in primary motor cortex, whereas the ones associated with improvements in hypokinetic symptoms originated from premotor regions in a more interspersed fashion. S = Superior, I = Inferior, A = Anterior, L = Lateral, P = Posterior.

Figure 3: Anatomical considerations of circuits associated with improvements of tremor and axial symptoms. A) Tremor tracts included projections from the cerebellar nuclei (to thalamus) as well as the cortical projections from primary motor cortex (to STN), matching current pathophysiological models of tremor²¹. B) Tracts associated with axial symptoms included a brainstem connection to the pedunculopontine nucleus region. C) Segregating axial symptoms into gait vs. all other (axial) items revealed that this connection was driven by gait (and not by other axial symptoms). D) Comparison to the projection site with a matching slice from an histological atlas published by Coulombe and colleagues at z = +5.08 mm (panel adapted under the Creative Commons Attribution (CC-BY) license from Coulombe et al., 2021 *Frontiers in Neuroanatomy*²⁵). A = Anterior, L = Lateral, P = Posterior.

Figure 4: Network Blending. A) Two example patient'' stimulation volume is shown alongside the optimal streamlines associated with symptom-associated tracts. This process led to four scores, each coding for one symptom. These were linearly weighted by the symptoms prevalent in each patient (since, for instance, a patient with severe tremor would profit more from modulating the tremor streamlines) and averaged, leading to a weighted-average score that was converted to UPDRS-III improvements based on the training data. These predicted improvements significantly correlated with actual improvements ($R = 0.33, p < 0.001$, mean absolute error: 17.87%). B) Stimulation volume of the same two patient'' shown alongside the optimal streamlines associated with global UPDRS-III improvements. This fiber score (0.54; 0.20) was transformed to a predicted value of global UPDRS-III improvement based on the training data within the 10-fold cross-validation process. These predicted improvements significantly correlated with actual improvements ($R = 0.28, p = 0.01$, mean absolute error: 18.11%). The two patients illustrate two extreme cases where our model has correctly estimated the empirical outcome (patient 01) and where our model has estimated clinical improvements which deviated significantly from the empirical value. S = Superior, A = Anterior, I = Inferior.

Figure 5. Retrospective validation on long term clinical outcome data. A) The fiber distribution of the original model as shown in previous figures, B) fiber distribution when recalculating the same model on the independent test dataset (N = 93). C) Prediction of UPDRS-III improvements in the test set based on the original symptom associated model. S = Superior, A = Anterior, I = Inferior.

Figure 7. Hypothetical future use of symptom-tract model. A) A well-placed, standard omnidirectional (Medtronic 3389) electrode is shown with a single stimulation volume that equally covers all symptom-specific tracts. B) A hypothetical future concept with a modern electrode (Boston Scientific Cartesia X electrode with 15 directional contacts and one omnidirectional contact) is shown. With some devices, it is possible to steer multiple stimulation volumes toward individual tracts. In our example, one volume could target tremor streamlines (potentially with a high frequency of 180 Hz). A second volume would focus on the axial/gait streamlines connecting to the PPN region (potentially with a low frequency of 25 Hz). S = Superior, A = Anterior, I = Inferior.

3) **Figure 1: Scale bars and orientation information are missing. The relationship of the imaged cortex to the displayed STN is unclear.**

The appropriate scale bar and orientation have been added to figure 1. The cortex has been moved to a coronal slice ($y = -7.7$ mm) that is closer to the STN.

Figure 1: Electrode placement. Active contacts are visualized on a coronal slice of the cortex separately for each of the three subcohorts of the discovery cohort (total $N = 129$, left) and the three validation cohorts ($N = 93$, 10 and 5, respectively, right). Please note that orientations here refer to Superior (S), Inferior (I) and Left (L).

4) **Figure 2: How are indirect pathway streamlines between STN and pallidum identified? In addition, it appears that all relevant tracts lie lateral to the STN, but most effective stimulation is medial—how are fiber tracts identified within and medial to the STN, which are the areas of effective clinical stimulation (and also where it is difficult if not impossible to perform accurate streamline tracing?)**

Indirect projections are taken from the Petersen atlas as defined by expert anatomists (10.1016/j.neuron.2019.09.030). While the information had been in the supplementary table before, we have now further clarified this in the text, as well:

“For instance, pallidosubthalamic and pallidothalamic projections were informed based by the Basal Ganglia Pathway Atlas ¹⁵, while most other connections were defined based by the DBS Tractography atlas ¹⁴. Missing connections not represented in any atlas were reconstructed following the exact same methodology used to create the latter atlas, as described in detail elsewhere²³.” – supplementary material, p. 8

While the view in figure 1 may suggest that most stimulation contacts are medial to the STN, this is not true – many of the contacts resided along the lateral aspect of the nucleus. Finally, electric fields as modeled here extend across a larger span of anatomical tissue. We prepared the reviewer figure below to show data from this atlas (which has also been made openly available by the original authors in case the reviewer would like to expect it further):

[REDACTED]

Reviewer Figure 3: Relationship between pallidosubthalamic projections as defined by the Basal Ganglia Pathway Atlas (and as used here). As can be seen, fibers extend through the aspect of the STN.

5) Figure 3: The very broad representation of tremor-associated fibers in Figure 3A is difficult to reconcile with the very narrow representation in Figure 2, subsequent figures, and Reviewer Figure 1. Under the premises of the LeadDBS, it would suggest that tremor should be effectively treated across broad swathes of the posterior/anterior, superficial and deep STN. However, this is not the case. How are the associated fibers tuned?

Indeed, while tracts in all other figures are thresholded by significance following corrections for multiple comparisons, we aimed at showing a broader landscape of the tract set in figure 3 to demonstrate that defining the borders exactly is not feasible, dependent on power and statistical results. We apologize that this had not been made clear and have amended this clarification (see below). Please note, that each streamline carries an R-value, i.e., correlation coefficients in the narrower set are higher (and significant after FDR correction), while not all streamlines in the larger set are. We are not as confident as the reviewer that no amount of tremor can be modulated in these broader regions of the STN. Based on our clinical experience, tremor often responds to stimulation in most electrode contacts at least to some degree, but maximally responds to contacts residing in the dorsolateral aspect and border of the nucleus. The following lines have been added:

“When lowering the threshold (i.e., when including streamlines with correlation coefficients that did not reach significance after corrections for multiple comparisons), tremor tracts additionally included the decussating cerebellothalamic pathway. These exact connections have been widely implicated with tremor across a large body of the literature ^{9,12,19–21}.” – results, p. 10

“Figure 3: Anatomical considerations of circuits associated with improvements of tremor and axial symptoms. As opposed to remaining figures, tracts in this figure are not thresholded at significance after FDR correction but include a broader set of tracts to appreciate the broader distribution of symptoms to streamlines (lower threshold).” – legend of figure 3, p. 11

6) Figure 4: In the figure, the patient selected for display is one of very few that happens to have an extremely accurate multi-tract and single-tract prediction. The cross-validation panel shows that the multi-tract analysis had very poor performance for individuals who had lowest quartile clinical outcomes. In general, the model predicted 0.4 to 0.6 improvement, although the actual response was -.2 to 0.4. This is exactly what would be expected, even if the atlas is correct, for patients whose lead location is distorted by atlas fitting.

We now added an additional patient to figure 4 to emphasize a patient that our model did not estimate correctly, and, as the scatter plot shows, our model has significantly deviated from the empirical value for this patient:

Figure 4: Network Blending. A) Two example patients' stimulation volumes are shown alongside the optimal streamlines associated with symptom-associated tracts. This process led to four scores, each coding for one symptom. These were linearly weighted by the symptoms prevalent in each patient (since, for instance, a patient with severe tremor would profit more from modulating the tremor streamlines) and averaged, leading to a weighted-average score that was converted to UPDRS-III improvements based on the training data. These predicted improvements significantly correlated with actual improvements ($R = 0.33, p < 0.001$, mean absolute error: 17.87%). B) Stimulation volume of the same two patients' shown alongside the optimal streamlines associated with global UPDRS-III improvements. This fiber score (0.54; 0.20) was transformed to a predicted value of global UPDRS-III improvement based on the training data within the 10-fold cross-validation process. These predicted improvements significantly correlated with actual improvements ($R = 0.28, p = 0.01$, mean absolute error: 18.11%). The two patients illustrate two extreme cases where our model has correctly estimated the empirical outcome (patient 01) and where our model has estimated clinical improvements which deviated significantly from the empirical value. S = Superior, A = Anterior, I = Inferior.

7) Figure 5: As commented above, here also the precision of prediction is poor.

We agree as commented above have extensively discussed this matter in the revised manuscript and in our responses above.

8) Figure 6: Here the axes values are presented at a scale that is unreadably small. There should be identical values on both axes in all panels. When this is done, it will be clear that, as in other figures, the predictions fall in a narrow range, while the empirical findings are very broad, indicating a lack of precision of the Lead DBS approach.

We agree that these correlations do not necessarily qualify as predictions, but, rather as predictions of the ranks among them. As discussed above, we have now changed the wording and denote the ranks (not the actual improvements) in the correlation plots. Indeed, for the purpose of our aim (validation of the *Cleartune* algorithm), ranks are sufficient: We need the algorithm to choose the optimal stimulation setting, not to predict its improvements, accurately. Beyond changing the axes to ranks, the font size of the numbers has been increased for better readability:

Figure 6. Retrospective validation on monopolar review dataset. A) The left panel illustrates a raincloud plot where each data point represents a Spearman's correlation coefficient between predicted and empirical UPDRS-III improvements for settings in one of the 20 electrodes. All correlation plots are shown in figure S32. The right panel gives four representative examples. A red eclipse is used to represent the stimulation contact that renders the highest improvement in a given patient, while the contact chosen by the model is marked with a blue eclipse, corresponding stimulation fields are shown for the example electrodes.

B and C) To assess symptom-specificity of the model, the analysis was repeated, this time maximally weighting either bradykinesia or rigidity symptoms, respectively. Correlations across settings in the 20 electrodes were almost all positive when the model was used to predict improvements in the correct symptom, but significantly dropped when used to predict improvements in the respective other symptom. In each panel, two representative examples of correct vs. incorrect symptom pairings are given.

9) In bilateral stimulation, how are the effects of each stimulation side accounted for and incorporated into the model? Are symptom responses lateralized in the analysis? While all patients underwent bilateral STN DBS and bilateral stimulation, the effects of each side are not differentiated, or methods explained in the figures or methods.

The reviewer brings up an important point and we apologize that this had not been clarified in the manuscript, a clear oversight on our part. Indeed, this is a fundamental issue in any type of symptom mapping given the fact that two electrodes contribute to one symptom. For some symptoms (e.g. tremor) but not all (e.g. axial symptoms), hemiscores could be used. However, one would lose critical information such as head tremor & head rigidity, and ipsilateral effects on these symptoms have been demonstrated (i.e., it is not a clear one-to-one mapping between DBS to the left hemisphere and symptom improvements on the right body side, even for symptoms such as tremor). In our view, there is no single clear strategy to resolve this issue. Some studies have flipped stimulation fields to the respective other hemisphere to calculate a joint field/VTA on one hemisphere, others have analyzed results separately on each hemisphere, and finally, others have flipped stimulation fields to the respective other side as additional datapoints to augment power. In our lab, for movement disorders, we have followed this latter approach thus far (e.g., in [10.1073/pnas.2114985119](https://doi.org/10.1073/pnas.2114985119) or [10.1016/j.neuroimage.2020.117018](https://doi.org/10.1016/j.neuroimage.2020.117018)), which is the approach we applied here, as well. This was now clarified in the methods section, and we again apologize that this had not been made clear, before:

“Since two fields (from the two electrodes implanted in a given patient) code for one improvement score, following the same approach as in our prior studies^{29,61}, electric fields were mirrored to the respective other side and both used to account for the same improvement value when running mass-univariate correlations during fiber filtering (below).” – methods, p. 28

10) The LeadDBS algorithms ignore side effects. However, how is stimulation from primary motor cortex to STN distinguished from fibers that would produce motor side effects? Does application of the authors’ methods allow these therapeutic and side-effect producing tracts to be distinguished? Is there any data that predicts motor side effects? This seems an important component towards validation of the overall approach.

We agree that this is a clear limitation of the approach and warrants further steps (which are planned and in part ongoing) to improve the model. Unfortunately, no side-effect data is available for these present cohorts. We discuss this limitation in the following parts of the manuscript:

“While three of the five patients preferred *Cleartune* over SoC settings, in two patients, *Cleartune* settings led to side-effects (dyskinesia in patient 05 and dizziness in patient 04). This emphasizes that the current model was purely driven by improvements (and not by side-effects), which is a clear limitation for clinical applicability. Tracts of avoidance that code for side-effects should be added to the model in future attempts. Alternatively (and additionally), clinicians may reduce the stimulation amplitude suggested by *Cleartune* in case of side-effects (while keeping the remaining parameter choices unchanged).”

– results p. 19 (*this section has not changed from the last version of the manuscript*)

“Additionally, our model only considers improvement scores and currently ignores side-effects. As the prospective application shows, this is a clear limitation of the algorithm that limits its potential utility in clinical practice. While side-effect data was not available for this retrospective multi-center cohort, this limitation warrants additional steps to improve the model (i.e. to include tracts of avoidance that are associated with capsular effects, speech problems or cognitive/affective disturbances⁵⁸).” – discussion, p. 25 (*this section has been newly added*)

Again, this is important work by an influential group that seeks to address very significant issues in the field. I am grateful for the opportunity to review this very detailed

and extensive collection of studies, which will no doubt be published in a high-impact journal. Upon review, I believe that DBS patients and the field (again 1000 studies) will most benefit if the limitations, assumptions, and potential shortcomings of the approach are now clearly and transparently reported by the authors, so that the capabilities of Lead DBS are not overstated and misunderstood.

We would like to thank the reviewer again for their thorough work and help to further optimize our manuscript. We would like to thank the reviewer explicitly for this ‘cookie’ in the end, which felt good after the lemonade.

REVIEWERS' COMMENTS

Reviewer #3 (Remarks to the Author):

The reviewer thanks the authors for responding to earlier concerns and suggestions. With the addition of the robust limitation section and additional supplementary materials, the majority of my major concerns have been addressed. One minor comment is that conclusions (e.g. Line 183 "explained significant amounts of variance") could be tempered or clarified ("explained statistically significant amounts of variance").

Points made by reviewer 03

Response by authors

Additions/Changes to the manuscript

Reviewer #3:

The reviewer thanks the authors for responding to earlier concerns and suggestions. With the addition of the robust limitation section and additional supplementary materials, the majority of my major concerns have been addressed. One minor comment is that conclusions (e.g. Line 183 "explained significant amounts of variance") could be tempered or clarified ("explained statistically significant amounts of variance").

We thank the reviewer for their comment, and we have clarified the conclusive lines from "explained significant amounts of variance" to "explained *statistically* significant amounts of variance".

"After FDR correction, this statistically significant set of fibers revealed a distinct rostrocaudal gradient of symptom improvements at the subthalamic level (figure 2)."- results, p. 7

"Second, we subjected tract models to cross-validations. Again, all but the tremor tract model explained statistically significant amounts of variance when subjected to 10-fold cross-validations (bradykinesia: $R = 0.20$, $p = 0.02$; rigidity $R = 0.20$, $p = 0.02$; axial symptoms $R = 0.22$, $p = 0.01$, also see figure 2)."- results, p.10

"The resulting model shared a similar topography with the one created by our default pipeline and performing a k-10 cross validation yielded statistically significant correlation coefficients ($R_{\text{multitract}} = 0.40$, $p = 0.001$; $R_{\text{singletract}} = 0.34$, $p = 0.03$; figure S29)."- results, p. 10

"To control for subcohorts within the discovery cohort, we reran the original model and applied a mixed-effects model that controlled for dataset as a random effect. Results were similar and remained statistically significant ($R = 0.30$, $p = 0.0015$)."- results, p .10

“UPDRS-III baseline scores ($t = 3.3$, $p = 0.001$) and sex also explained statistically significant amounts of variance ($t = 3.0$, $p = 0.03$), while the other variables did not (LEDD reduction: $p = 0.43$, age: $p = 0.39$).”- results, p. 15

“.... Naturally, a one-sample t-test across these R-values was statistically significant ($T = 4.155$, $p < 0.001$; figure 6).” - results, p.16